# Evaluating flow modulating treatment response in intracranial aneurysms using black-blood MRI in vitro

Mariya S. Pravdivtseva [1] ✉, Hivnu Toraman[2], Jana Korte[3,4], Franziska Gaidzik [3,4], Oluwabusayo A. Oni[5], Philipp Berg [3,6], Prasanth Velvaluri[7], Lana Bautz[1], Fritz Wodarg[2], Jan-Bernd Hövener[1,8], Olav Jansen[2] & Naomi Larsen[2]

## Abstract

**Background** Changes in blood flow after brain aneurysm treatment are linked to treatment success. While 4D flow magnetic resonance imaging (MRI) can measure these changes, it is sensitive to metal artifacts from implants. Black-blood MRI, less affected by artifacts, may complement 4D flow MRI. We investigated whether changes in black-blood signal reflect reductions in blood flow and could indicate the success of aneurysm treatment.

**Methods** We performed 22 flow experiments using 3D printed models of two patient-derived brain aneurysms and two straight vessels. Flow-modulating devices, including flow-diverter stents and intrasaccular devices, were inserted into 15 aneurysm models, while untreated models served as controls. All models were imaged with 4D flow and black-blood MRI on a 3 T clinical system. Numerical flow simulations were also performed. Blood flow velocity and black-blood signal were compared between treated and untreated models using non-parametric statistical tests, and their relationship was evaluated with correlation analysis.

**Results** Here we show that the black-blood signal in straight vessels decreases with increasing velocity (rho = −0.92, p-value = 3.29E-04). Implanted devices reduce blood flow within aneurysms while leaving flow in the parent vessels essentially unchanged. Treated aneurysms exhibit a significant increase in black-blood signal, which correlates inversely with measured velocity (rho = −0.82, p-value = 6.24E-04). The experimentally observed flow reductions match the numerical simulations. Metal artifacts are more pronounced on 4D flow compared to black-blood MRI.

**Conclusions** Black-blood MRI may serve as a surrogate marker of blood flow reduction after aneurysm treatment, particularly in cases where metal artifacts limit conventional imaging.

## Plain language summary

Brain aneurysms are bulges in blood vessels that can cause life-threatening bleeding if they rupture. Implanted devices such as stents or coils can be used as treatment to reduce blood flow inside the aneurysm and promote healing, but they can also fail to work. Therefore, it is important to monitor flow changes after treatment. We tested an imaging technique called black-blood MRI, using 3D printed aneurysm models with and without implanted devices. The used method showed that areas with reduced blood flow after treatment exhibited a stronger MRI signal, whereas normal vessels remained unaffected. This suggests that black-blood MRI could provide a way to assess treatment success and complement other methods. However, future studies in patients are needed to confirm its usefulness.

An intracranial aneurysm (IA) is a life-threatening cerebrovascular condition[1] that can be treated endovascularly using flow-modulation devices (FMDs)[2]. These devices reduce intra-aneurysmal blood flow and promote aneurysm healing through neck endothelialization[2] and thrombosis[3], ultimately leading to the occlusion of the aneurysm. Although most IAs are successfully occluded within 6–12 months after FMD treatment, some aneurysms may continue to fill with blood, enlarge, or even rupture after the treatment[4,5]. This highlights the need for close follow-up examinations to identify treatment failure early. It has been shown that insufficient reduction of blood flow velocity within the aneurysm after FMD

[1]Department of Radiology and Neuroradiology, Section Biomedical Imaging, University Hospital Schleswig-Holstein (UKSH), Kiel University, Kiel, Germany. [2]Department of Radiology and Neuroradiology, UKSH, Kiel University, Kiel, Germany. [3]Research Campus STIMULATE, University of Magdeburg, Magdeburg, Germany. [4]Department of Fluid Dynamics and Technical Flows, University of Magdeburg, Magdeburg, Germany. [5]Carle Illinois College of Medicine, University of Illinois Urbana-Champaign, Urbana, IL, USA. [6]Department of Medical Engineering, University of Magdeburg, Magdeburg, Germany. [7]Chair of Inorganic Functional Materials, Kiel University, Kiel, Germany. [8]Molecular Imaging North Competence Center (MOIN CC), Kiel, Germany. ✉e-mail: mariya.pravdivtseva@rad.uni-kiel.de

placement is associated with treatment failure[6], whereas a high reduction (e.g., >35%) was related to successful occlusion[7]. Therefore, immediate velocity reduction (IVR) may serve as a diagnostic marker for evaluating aneurysm treatment success.

IVR can be estimated preoperatively with no risk to patients using computational fluid dynamics (CFD) simulations based on patient-specific vascular geometries. CFD-derived IVR has been shown to potentially predict and replicate aneurysm treatment outcomes in vivo[7–10]. However, the actual post-implantation performance of FMDs may differ from pre-treatment simulations due to individual patient-specific factors such as blood velocity, viscosity, and heart rate. Furthermore, the accurate deployment of virtual implants in numerical models is limited by the absence of detailed "device/vessel wall/blood flow" interactions, which are not yet fully modeled[11].

Thus, an in vivo assessment of IVR after FMD placement is crucial for evaluating treatment efficacy. Clinically, the flow-modulation effect can be evaluated through the contrast agent stasis observed on digital subtraction angiography (DSA) within the aneurysm, with increased stasis correlating with better long-term outcomes[12,13]. The stasis can be quantified by measuring the washout time (WOT), the time it takes for the contrast agent to exit the aneurysm sac. However, the accuracy of WOT can be dependent on the contrast injection technique and includes exposure to harmful radiation. In cases of contrast agent injection, "spike-like" artifacts have been described[13], while a large variation in WOT—up to 50%—was observed in an in vitro study with manual injection[14]. Additionally, DSA provides only 2D projections, introducing an additional source of errors. In some cases, it is not possible to obtain a clear projection of the aneurysm without overlapping with other arteries or veins[13]. Moreover, there is no clear correlation between the flow reduction detected with DSA-based optical flow methods and aneurysm occlusion[15]. Assessing flow patterns or stasis in 3D might solve this problem by providing comprehensive information on the flow patterns inside the aneurysm and increasing diagnostic accuracy.

Another approach to assess IVR is through 4D flow MRI[16], which offers time-dependent 3D, voxel-wise blood flow velocity measurements. IVR measured with 4D flow MRI has been associated with successful aneurysm treatment outcomes[17]. Despite its potential, 4D flow MRI faces challenges, including metal artifacts from FMDs[16] and long examination times.

Spin-echo-based black-blood (SE BB) MRI is less sensitive to metal artifacts than gradient-echo-based 4D flow MRI. Fast-flowing blood appears dark on SE BB MRI, while slow-flowing blood appears bright. This known flow-dependent contrast mechanism can be leveraged to visualize regions of slow flow, making it a promising tool for assessing IVR immediately after FMD placement and potentially predicting treatment outcomes. Prior studies have shown areas of increased SE BB MRI signal corresponding to the slow flow regions in untreated aneurysms in vitro[18,19]. Moreover, SE BB MRI was employed to visualize long-term IA healing after coiling[20,21], stenting[22,23], and aspirin therapy[23].

To our knowledge, the influence of FMDs on the SE BB signal immediately after treatment remains unexplored. We hypothesize that the SE BB signal increases after FMD placement, reflecting IVR, and may serve as a 3D surrogate for flow reduction. This may complement velocity measurements obtained from 4D flow MRI, particularly in cases where metal artifacts prevent reliable velocity assessment within the aneurysm sac.

In this study, we investigate the relationship between intra-aneurysmal BB signal and velocity immediately after FMD placement using SE BB and 4D flow MRI, as well as numerical simulations. We use patient-derived in vitro models to isolate the effects of FMDs on flow, thereby eliminating coagulation-related variables. We assess the influence of flow rate and sequence parameters on the SE BB signal in aneurysms and vessels. Our results show that the BB signal increases in the presence of FMDs while measured and simulated velocities are reduced, suggesting that SE BB MRI can serve as a 3D surrogate for flow reduction and complement 4D flow MRI, particularly in cases affected by metal artifacts.

## Materials and methods

This study used retrospectively obtained 3D rotational angiographic (3D-RA) data from patients with IAs who had undergone endovascular treatment at the Department of Radiology and Neuroradiology, University Hospital Schleswig-Holstein, Campus Kiel. The use of these imaging data to generate in vitro aneurysm models was approved by the Ethics Committee of the Faculty of Medicine at Kiel University (reference numbers D 576/18 and D 508/18). All patients provided broad written informed consent permitting the use of their anonymized imaging data for research. The study was conducted in accordance with the Declaration of Helsinki. No additional patient intervention was required.

### In vitro vascular models

**Straight vessels**. Two straight tubes with inner diameters (IDs) of 4 and 6 mm were designed (Fusion 360 2.0, Autodesk, USA). Before considering complex IA models, straight vessels were used to assess the dependence of the SE BB signal on velocity as well as on the sequence parameters of SE BB MRI.

**Intracranial aneurysm flow models**. Two patients with unruptured IAs were retrospectively identified from the institutional MR imaging database: (1) Patient 1: female, aged between 55 and 60 years, paraophthalmic left internal carotid artery (ICA) aneurysm (max. dimension 14 mm); (2) Patient 2: female, aged between 60 and 65 years, basilar tip aneurysm (max. dimension 6 mm). The inclusion criteria comprised: (1) saccular aneurysm > 5 mm; (2) availability of 3D-RA datasets; and (3) indication for treatment with a flow-diverter stent or intrasaccular device.

Based on the corresponding in vivo 3D-RA (Allura XperFD 20/10, Philips Healthcare, The Netherlands, reconstructed with an isotropic voxel size of $(0.27 \text{ mm})^3$) from patients 1 and 2, patient-derived IA flow models of the internal carotid and basilar arteries (ICA model and BA model; Fig. 1a, b) were designed by a research fellow with 6 years of experience in intracranial flow imaging using in-house[24] protocol.

In brief, the patient-derived IA models were constructed as follows: (1) the vessel lumens were segmented from the 3D-RA dataset (threshold-based segmentation followed by marching cubes[25,26], MevisLab 3.5.0, MeVis Medical solution, Germany); (2) the resulting lumens were optimized for 3D printing by cutting all branches smaller than 1 mm in diameter, smoothing the rudiments, and closing holes (Meshmixer 3.5, Autodesk, USA); and (3) an outer layer with 3 mm thickness was added to form an artificial vessel wall. MRI-visible markers were added to ensure reproducible planning of the MRI examination, and flow connectors were applied to integrate the models into a flow loop, as illustrated in Supplementary Fig. 1 (Fusion 360 2.0, Autodesk). More details were published elsewhere[14,24,27].

The aneurysm of the ICA model was kept completely patient-specific with aneurysm sac dimensions of $14.1 \times 7.7 \times 13.4 \text{ mm}^3$ (height length, neck and dome width diameters, Fig. 1 a). The aneurysm of the BA model was modified four times (Fig. 1a) so that different sizes of intrasaccular devices could be placed[14] (Fig. 1c). To modify the BA model the following steps were applied: (1) the distal branches were cut at about 3 cm distally from the aneurysm neck and the patient-specific aneurysm sac was removed (Meshmixer 3.5, Autodesk, USA); (2) four artificial aneurysm sacs were combined with the patient-specific vessel lumen (Fusion 360 2.0, Autodesk, USA). The diameters of the resulting artificial aneurysms (height length, neck and dome width diameters) were $3.5 \times 2.7 \times 3.2 \text{ mm}$ (BA model 1), $6.9 \times 2.8 \times 3.3 \text{ mm}$ (BA model 2), $8.4 \times 6.7 \times 8.4 \text{ mm}$ (BA model 3), and $16.4 \times 9.2 \times 10.2 \text{ mm}$ (BA model 4), respectively. The artificial aneurysms were wide-necked bifurcation aneurysms with a comparable dome-to-neck ratio $(1.2 \pm 0.1)$.

All models were 3D printed using stereolithography and possess rigid walls (Clear Photoreactive Resin composed of a methacrylated oligomer, methacrylated monomer, and photoinitiator; From 3B, Formlabs, USA).

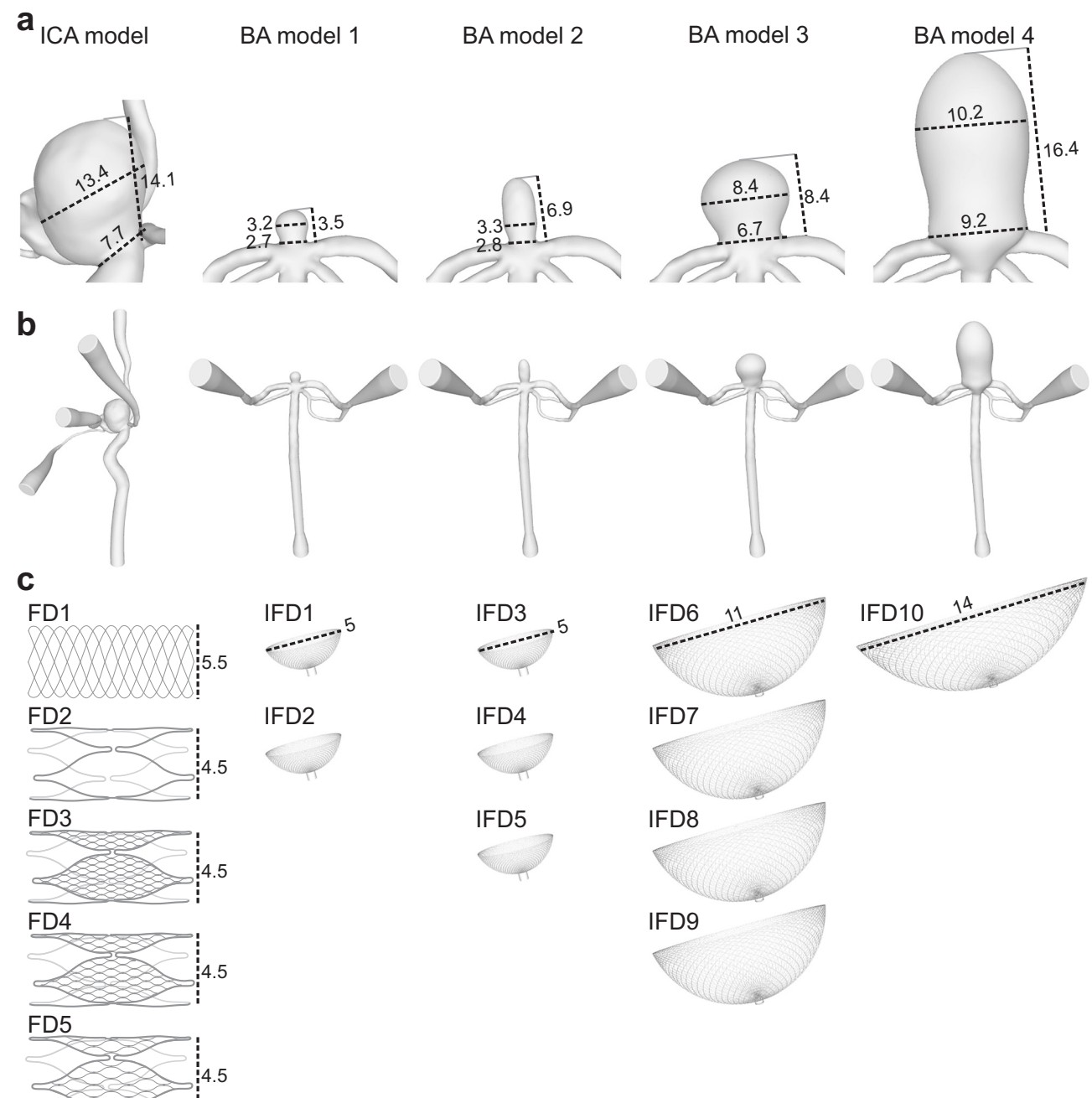

**Fig. 1 | 3D renderings of patient-derived aneurysm models of the internal carotid artery (ICA) and basilar artery (BA), together with schematic illustrations of the flow modulation devices (FMDs) used in this study. a** Close-up views of the aneurysm sacs with corresponding size measurements. **b** Full models including the feeding (parent) arteries and relevant distal branches. **c** Schematic illustrations of FMDs used to mimic endovascular treatment: flow-diverter stents (FD1–FD5) were deployed in the ICA models, whereas intrasaccular flow-disrupting devices (IFD1–10) were placed in the BA models. All dimensions in the figure are given in mm.

## FMDs and in vitro deployment

Five flow-diverter stents (FD1-5) and ten intrasaccular flow-disrupting devices (IFD1-10) were investigated in this study (Table 1, Fig. 1c).

FD1 was a commercial braided flow-diverter stent with a diameter of 5.5 mm and a length of 25 mm (Derivo, Acandis GmbH, Germany). FD2-FD5 were in-house-fabricated thin-film nitinol FDs, all featuring the same diameter of 4.5 mm and length of 43 mm, but different cell areas ranging from 8.6 to 0.1 mm², respectively[27]. FD1-5 were deployed into ICA models. FD1 was placed under fluoroscopy (Allura Xper FD, Philips) by an experienced neuroradiologist (>10 years of experience). FD2-5 were placed manually into the ICA model through the opening in the aneurysm sac[27].

IFD1-10 were commercial intrasaccular devices (Contour Neurovascular System, Stryker Corporation, USA) featuring three sizes of nominal diameters—5 mm (IFD1–5), 11 mm (IFD6–9), and 14 mm (IFD10). IFD1-10 were deployed into BA models selected based on the manufacturer's recommendations under fluoroscopy (BA model 1: IFD1–2; BA model 2: IFD3–5; BA model 3: IFD6–9; BA model 4: IFD10).

## Experimental flow setup

Before MRI experiments could be carried out, all models were submerged in an agarose gel with a 3% concentration of agar (Special Ingredients Ltd, UK) and connected to a closed-loop flow system consisting of a fluid reservoir, pump, pressure and flow sensors, and tubing. The pump and fluid reservoirs

**Table 1 | List of flow-modulation devices used in the study and their characteristics**

| Type of device | Abbreviation | Diameter [mm] | Length [mm] | Cell-area [mm²] | Commercial name, Company/Developer | Aneurysm model |
|---|---|---|---|---|---|---|
| Flow-diverter stent | FD1 | 5.5 | 25 | - | Derivo, Acandis GmbH | ICA model |
| | FD2 | 4.5 | 43 | 8.6 | Backbone, Velvaluri et al.[27] | |
| | FD3 | 4.5 | 43 | 0.1 | Mesh-1, Velvaluri et al.[27] | |
| | FD4 | 4.5 | 43 | 0.2 | Mesh-2, Velvaluri et al.[27] | |
| | FD5 | 4.5 | 43 | 0.5 | Mesh-3, Velvaluri et al.[27] | |
| Intrasaccular device | IFD1-2 | 5 | - | - | Contour Neurovascular System, Stryker Corporation | BA model 1 |
| | IFD3-5 | 5 | - | - | | BA model 2 |
| | IFD6-9 | 11 | - | - | | BA model 3 |
| | IFD10 | 14 | - | - | | BA model 4 |

*FD* flow-diverter stent, *IFD* intrasaccular flow-disrupting device, *ICA* internal carotid artery, *BA* basilar artery.

were located in the operator room. The printed models were placed at the isocenter of the MRI scanner and connected to the pumps through 3-meter-long reinforced tubing with an ID of 6 mm routed through the wall access port. Near the model inlet and outlets, the tubing was transitioned to silicone tubes to facilitate sensor placement and flexibility. The setup scheme is illustrated in Supplementary Fig. 2a.

The straight and ICA models were perfused with a glycerol-water mixture (40:60 by volume, solution viscosity 3.72 cP[28]) using a pulsatile displacement piston pump (PD-1100, BDC Laboratories). The BA models were perfused with pure water using a peristaltic pump (Ismatec MCP Standard, Cole-Parmer, USA), consistent with our previous work evaluating contrast washout in the same models using DSA[14]. The experimental fluids contained 0.3 mmol/l of gadobutrol (Gadovist, Bayer Vital, Germany).

The mean flow rate were set by adjusting the pump speed and were (1) straight vessel (ID 4 mm)—0, 0.7, 1.4, 3, and 4.7 ml/s, (2) straight vessel (ID 6 mm)—0, 1, 2.3, 4.8, and 5.4 ml/s, (3) ICA models—0 and 4.5 ml/s, (4) BA models—2.4 ml/s. The cardiac cycle period was 0.8 s for ICA/straight models and 1.26 s for BA models. The mean supplied flow rates were selected to match physiological flows observed for the intracranial vessels in vivo[29]. Pressures were not actively regulated, but any excess pressure was relieved manually through the reservoir valve (Supplementary Fig. 2). The supplied flow rate was measured at the inlets of all models using a clamp-on ultrasonic sensor (ME8PXL-M12, Transonic System Inc., USA). The representative flow profiles are shown in Supplementary Fig. 3. Pressure values were recorded using pressure transducers (Press-N-000, PendoTech, USA) installed at tubing junctions near the IA model inlets and outlets via Luer-lock connectors. The representative pressure profiles are shown in Supplementary Fig. 4. The flow and pressure measurements were performed in the MRI operator room (Supplementary Fig. 2b).

### MRI in vitro
**MRI acquisition.** MRI was performed on a 3 T whole-body MR system with a 32-channel head coil (Ingenia CX, R5 V6.1, Philips Healthcare). The protocol included time-of-flight (TOF) MRI, 2D phase-contrast (PC) MRI, 4D flow MRI, and SE BB MRI. Imaging parameters are summarized in Table 2.

TOF MRI was performed using a T1-weighted (T1w) gradient-echo sequence (GRE) with four slabs to plan the subsequent sequences. 2D phase-contrast (PC) and 4D flow MRI were acquired with T1w spoiled fast GRE PC sequence with one and three velocity encoding directions, respectively. The 4D flow MRI sequence employed non-symmetric 4-point velocity encoding to acquire velocity in all three spatial directions with flow compensation (Philips Healthcare). The readout encoding direction for 4D flow MRI was set parallel to the parent vessel for the ICA model without devices and with FD1 in the experiments described in Sections "BB signal dependence on flow rate and sequence parameters in straight vessels" and "BB signal dependence on sequence parameters in aneurysm models". In the

further experiments (Section "Impact of flow-modulation fevices on BB Signal in aneurysms"), which included the ICA model without devices and with FD1–FD5 and all BA models, the readout was set perpendicular to the parent vessel. Velocity in the straight vessels was measured using 2D PC MRI, with the imaging plane positioned at the middle of the tube (Fig. 2a, left). Velocity in the IA models were evaluated with 4D flow MRI, with the imaging volume covering the aneurysm sac, the parent vessel, and partially the outlet branches. To determine the appropriate velocity encoding (VENC) values, a series of 2D PC MRI examinations were conducted with VENCs ranging from 10 to 200 cm/s. Final VENC settings were chosen to be approximately 10% higher than the maximum measured velocity for each condition. Specifically: (1) Straight vessel with ID of 4 mm: VENCs of 10, 20, 60, 120 and 175 cm/s for flow rates of 0, 0.7, 1.4, 3, and 4.7 ml/s respectively; (2) Straight vessel with ID of 6 mm: VENCs of 10, 15, 40, 65, and 85 cm/s for flow rates of 0, 1, 2.3, 4.8, and 5.4 ml/s, respectively; (3) ICA models: without FMDs: VENC = 80 cm/s, with FD1: 20, 25, 100, 120 cm/s, with FD2: 80 cm/s, FD3–FD5: 50 and 120 cm/s. Multiple VENC settings were used for some models to optimize velocity sensitivity in both the aneurysm sac and parent vessel regions; (4) BA models: VENC = 75 cm/s.

SE BB MRI was performed using a T1w, black-blood, 3D variable-refocusing flip angle turbo spin-echo (TSE) sequence (Volume ISotropic Turbo spin echo Acquisition [VISTA][30]) with and without motion-sensitized driven-equilibrium (MSDE) preparation[31]. Two frequency-encoding (FE) directions were considered: perpendicular (transverse) and parallel (sagittal) to the straight vessel model or parent vessel of the aneurysm models. The parameters of BB MRI were varied to assess the dependence of the BB signal on different MRI settings. In total, four SE BB MRI sequences were performed: (1) $BB_\perp$ - SE BB without MSDE with FE direction perpendicular to the flow; (2) $BB_\perp MSDE$ – SE BB with MSDE with FE direction perpendicular to the flow; (3) $BB_\parallel$ - SE BB without MSDE with FE direction parallel to the flow; and (4) $BB_\parallel MSDE$ – SE BB with MSDE with FE direction parallel to the flow. Straight vessels and ICA model with FD1 were imaged with $BB_\perp$, $BB_\perp MSDE$, $BB_\parallel$, and $BB_\parallel MSDE$. ICA models with FD2-5 were imaged with $BB_\perp$ and $BB_\perp MSDE$. All BA models were imaged only with $BB_\perp$.

### MRI processing
**2D PC and 4D flow MRI.** Magnitude and phase datasets were reconstructed on the MRI system. Linear phase offset correction, velocity aliasing, and vessel masking were performed using GTflow (V3.1.12, Gyrotools, Switzerland). 2D PC MRI resulted in a 2D cross-sectional plane of the vessel. A spline region-of-interest (ROI) was manually drawn, outlining the flow lumen of the straight vessel. In 4D flow MRI datasets, five equidistant, cross-sectional 2D planes were placed in the parent vessel and aneurysm sac (Fig. 2a). MRI-visible markers integrated into the models (Supplementary Fig. 1) were used to ensure consistent placement of evaluation planes across multiple identical models. The

**Table 2 | MRI sequence parameters for the in-house TOF, 2D PC, 4D flow, and SE BB MRI sequences modified from the vendor protocols**

| MRI sequence parameter | TOF MRI | 2D PC MRI | 4D flow MRI | BB MRI |
|---|---|---|---|---|
| Sequence type | 3D GRE | 2D RF-spoiled GRE | 3D RF-spoiled GRE | 3D TSE |
| velocity encoding | - | PC, 1 velocity direction | PC, 3 velocity directions | - |
| Temporal phases | 1 | 20 | 20 | 1 |
| TE/TR [ms] | 5.8/25 | 7.2 (5.2)/11(8.3) | 5.0/8.3 | 29-35/700 |
| FOV [mm³] | 200 × 200 × 160(120) | 200(180) × 200(180) × 2 | 110 × 110 × 40 | 200 × 250 × 160 |
| vox. size [mm³] | 0.5 × 0.5 × 0.5 | 0.5 (1) × 0.5(1) × 2 | 0.75 × 0.75 × 0.75 | 0.65 × 0.55 × 0.65 |
| Flip angle [°] | 20 | 10 | 8 | 90 |
| NSA | 1 | 3(2) | 1 | 2 |
| CS factor | 4.7 | 2.5 | 4.5 | 6.5 |
| Sequence duration [min] | 17.5 (13.2) | 3.1(0.5) | 26.1 (41.4) | 5.1 |

The values in parentheses indicate the parameters used for the MRI acquisition of BA models, if they differ from those used for all others.
*TOF* time-of-flight, *BB* black-blood, *GRE* gradient echo, *RF* radiofrequency pulse, *TSE* turbo spin echo, *PC* phase-contrast, *TR* repetition time. *TE* echo time, *FOV* field of view, *vox. size* acquired voxel size, *NSA* number of scan averages, *CS factor* acceleration factor of the compressed SENSE technique.

**Fig. 2 | Evaluation planes in 2D phase-contrast and 4D flow MRI, with volumes of interest (VOIs) for black-blood (BB) MRI used in post-processing. a** 2D evaluation planes were positioned in the middle of the straight vessel and within the parent vessel and aneurysm sac within the internal carotid artery (ICA) and basilar artery (BA) models. **b** BB signal was evaluated in computer-aided design (CAD) VOIs created in the CAD models of the straight vessel (green), parent vessels (green), and aneurysm sacs (red).

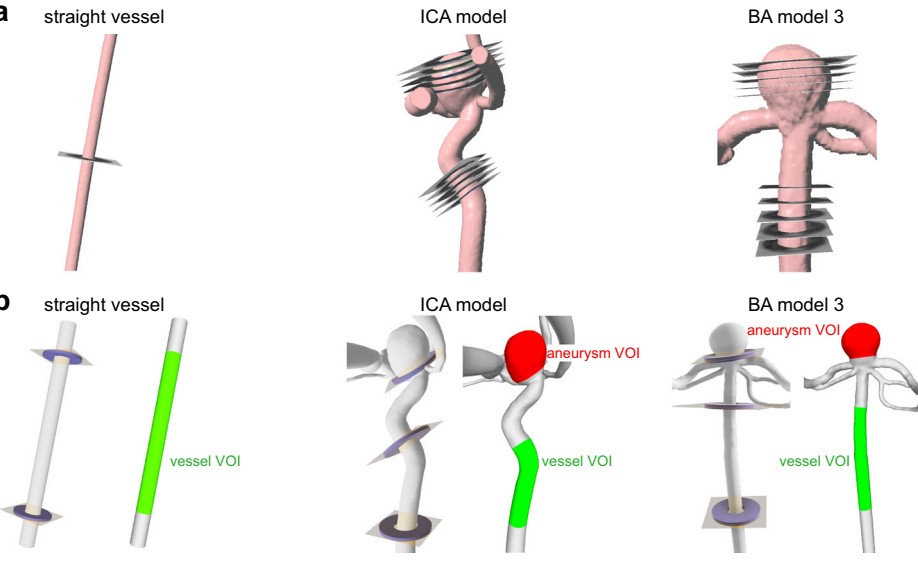

central plane was aligned with the marker, with two additional planes placed equidistantly above and below. In BA models 1 and 2, the marker was obscured by metal artifacts; thus, only artifact-free regions above the marker were used for analysis. The distance between planes within a parent vessel was set to 1.5 mm for the ICA model and 2 mm for BA models. In the aneurysm sac this value varied depending on the aneurysm size and area unaffected by metal artifacts, specifically, for ICA model it was 1.5 mm, for BA model 1 and 2–0.5 mm, for BA model 3 and 4–1 mm. The vascular lumen was manually outlined in five predefined 2D planes to create 2D ROIs. For each temporal phase, velocity values were spatially averaged within each ROI and then averaged across all planes (i.e., one plane for 2D PC MRI and five planes for 4D flow MRI). Finally, the temporal median velocity was calculated for each model due to the skewness of the velocity data. Qualitatively, velocity maps are presented using time-averaged velocity magnitude data (Matlab R2022a, The MathWorks, USA).

**SE BB MRI**. BB MRI signal was quantified in volumes of interest (VOIs) in the vessel and aneurysm (Fig. 2b). First, VOIs were defined on a

computer-aided design (CAD) of straight vessel and aneurysm models. The size and location of these VOIs were selected to ensure consistent placement across all models and to exclude areas affected by metal artifacts. All VOIs included the locations where flow was quantified using either 2D PC or 4D flow MRI. To ensure that the BB signal is quantified in the same locations for the IA models with and without devices, the same VOIs were translated into BB MR images (Fig. 3) as follows:

1. The vascular lumen of the vessel models was segmented from TOF MRI data using a threshold-based region-growing algorithm[25]. The resulting mask was connected to a 3D mesh using a marching cubes algorithm[26] (Fig. 3a, MevisLab). The TOF lumen was aligned with a CAD lumen using an iterative closest point algorithm (ICP)[32] with an average alignment error of 0.11 to 0.86 mm, resulting in the transformation matrix from the TOF to CAD coordinates – $\{T_{lumen}\}$ (Meshlab v2022.02, ISTI-CNR, Italy).

2. The vascular wall was segmented from TOF and BB MRI data using a threshold-based region-growing algorithm[25] followed by marching cubes[26] similar to the vascular lumen segmentation (Fig. 3b, MevisLab).

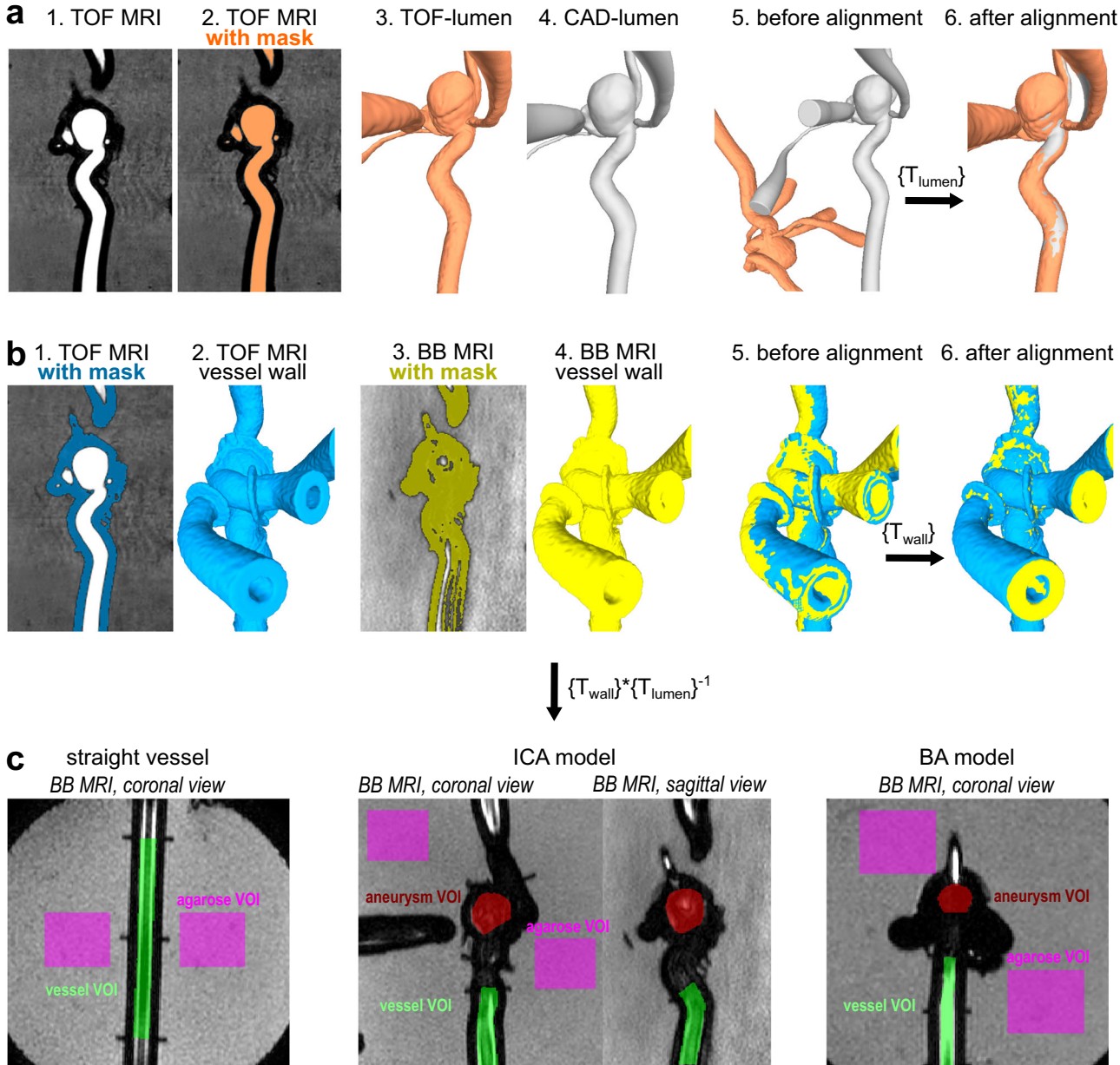

**Fig. 3 | Workflow for consistent volumes of interest (VOIs) placement on black-blood MR images. a** The vascular lumen was segmented from the time-of-flight (TOF) MR images (orange) and aligned with CAD lumens (gray), resulting in the transformation mask from TOF to CAD coordinates – $\{T_{lumen}\}$. **b** Vessel wall was segmented from TOF (blue) and BB MR images (yellow) and aligned together, resulting in the transformation matrix from TOF to BB coordinates – $\{T_{wall}\}$. **c** CAD VOIs were translated into BB MRI coordinates by applying the transformation matrix – $\{T_{wall}\} * \{T_{lumen}\}^{-1}$, vessel VOI (green), aneurysm VOI (red). Agarose VOI was placed outside of the vascular model to normalize the BB signal (magenta).

The TOF wall was aligned with the BB wall using ICP[32] with an average alignment error of 0.09 to 0.31 mm, resulting in the transformation matrix from the TOF to BB coordinates – $\{T_{wall}\}$ (Meshlab).

3. CAD VOIs of aneurysm and vessel models were translated into BB MRI coordinates by applying the transformation matrix – $\{T_{wall}\} * \{T_{lumen}\}^{-1}$ (Matlab).

4. Translated VOIs were transformed into multiple 2D ROIs across BB MR images (Fig. 3c, MevisLab). Firstly, VOIs were clipped with a given plane (individual BB MR image), resulting in a 2D contour located in the intersection of the surface of the VOI and the clipping plane. Secondly, each 2D contour was voxelized, resulting in a 2D mask comprising the voxels within the 2D contour. The mask included voxels if the 2D contour covered more than 50% of the voxel volume.

Finally, the BB signal in the parent vessel and the aneurysm sac was calculated within the resulting masks.

The BB signal in a static tissue (agarose gel) was determined within two rectangular cuboids with dimensions of $19.5 \times 16.3 \times 23.4$ mm³ placed on the right and left sides outside of the flow volume (Matlab). To enable the comparison between BB signals derived from different BB sequences without dependence on SNR, the BB signal in VOIs calculated in the aneurysm and vessel regions was normalized by the mean values of the BB signal in agarose gel, and the normalized BB signal was reported in arbitrary units (a.u.). Normalized BB signal distributions are presented with violin plots normalized by the width (Phyton 8.2.0, Jupyter Notebook 6.4.8, library seaborn). The postprocessing workflow described above, including scripts, the manual, and example files, is publicly available on Zenodo[33].

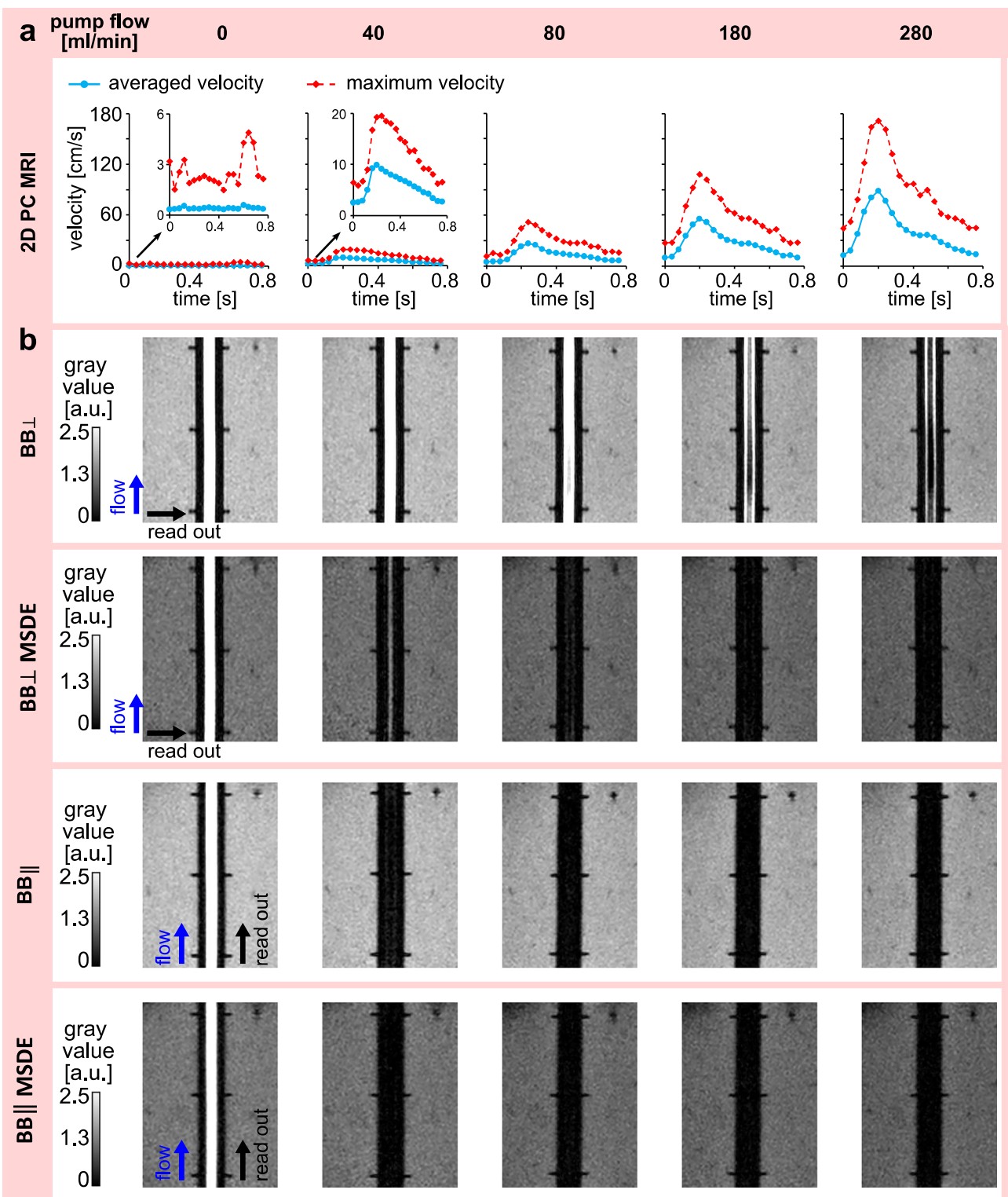

**Fig. 4 | Dependence of the black-blood (BB) signal on the flow rate, readout gradient orientation, and the presence of the motion-sensitized driven-equilibrium (MSDE) module in the straight vessel model. a** Time-dependent averaged and maximum velocity profiles measured with 2D phase-contrast (PC) MRI at the middle of the straight vessels. **b** BB MR images displayed with a gray-value color map at the cross-section along the vessel for different flow rates.

## Image-based flow simulations

To verify 4D flow MRI results, numerical flow simulations based on CFD were performed on BA models without and with devices (IFD1−10). The commercial finite volume solver STARCCM+ was used (StarCCM +2021.3 v16.6, Siemens, Erlangen, Germany) and laminar flow conditions were applied. Fluid properties were set mimicking the experiments. Original

CAD models of the BA phantom models were set as the basis for the IA geometries in CFD. The vessel walls were assumed to be rigid, corresponding to the in vitro experiment. Regarding the deformation of the IFDs, first, the deployed geometry of IFDs in the BA models was acquired and segmented from μ-computed tomography (CT) data (a vivaCT 80; Scanco Medical AG, Brüttisellen, Switzerland; 45 keV, 80 mm field of view,

**Fig. 5 | Relationship between velocity and black-blood (BB) MRI signal.** The scatter plot shows median values of spatially averaged velocity measured with 2D phase-contrast MRI plotted against the median values of the BB signal measured with $BB_\perp$, $BB_\perp$ MSDE, $BB_\parallel$, and $BB_\parallel$MSDE.

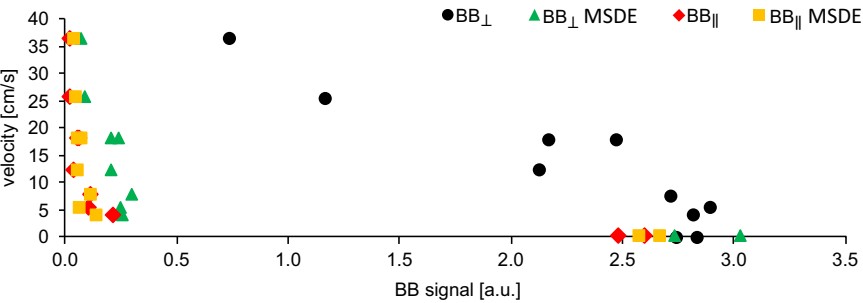

reconstructed to 26 μm isotropic voxel size). Then, CAD models of the IFDs were registered and deformed to match the shape and location of the segmentations from μ-CT devices[34]. Flow and pressure curves measured at the model inlet and outlets, respectively, before MRI were used as boundary conditions (ME8PXL-M12, Transonic System Inc, United States; press-N-000; PendoTech, NJ). The spatial resolution was set to a mesh base size of 0.1 mm, with refinement to 0.02 mm at the IFD struts. The temporal resolution was 0.1 ms, and three cardiac cycles were simulated, with only the last cycle considered in the evaluation. Postprocessing was performed using the software ANSYS EnSight (v2021.R2; ANSYS, USA) and Matlab. The median temporal velocities were calculated within the IA sac and parent vessel (like VOIs in MRI results, as presented in Fig. 2b, with minor manual deviations). Here, the IA sac was separated from the parent vessel using a planar neck plane. The location of the planes within the IA flow models for evaluation was set based on the experiments.

### Statistics and reproducibility
Normalized BB signal distributions of empty IA models were compared with those of the IA models with implanted FMDs using the Kruskal-Wallis test. In addition, if more than two groups were tested simultaneously, the post hoc Dunn pairwise comparisons with Benjamini-Hochberg p-value adjustment were used. The treatment effect (Δ) of FMDs on BB signal and velocity was calculated as follows, $\Delta BB = \frac{BB_{wo\,FMD} - BB_{with\,FMD}}{BB_{wo\,FMD}}$ and $\Delta velocity = \frac{velocity_{wo\,FMD} - velocity_{with\,FMD}}{velocity_{wo\,FMD}}$, respectively. These treatment effects (ΔBB and Δvelocity) were then tested against a zero-effect hypothesis using a two-sided Wilcoxon signed-rank test (sample size = 16). The relationship between velocity and BB signal was assessed using the Spearman correlation coefficient (rho). The statistical significance for each test was defined as $p < 0.05$. The statistical analyses were performed in Matlab and Python 8.2.0 (Jupyter Notebook 6.4.8, library scipy.stats).

The untreated ICA model and the ICA model with FD1 were imaged with 4D flow MRI and $BB_\perp$ MRI twice, approximately one year apart. Additionally, multiple replicas of identical devices were implanted in the BA models to assess the reproducibility of the treatment effect.

### Results
#### BB signal dependence on flow rate and sequence parameters in straight vessels
To assess the dependence of the BB signal on supplied flow rates, straight vessels were perfused to a mean flow rate in the range of 0 to 5.4 ml/s. All flow conditions resulted in a similar time-dependent flow waveform with a peak at approximately 0.2 s, reflecting the consistent cardiac cycle waveform generated by the pulsatile pump (Fig. 4a). The waveform shape remained largely unchanged, while its amplitude was adjusted by modifying the mean flow rate. Representative inlet flow curves measured with flow sensor for different flow rates are provided Supplementary Fig. 3. The BB signal decreased with increasing flow rates for both vessel IDs across all sequences (Fig. 4b), resulting in a negative correlation coefficient between the BB signal and velocity, namely $BB_\perp$ (rho = −0.87, $p$ = 1.08E-03), $BB_\perp$ MSDE (rho =

−0.95, $p$ = 2.93E-05), $BB_\parallel$ (rho = −0.92, $p$ = 1.32E-04), and $BB_\parallel$MSDE(rho = −0.94, $p$ = 6.72E-05).

A gradual linear decrease in the BB signal was observed for the $BB_\perp$ (velocity = $-14.7 \times BB_\perp + 46.1$, $R^2 = 0.87$, Fig. 5). A sharp decline in the BB signal occurred at a flow rate of 0.7 ml/s and a median velocity of 5.5 cm/s for $BB_\perp$ MSDE, $BB_\parallel$, and $BB_\parallel$MSDE. Regardless of MSDE presence, $BB_\parallel$ reduced the BB signal more effectively than $BB_\perp$. For instance, at a flow rate of 3 ml/s, $BB_\parallel$-signal was 0.03 (a.u.), while $BB_\perp$-signal was 1.16 a.u. The individual results are listed in the Supplementary Table 1.

#### BB signal dependence on sequence parameters in aneurysm models
4D flow MRI and $BB_\perp$, $BB_\perp$ MSDE, $BB_\parallel$, and $BB_\parallel$MSDE were performed in the ICA model, both without and with FD1 (Fig. 6). Imaging was performed twice: with the pump active, supplying ICA models at a mean flow rate of 4.5 ml/s, and with the pump off, resulting in a static fluid inside the ICA model.

When flow was circulating, FD1 reduced the velocity within the aneurysm sac (0.7 vs. 11.5 cm/s), while the velocity in the parent vessel was similar (15.3 vs. 14.8 cm/s) to the untreated empty ICA model (Fig. 6a, b). The BB signal in the treated aneurysm increased for all BB sequences ($p$ = 3.56E−128), with no apparent change, although statistically different ($p < 8.39E-03$) in the parent vessel across all tested BB sequences compared to the empty model (Fig. 6c). $BB_\perp$ signal in the aneurysm sac without FD1 was higher than $BB_\perp$ MSDE, $BB_\parallel$, and $BB_\parallel$MSDE signals ($BB_\perp$ signal = 0.15, $BB_\perp$ MSDE signal = 0.07, $BB_\parallel$ signal = 0.09, and $BB_\parallel$MSDE signal = 0.06 a.u.). Quantitative results are listed in Supplementary Table 2.

The BB signal in the aneurysm sac with FD1 was not apparently affected by FE orientation or MSDE ($BB_\perp$ signal = 3.41, $BB_\perp$ MSDE signal = 3.88, $BB_\parallel$ signal = 3.35, and $BB_\parallel$MSDE signal = 3.74 a.u.). The $BB_\perp$ signal was higher than the $BB_\perp$ MSDE, $BB_\parallel$, and $BB_\parallel$MSDE signals in the parent vessel regardless of FD1. Increased BB signal in the aneurysm sac was colocalized with the reduced flow caused by FD1 (Fig. 6). Without flow, no velocity was measured, and the BB signal did not change, apparently, regardless of the presence of the FD1 (Supplementary Table 3 and Supplementary Fig. 5). FD1 caused a minor metal artifact near the surface of the parent vessel, resulting in MRI signal voids pronounced on 4D flow MR images for both flow conditions (Fig. 6a, yellow arrowhead).

#### Impact of flow-modulation devices on BB signal in aneurysms
**Flow-diverter stents.** In a separate experiment from the one described above, 4D flow MRI and $BB_\perp$ and $BB_\perp$MSDE were performed in the ICA model with and without FD1-5 perfused at a mean flow rate of 4.5 ml/s (Fig. 7).

FD1 induced minor metal artifacts near the surface of the parent vessel, while FD2 and FD4-5 caused strong metal artifacts within the vessel lumen, resulting in MRI signal voids (Fig. 7a). During the MRI experiment, the distal end of the FD3 dislocated into the aneurysm sac from its initial position in the parent vessel, causing metal artifacts in the aneurysm sac (Fig. 7a, yellow and blue arrowheads).

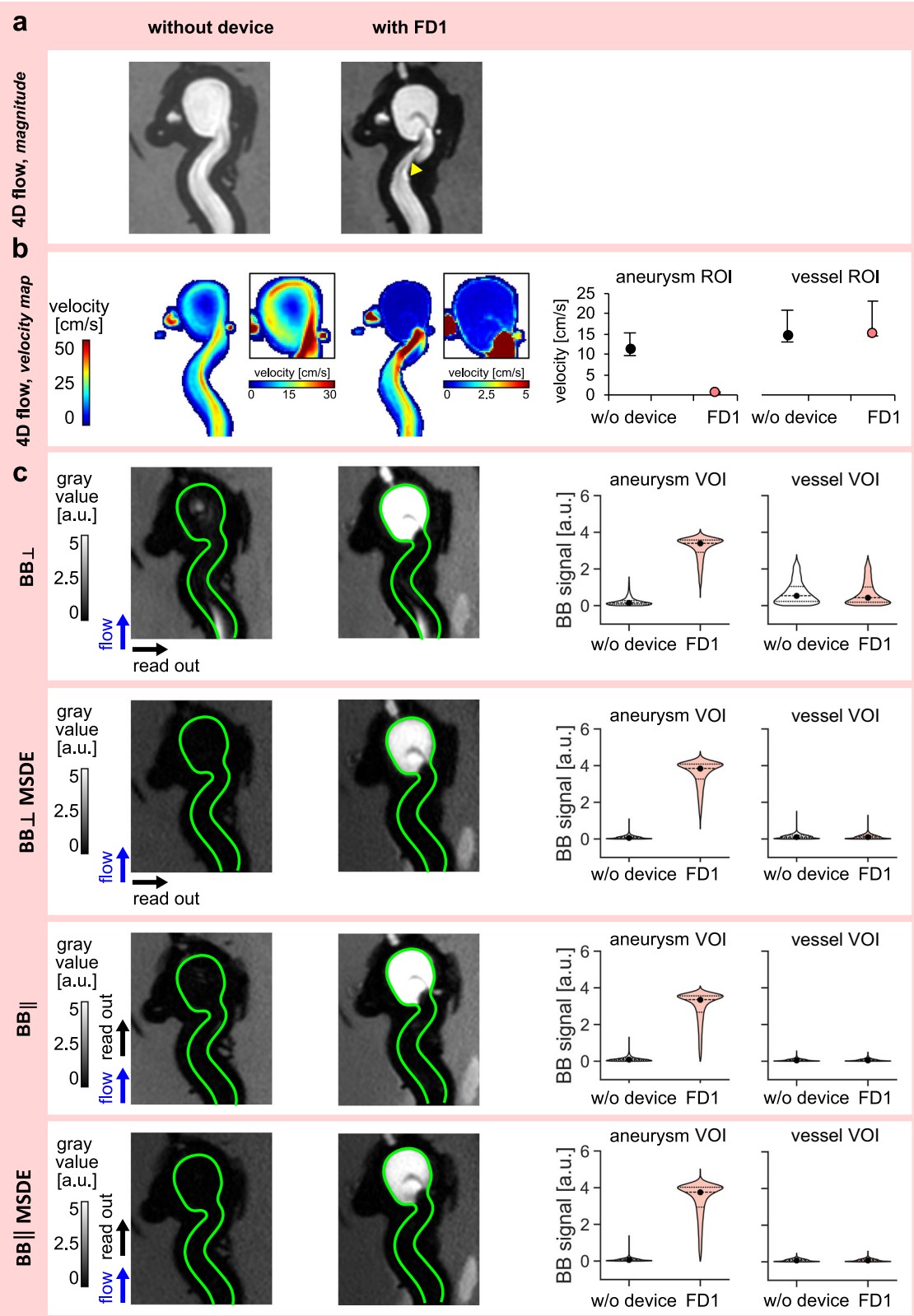

**Fig. 6 | Effect of a flow-diverter stent (FD1) on velocity and black-blood (BB) signal in an internal carotid artery (ICA) model.** Representative images include 4D flow magnitude MRI (**a**), velocity maps (**b** left), and BB MRI (**c** left). Green lines outline the surface of the aneurysm model's inner wall. The scatter plot illustrates the velocity change induced by FD1 in the aneurysm sac and vessel (**b** right), where each point represents the median velocity across $n = 20$ time points, and the error bars indicate the first and third quartiles of these time-dependent measurements. The violin plots demonstrate the change in the BB signal caused by FD1 in the aneurysm sac and vessel (**c** right, points represent a median velocity value, while dotted lines represent first and third quantiles). FD1 caused some metal artifacts at the parent vessel (**a** yellow arrowhead), reduced the aneurysmal velocity (**b**), and increased the aneurysmal BB signal (**c**).

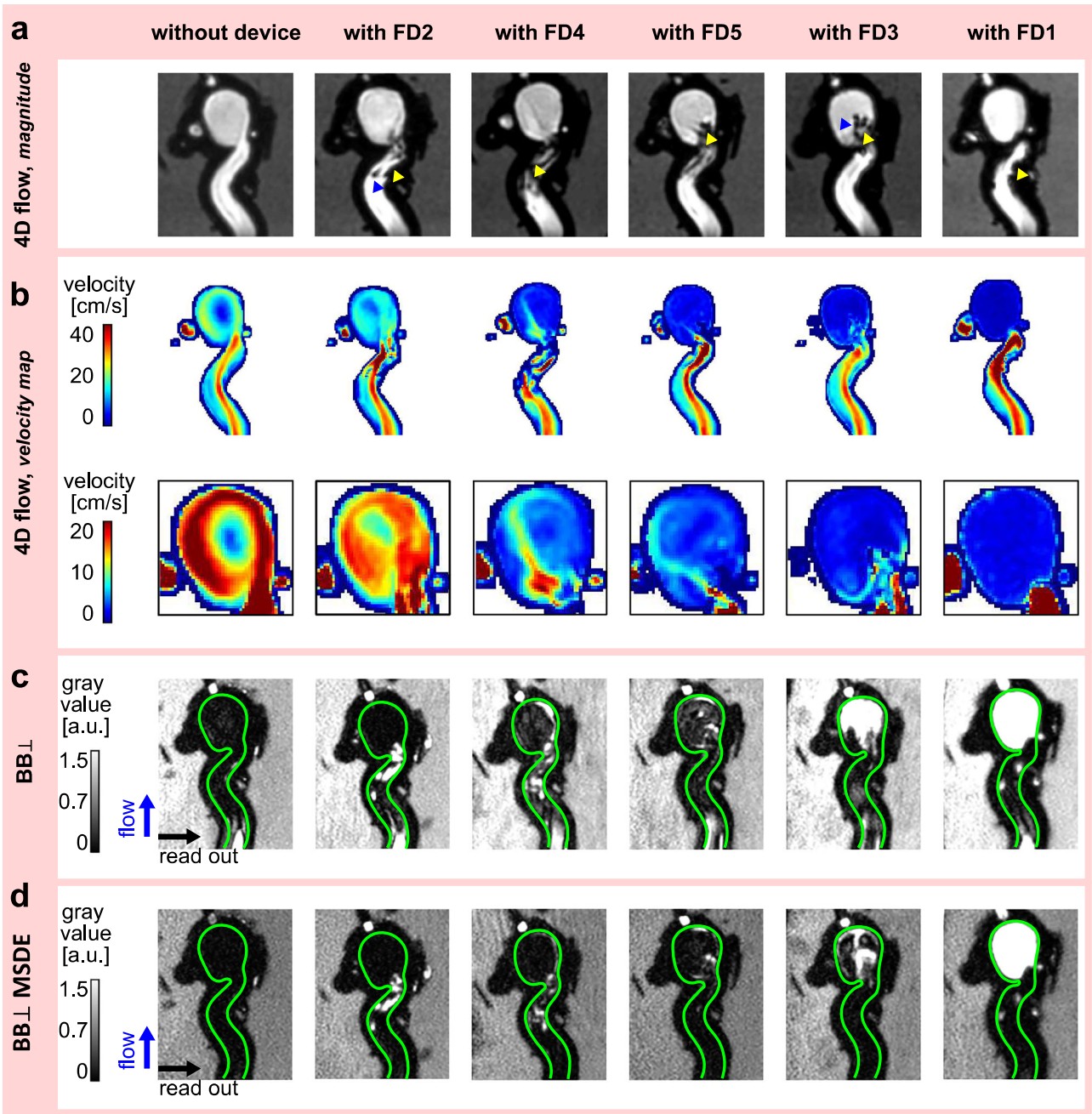

**Fig. 7 | Effect of flow-diverter stents (FD1-5) on velocity and black-blood (BB) signal in the internal carotid artery (ICA) model.** Representative images include 4D flow magnitude MRI (**a**), velocity maps (**b**), and BB MRI (**c**, **d**). The green lines outline the surface of the model's inner wall. FDs caused metal artifacts at the parent vessel and aneurysm sac (**a** yellow and blue arrowheads), reduced the aneurysmal velocity (**b**), and increased the aneurysmal BB signal (**c**, **d**). Note that the images in (**a**) are ordered not by the number of the device, but rather by the extent of the effect on flow and BB signal.

FD1 and FD3-5 substantially reduced the velocity within the aneurysm sac, while the velocity in the parent vessel remained similar across the models (Fig. 7b). FD1 was the most effective (with [w.] = 0.6 vs. without [w/ o.] = 12.2 cm/s), whereas FD2 showed minimal flow reduction (w. = 11.2 vs. w/o. = 12.2 cm/s), likely due to its large cell size and low metal coverage. In line with these observations, the BB MRI signal in the aneurysm sac increased in the presence of FD1 and FD3–FD5 for BB$_\perp$, while there was no change in the presence of FD2. BB$_\perp$ MSDE signal increased in the presence of FD1, FD3 and FD5 (Fig. 7c, d, Fig. 8a, b).

The BB signal was inversely related to the intra-aneurysmal velocity, indicating that lower velocity corresponded to a higher BB signal for both BB$_\perp$ (rho = −0.94, $p$ = 1.6E-02) and BB$_\perp$ MSDE (rho = −0.94, $p$ = 1.6E-02) sequences (Fig. 8c). Flow suppression in the parent vessel was weaker for BB$_\perp$ than for BB$_\perp$ MSDE sequences (Supplementary Fig. 6a). The BB$_\perp$ signal showed moderate variation in the parent vessel, but no consistent relationship with velocity (rho = −0.66, $p$ = 1.75E-01). In contrast, the BB$_\perp$ MSDE signal did not show significant differences, which was consistent with the lack of change in velocity (rho = −0.94, $p$ = 1.67E-02), as illustrated in Supplementary Fig. 6. The quantitative results are listed in the Supplementary Table 4 and Supplementary Table 5. Overall, metal artifacts were less pronounced on BB MRI than on 4D flow MRI.

**Intrasaccular devices.** 4D flow MRI and BB$_\perp$ were used to examine BA models without and with IFD1-10 supplied with a mean flow rate of 2.4 ml/s. Metal artifacts from IFD1-10 greatly affected 4D flow MRI;

**Fig. 8 | Effect of flow-diverter stents (FD1-5) on black-blood (BB) signal and its relationship with velocity.** Violin plots illustrate the change in the BB signal induced by FDs in the aneurysm sac, depicted by $BB_\perp$ (**a**) and $BB_\perp$ MSDE (**b**), points represent a median velocity value, while dotted lines represent first and third quantiles. The scatter plot illustrates the relationship between the median value of spatially averaged velocity and the median value of the BB signal (**c**).

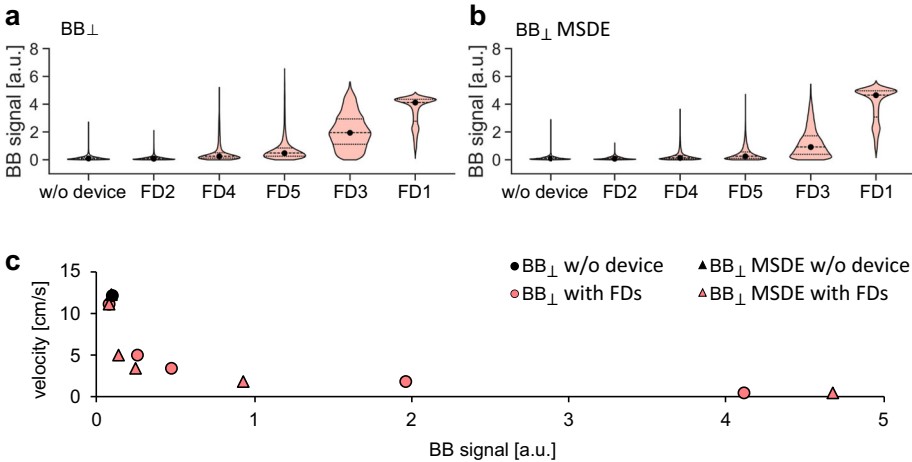

therefore, numerical flow simulations were performed to verify the velocity measured by 4D flow MRI and to assess flow information in the regions impaired by artifacts (Fig. 8).

Based on 4D flow MRI experiments, all intrasaccular devices (IFD1-10) qualitatively reduced the velocity within the aneurysm sac (Fig. 9), while the velocity in the parent vessel was similar across the models. Velocity reduction for IFD1-2 could not be assessed by 4D flow MRI due to MRI signal voids caused by metal artifacts from the implants (visualized in Fig. 9 a, yellow and black arrowheads). IFD3-10 were placed in larger aneurysm sacs than IFD1-2. Consequently, some artifact-free areas were present within the aneurysm sacs where velocity was calculated. IFD6-9 achieved the most effective velocity reduction (w. = 1.9, 2.0, 1.9, 1.9 vs. w/o. = 16.4 cm/s), followed by IFD10 (3.7 vs. 10.2 cm/s), while IFD3-5 showed the least reduction (w. = 2.3, 1.8, 2.0 vs. w/o. = 2.7 cm/s) compared to BA models without a device. Velocity in the parent vessel was similar among the BA models, regardless of the presence of the devices and size of the aneurysm, ranging from 28.4 to 34.2 cm/s.

Results calculated with CFD confirmed the velocity reduction observed by 4D flow MRI (Fig. 9). While flow through the aneurysm can be observed when no device is placed, there is little to no flow in the aneurysm sac with a device. Velocity temporal median values were reduced from 14.3 to 3.2 cm/s for the largest BA model 4 and from 11.8 to 0.6 cm/s for the smallest BA model 1. The effect of IFDs on other hemodynamic parameters (e.g., wall shear stress, oscillatory shear index) is discussed in detail in our previous publication[34].

The $BB_\perp$ signal in the aneurysm sac increased in the BA models with IFD1-10 compared to empty BA models (Fig. 9). The gain in $BB_\perp$ signal in the aneurysm sac was consistent for IFDs in the same BA model, except for IFD4 (Fig. 10a). Specifically, $BB_\perp$ signals were 1.24 and 1.32 a.u. for IFD1-2, respectively; 2.81, 0.77, and 2.01 a.u. for IFD3-5; 3.61, 4.13, 4.05, and 5.62 a.u. for IFD6-9; and 1.38 a.u. for IFD10. The $BB_\perp$ signal showed moderate differences in the parent vessel without any apparent trend related to the device's presence or size of the aneurysm neck (Supplementary Fig. 7). Once again, the BB signal was inversely associated with intra-aneurysmal velocity measured with 4D flow MRI (rho = −0.73, p = 1.00E-02), with lower velocity corresponding to a higher BB signal (Fig. 10). A negative trend was observed between the BB signal and velocity calculated with CFD, but it was not statistically significant (rho = −0.5, p = 9.88E-02). No apparent relationship between BB signal and velocity in parental vessel measured by 4D flow MRI (rho = 0.06, p = 8.28E-01) and calculated by CFD (rho = 0.3, p = 2.91E-01) was found. Metal artifacts were less pronounced on SE BB MRI than on 4D flow MRI, allowing BB signal assessment even for the smallest aneurysm, BA model 1, with IFD1 and IFD2.

Quantitative results for velocity measured with 4D flow MRI, calculated with CFD and BB signal, are presented in Supplementary Tables 6 and 7.

**Statistical analysis of device effects.** 4D flow and $BB_\perp$ MRI were performed for five FDs and ten IFDs. Differences in intra-aneurysmal velocity and BB signal between treated and untreated cases were statistically evaluated. Since only one untreated control model was available per aneurysm geometry, a normalized treatment effect was calculated for each FMD as the relative change in velocity (Δvelocity) and BB signal (ΔBB) compared to the control, and these values were compared against a null (zero-effect) distribution. The median intra-aneurysmal velocity was significantly lower with FMDs (Δvelocity = −0.78, p = 1.22E-04), while velocity in the parent vessel remained unchanged (Δvelocity = −0.03 %, p = 1.09E-01). The BB signal within the aneurysm sac increased after treatment (ΔBB = 16.92, p = 2.44E-04), whereas no significant change was observed in the parent vessel (ΔBB = −0.18, p = 6.69E-01). Individual treatment effects for each device are provided in Supplementary Table 8.

Furthermore, Δvelocity was inversely correlated with ΔBB in the aneurysm (rho = −0.82, p = 6.24E-04), while no correlation was found in the parent vessel (rho = −0.07, p = 7.88E-01).

## Discussion

This study aimed to explore whether the signal behavior of SE BB MRI in regions of slow flow—typically considered as an undesired property—can instead serve as a diagnostic feature. Specifically, we assessed whether the BB signal can provide complementary information to 4D flow MRI in the presence of severe metal-induced artifacts after aneurysm treatment.

To address this, we conducted in vitro experiments using simplified straight vessel phantoms and patient-derived aneurysm models originating from the ICA and BA. FMDs of various sizes, including five FDs and ten IFDs, were placed in the aneurysm models to mimic aneurysm treatment. We evaluated how the SE BB MRI signal intensity changes with (1) varying flow rates, (2) MRI acquisition parameters and (3) presence of FMDs. Our results demonstrate a consistent association between the SE BB signal and intra-aneurysmal flow conditions, suggesting the potential for using SE BB imaging as an adjunct to flow quantification, in conjunction with 4D flow MRI. In the discussion, we summarize the key results of this study and compare them to the existing literature, while also discussing the validity of the in vitro approach and the overall limitations of our work.

In this study, we found an increased intra-aneurysmal BB signal in IA models with FMDs, likely due to reduced aneurysmal flow and poor slow-flow suppression of BB MRI. Previously, we reported that slow flow in ICA and BA aneurysm models might lead to increased BB signals without FMDs,

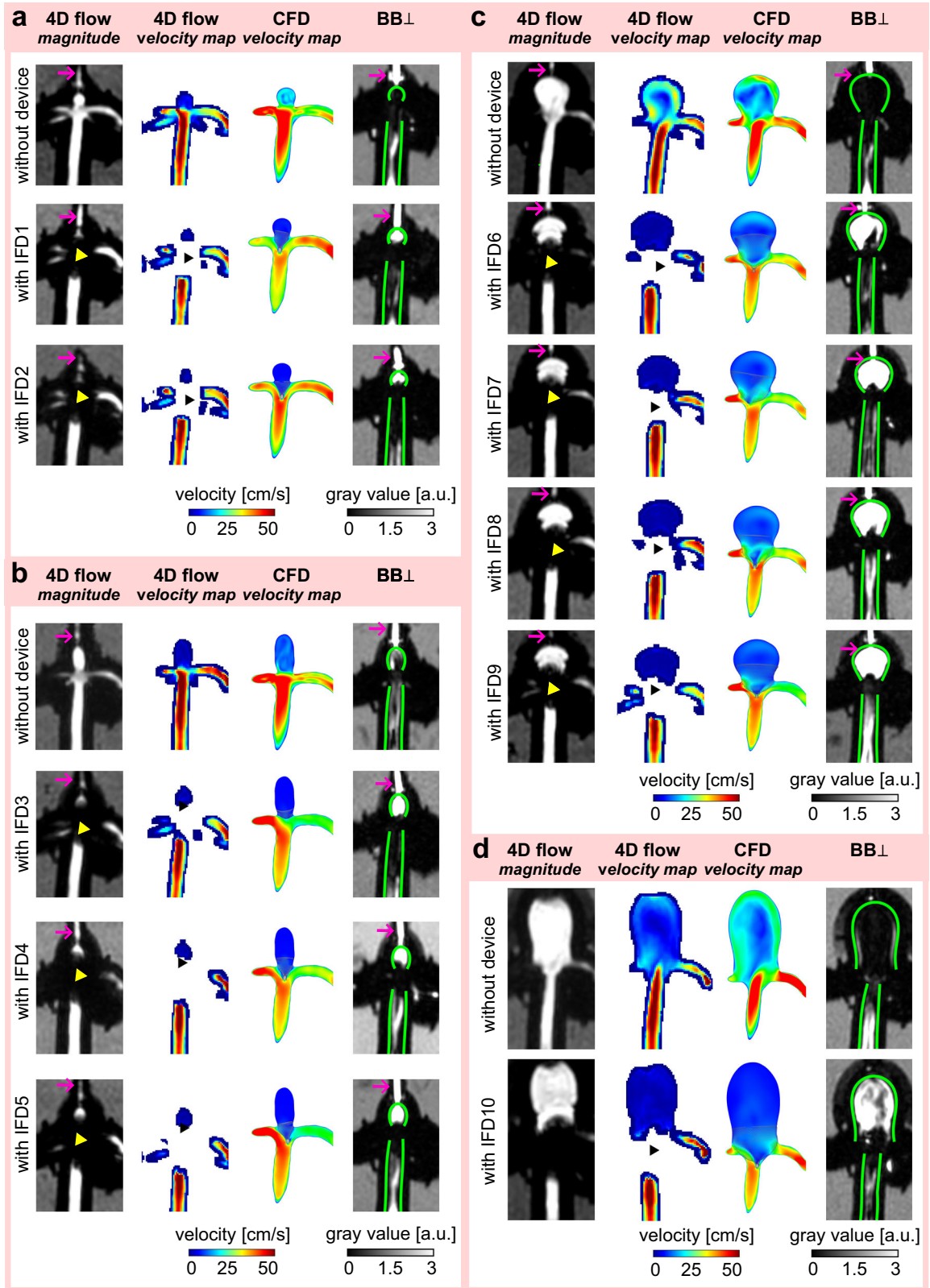

**Fig. 9 | Qualitative effect of intrasaccular flow-disrupting devices (IFDs) on velocity and black-blood (BB) signal in basilar artery (BA) aneurysm models.** Velocity was measured with 4D flow MRI and calculated with computational fluid dynamics (CFD) simulations, whereas the BB signal was measured with $BB_\perp$ MRI. Color-coded velocity maps and grayscale magnitude images from 4D flow and BB MRI are shown separately for each BA model: BA model 1 (**a**), BA model 2 (**b**), BA model 3 (**c**), and BA model 4 (**d**). The green lines outline the surface of the aneurysm model's inner wall. IFDs caused severe metal artifacts at the parent vessel and aneurysm sac (**a–d**, yellow and black arrowheads), reduced the aneurysmal velocity, and increased the aneurysmal BB signal. The pink arrow indicates flow stagnation within the pressure sensor attached to the aneurysm sac.

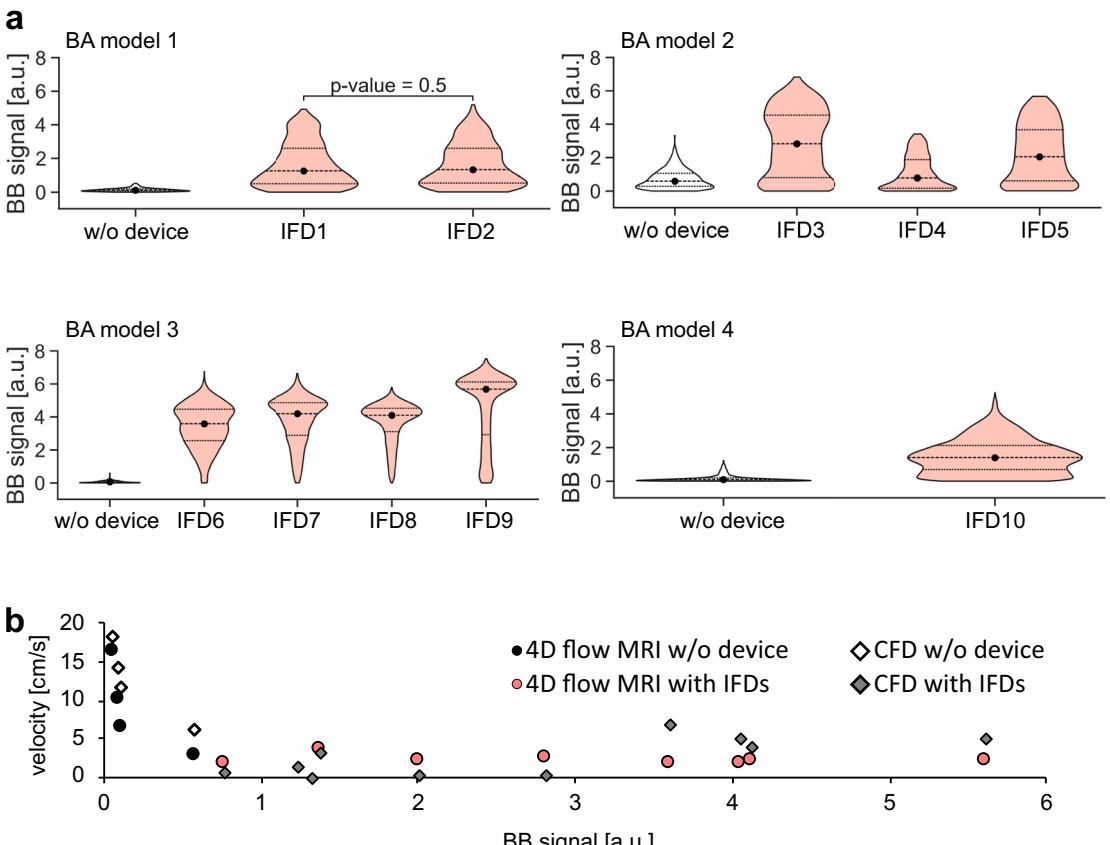

**Fig. 10 | Quantitative effect of intrasaccular flow-disrupting devices (IFD1-10) on velocity and black-blood (BB) signal in the aneurysm of the basilar artery (BA) models.** Violin plots illustrate the BB signal in the aneurysm sac, depicted by $BB_\perp$ (**a**) with IFD1-10 and without IFD. A scatter plot illustrates the relationship between the median value of spatially averaged velocity measured with 4D flow MRI and calculated with CFD and the median value of the BB signal in the aneurysm sac (**b**). Only IFD1 and IFD2 resulted in a statistically similar distribution of BB signal within the aneurysm sac ($p = 0.5$), while the others were different ($p < 0.01$). Statistical analysis was performed using the Kruskal–Wallis test, followed by post hoc Dunn pairwise comparisons with Benjamini–Hochberg p-value adjustment.

offering insights into the hemodynamic conditions associated with aneurysm progression[18]. In the current study, we observed an increased BB signal in the aneurysm sac with FMDs. While "failing" BB MRI is usually undesirable for diagnosing vessel wall diseases, assessing immediate flow reduction post-endovascular treatment may be valuable here. Moreover, BB MRI is less susceptible to metal artifacts and has shorter acquisition times than 4D flow MRI (here 5.1 min vs. 26.1 min, Table 2)[17].

SE BB MRI has shown potential for evaluating long-term IA healing by imaging the lumen of coiled aneurysms[20], and it is considered a viable alternative to TOF MRI and DSA for detecting aneurysm remnants[35] and assessing parent vessel patency[35,36]. In this study, SE BB MRI exhibited fewer metal artifacts than 4D flow and TOF MRI, especially when imaging IFDs (here, using the Contour Neurovascular System). IFDs induce enormous metal artifacts on MRI, making such images unsuitable for follow-up of patients treated with Contour[37,38]. These advantages position BB MRI as a promising technique for evaluating IAs, particularly in patients treated with Contour, who are currently being followed up with DSA and CTA.

As shown in this study, BB MRI allowed visualization of nearly the entire aneurysm sac, even in small aneurysms (3.5–6.9 mm in height). This is clinically relevant, as small aneurysms represent 20–60% of all detected aneurysms[39] and still carry a risk of rupture[40]. Beyond anatomical visualization, our results indicate that BB MRI may help highlight regions of slow intra-aneurysmal flow following treatment, potentially supporting the assessment of treatment efficacy. Flow measurements were obtained using 4D flow MRI, and, where metal artifacts impaired visibility—particularly in BA models 1–4 treated with intrasaccular devices—were supplemented by numerical simulations. This was especially necessary in BA models 1 and 2, where the artifact size on 4D flow MRI exceeded the size of the aneurysm dome. While SE BB MRI does not directly quantify velocity, it may serve as a complementary tool when severe metal artifacts compromise 4D flow MRI. In addition, the increased BB signal observed in BA models after placement of IFDs corresponded to the increased contrast washout times measured in our prior study[14], reinforcing the potential of the BB MRI signal to reflect the intra-aneurysmal flow reduction qualitatively. Moreover, since SE BB MRI is already routinely performed for vessel wall assessment in patients with IAs before endovascular repair, leveraging this existing data to gain flow-related insights could provide additional benefit to patient care.

Unlike velocity measurements with 4D flow MRI, BB MRI signal intensity is influenced by a range of acquisition parameters that impact the efficiency of flow suppression. Prior studies have shown that changes in voxel size and the inclusion of additional suppression modules, such as MSDE, can alter the BB signal in untreated aneurysms[18,19]. In our study, we further demonstrated that readout gradient orientation and MSDE application affect flow suppression efficiency, particularly in straight vessel models and parent arteries. For example, aligning readout gradients with flow direction improves suppression due to enhanced intra-voxel dephasing[41]. This effect was less pronounced in aneurysm sacs, likely due to the complex, multi-directional vortex flow patterns.

Despite these dependencies, a consistent increase in BB MRI signal within the aneurysm sac after treatment was observed across all tested variations: (1) BB parameters (MSDE on/off, readout direction); (2) Aneurysm geometry (ICA vs. BA); (3) Device type (flow-diverters and

intrasaccular devices); (4) Repeated independent implantations of the same device (e.g., four devices were implanted in BA model 3); (5) Different fluid viscosities (glycerol/water in experiments with ICA models vs. water in experiments with BA models). Although the extent of the BB signal increase varied, the effect was robustly present. This suggests that BB MRI may provide a reproducible, semi-quantitative indication of post-treatment flow stasis if acquisition parameters are kept constant between pre- and post-implantation imaging. Future work should focus on standardizing acquisition protocols and exploring quantitative thresholds that may allow BB MRI to transition toward a more formal diagnostic role.

To isolate the effects of velocity reduction on the BB signal before coagulation, we conducted this study in vitro using patient-derived aneurysm models. The in vitro scenario mirrors the in vivo MR imaging of patients with IAs immediately after treatment, before potential aneurysm neck endothelization and thrombus formation, which takes several weeks to up to 6 months[42,43].

Previous studies using DSA, 4D flow MRI, and CFD have shown that IVR can predict long-term aneurysm occlusion[7,10,13,16,17]. Moreover, it was previously shown that velocity reduction measured with 4D flow MRI in the aneurysm models is similar to that calculated with CFD when the same vascular geometry is used[34,44]. Our findings suggest that an increased BB signal post-treatment could similarly predict treatment outcomes (in vivo and with shorter acquisition times than 4D flow MRI).

The in vitro approach enabled systematic evaluation of different FMDs and BB MRI sequences in the same aneurysm geometry, which is not feasible in clinical settings. The decreased velocity observed with all FMDs has been associated with increased BB signal within the aneurysm sac, suggesting that the BB signal may serve as a surrogate marker for flow reduction. However, in vivo studies are needed to demonstrate whether BB could provide a measure of aneurysm occlusion outcomes, like DSA's washout time[13] or 4D flow MRI's and CFD's velocity reduction[10,17].

The current study has a few limitations. First, this study presents in vitro findings from a single-center conducted on a limited number of patient-derived IA models and straight vessels, which limits the generalizability of the results. However, the selected aneurysms represented different types (lateral versus terminal) and sizes, ranging from 3.5 to 16.4 mm in maximum dimensions. Moreover, all models possessed rigid walls, which might result in hemodynamic conditions different from those observed in vivo, where vessel walls are elastic. Still, this assumption has a smaller effect on the assessment of treatment efficiency. While we included various treatment types, other devices, such as coiling and woven endo bridges, were excluded due to known severe metal artifacts on MRI. A blood-mimicking fluid (water or water-glycerine mixture) was used, therefore the BB signal intensities would likely differ if real blood were present, preventing the direct translation of in vitro results to in vivo scenarios. Nevertheless, assessing relative changes in signal intensities remained feasible. The in vitro approach also does not account for post-treatment coagulation, though this process typically takes weeks to develop. Our model effectively reflects immediate post-treatment velocity reduction. Not all four BB MRI sequence variations ($BB_\perp$, $BB_\perp MSDE$, $BB_\parallel$, and $BB_\parallel MSDE$) were used in all 20 flow experiments with IA models, limiting generalizability. Velocity analysis on 4D flow MRI was limited to artifact-free areas; however, numerical simulations mostly confirmed the results. Velocity phase offsets were corrected using a linear fit; however, higher-order polynomial corrections would likely improve accuracy and should be considered in future work. Due to the limited number of untreated models ($n = 5$), it was not possible to assess the variability of baseline BB signal across different IA anatomies. This limitation should be considered when interpreting the treatment effect results. Finally, the location of ROI for flow evaluation and BB signal variation might differ slightly between cases with and without FMDs. BB MRI signal does not provide velocity information, unlike 4D flow MRI, and can only be used as a surrogate marker at this stage. Future work will focus on testing the hypothesis in vivo, standardizing signal thresholds, calibrating the BB signal against known flow regimes, and validating these findings across multiple centers and imaging platforms.

In conclusion, increased intra-aneurysmal SE BB MRI signal was observed in IA flow models following endovascular flow modulation and was associated with reduced intra-aneurysmal flow. SE BB MRI exhibited fewer metal artifacts than 4D flow MRI. Therefore, SE BB MRI should be investigated as a clinically applicable non-invasive surrogate for assessing IA flow reduction and as a potential diagnostic modality for IA treatment outcomes.

## Data availability
The MRI datasets obtained from aneurysm models are available in the Zenodo research repository[45,46]. Digital aneurysm models of basilar tip aneurysm (BA models) were previously used elsewhere[14,34] and are publicly available at Zenodo[47]. The digital aneurysm model of the internal carotid artery (ICA model) was previously used elsewhere[27,48] and available at Zenodo[49]. CFD data are available from the corresponding author upon reasonable request. The processed data underlying the plots in the figures accompanying this manuscript are available as supplementary data. Namely, the source data for Fig. 4a is in Supplementary Data 1, the source data for Fig. 5 is in Supplementary Data 2, the source data for Fig. 6b is in Supplementary Data 3, the source data for Fig. 6c is in Supplementary Data 4, the source data for Fig. 8a,b and Supplementary Fig. 6a,b is in Supplementary Data 5, the source data for Fig. 8c and Supplementary Fig. 6c is in Supplementary Data 6, the source data for Fig. 10a and Supplementary Fig. 7a is in Supplementary Data 7, the source data for Fig. 10b and Supplementary Fig. 7b is in the Supplementary Data 8, the source data for Supplementary Fig. 3 is in the Supplementary Data 9, the source data for Supplementary Fig. 4 is in the Supplementary Data 10, the source data for Supplementary Fig. 5c is in the Supplementary Data 11.

## Code availability
A workflow for MRI data post-processing is publicly available at Zenodo[33].

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

## Acknowledgements
We acknowledge Philips Healthcare for support in MRI experiments. We thank Cerus Endovascular and Stryker Corporation for providing the Contour Neurovascular System, and Acandis for providing Derivo for in vitro testing. We acknowledge the support of Kiel University and the Medical Faculty for the 3D lab at SBMI. O.J. discloses support for the research of this work from German Research Foundation (GRK2154: P2b); P.B. discloses support for the research of this work from German Research Foundation (SPP2311: project no. 548907942), the German Federal Ministry of Research, Technology and Space within the Research Campus STIMULATE (grant no. 13GW0835A) and the European Regional Development Fund under the operation number 'ZS/2023/12/182010' as part of the initiative "Sachsen-Anhalt WISSENSCHAFT Schwerpunkte"; M.S.P. discloses support for the research of this work from the Bruhn Foundation (2022).

## Author contributions
Conceptualization and design of the project: M.S.P., N.L., O.J., and J.B.H. Design and production of 3D-printed aneurysm models: M.S.P. and L.B. Implantation of the FMDs and choice of the aneurysm models: M.S.P., F.W., and P.V. Acquisition of MRI data: M.S.P., H.T., and O.A.O. Conduction of CFD simulations: J.K., F.G., and P.B. MRI data analysis and evaluation: M.S.P. and H.T. Data visualization: M.S.P., J.K., and F.G. Statistical analysis: M.S.P. Supervision and funding acquisition: J.B.H., O.J., N.L., M.S.P., and P.B. Writing— original draft preparation: M.S.P., J.K., and F.G. Writing–review and editing: M.S.P, H.T., J.K., F.G., O.A.O., P.B., P.V., L.B., F.W., J.B.H., O.J., and N.L. All authors have read and approved the published version of the manuscript.

## Funding

## Competing interests
The author(s) declared the following potential conflicts of interest with respect to the research for this article: F.W. and N.L. have received fees for consulting and speaking from Balt International SAS, Acandis GmbH, Stryker Corporation, and Cerus Endovascular.
