## [Transparent Peer Review file · Communications Medicine]

Evaluating flow modulating treatment response in intracranial aneurysms using black-blood MRI in vitro.

Corresponding Author: Dr Mariya Pravdivtseva

Version 0:

Reviewer comments:

Reviewer #1

(Remarks to the Author)

The authors submitted a manuscript using what they call black-blood (BB) MRI in combination with 4D flow MRI to assess intracranial aneurysms in in-vitro models of two patients. They aim to use the BB MRI sequence as a prognostic biomarker for the success of the treatment of intracranial aneurysms using flow-modulating devices. The title of the manuscript clearly shows one of the significant issues of this manuscript. I am sure that the authors do not mean that the imaging sequences are biomarkers, but derived parameters or signals at specific locations are potential prognostic markers. However, the language used throughout the manuscript lacks preciseness, which is required to allow reproducibility and thus opens space to speculations the readers must make.

It has been unclear for a long time what kind of BB sequence was used. Once the reader finds out, he/she should not be surprised by the signal behavior. Researchers such as Edelman or, recently, Henningsson have already published much of it (<https://doi.org/10.1002/jmri.27399>).

In addition, reading the abstract, the reader long believed that the authors had 20 patients from whom they extracted individual aneurysms. Later, in the methods section of the main manuscript, one finds out that besides the basic two straight tubes, only two patient data sets were used.

The manuscript also lacks statistics. The authors claim a signal increase or decrease but provide no p-value. A signal remains similar if it is not statistically proven, even if it visually looks like it is increasing/decreasing. This is how it is done in scientific work; I am aware that, unfortunately, this is often not the case in non-scientific work.

Also, CFD results are not shown. I wonder why the authors went through the hazard of simulating if the results are not shown. They only appear as visualization once in Figure 7.

The authors did not show how to make this method reproducible or reliable or how it can be changed to become a quantitative measure. They showed the sensitivity of BB TSE to frequency encoding direction, voxel size, and so on, but unfortunately, they did not suggest how to make this quantitative.

The manuscript needs some work on the language. The formulations are not very precise, and there are few details about the experimental setup and simulations.

While the work is interesting and important, I do not understand why the authors want to replace 4D flow MRI with BB MRI. At least, this is how the manuscript reads. If the authors did not mean that, they should reformulate. I think the results show that they should be complementary, given the lack of reproducibility, and the authors should actually discuss how they can be complementary. Another option would be to combine the information gained with CFD simulations, but also that was not done.

Below are some more detailed critique points.

Abstract:

- It was not proven that 4D flow can predict the outcome of an intracranial aneurysm (IA) treatment. There are potential parameters derived from 4D flow that can.
- I find the sentence "However, it is prone to metal artifacts." strange. All MRI methods, as well as CT and X-ray imaging, are prone to metal artifacts. The only way to avoid those is not to use MRI. I think this needs reformulation.
- The term "Black-blood (BB) MRI" is very generic. BB can come from using SE or IR pulses. More details are needed on what kind of BB sequence this is.
- It would be beneficial to provide SI units for flow rate, so ml/s
- What kind of aneurysms? Which sizes? Was it including the feeding artery and outflow artery or was it just the aneurysms? Or was it even the entire CoW?

- How can BB be scanned in a model? Did you use blood as a fluid?
- What is meant by velocities are calculated? This is just a conversion from phase to velocity.
- Unclear what is meant by that "with 4D flow MRI verified by numerical simulations". Did you use CFD simulations of the exact geometry?
- In results: "BB signal decreased with increasing flow rate in straight vessels. "
- > what is meant by "BB signal is decreased"? Does this mean that the signal in regions with blood is lower? Or is it lower overall? Or do they mean that the blood signal suppression is lower, so it is not working well?
- > One does not know if this was a BB sequence because of saturation, spoiling or IR. Was it GRE or SE? If this was a GRE with flow spoiling, this effect is not surprising, but we do not have information on velocities or sequence types and timings.
- "FMDs reduced intra-aneurysmal flow ($w_{in}=2$ vs. $w_{out}=12.2$ cm/s, p -value <0.01) while maintaining parent vessel flow ($w_{in}=29.4$ vs. $w_{out}=28.4$ cm/s, p -value = 0.18)." is this now results from the 4D flow MRI or the simulations?
- If this is a SE or GRE with flow spoiling, this result is well known and not surprising "Increased BB signal was colocalized with reduced flow, and the effect was reproducible across models."
- No information was given on the 4D flow sequence. Was it a two-sided 4-point encoding? Or was it one-sided with flow compensation?
- Conclusion: the sentence again lacks preciseness "Increased BB MRI signal found in treated IA models ". where was the signal increased? In the aneurysm sac or inflow or outflow?

Main manuscript

Introduction

- The first paragraph of the introduction is most important and needs to be understandable for both technical and medical readers
- o The reader needs more information. For example: Why is it the goal to promote thrombosis?
- o What is the meaning of this sentence for your reasoning? "However, treatment complications like delayed aneurysm rupture and reperfusion still occur" What is delayed rupture? During treatment? What is reperfusion, and why?
- o Why velocity reduction? The authors need to first describe why FDMs are chosen and what they are supposed to achieve.
- o Again, preciseness is lacking: I am sure that the simulations did not predict the outcome. Instead, certain parameters resulting from the simulations are used. Though was it ever proven? It was only shown that they are potential biomarkers.
- Page 3, line 17 ff: what are image-based simulations? Do the authors mean CFD using medical images to create anatomies and boundary conditions?
- Page 4, line 16 ff: the depiction of blood flow of a SE sequence is not comparable to the parameters that can be derived from 4D flow MRI. Please reformulate
- Now that I know that the authors used a SE sequence, how was the tissue differentiated from calcifications or static tissue? Why was it faster? Was it a FSE?

Methods

- Page 5, lines 24 ff: Missing reproducibility - I only count 5 models, the original ICA aneurysm and the original BA aneurysm plus three BA alterations. Why were the 20 models?
- Reproducibility: What was the recipe for the water glycerin mixture, and what was the viscosity?
- Reproducibility: missing description of flow curves for the different flow rates. Were the flow curves individually adapted for the different flow rates, or is it just (like most do) the same flow curve with an offset?
- 4D flow with FA of 8 is very low. Ernst angle is usually between 13 and 15.
- 4D flow and 2D PC lack information on VENCs
- 2D PC is not a "2D flow MRI". The term "flow" can only be used if it is a 3-directional flow encoding
- Was the TOF a single slab?
- Read-out direction would be of interest for 4D flow concerning flow direction
- It is phase offset correction and not offset phase correction.
- Phase offsets in phase-contrast imaging are not linear and typically are corrected using higher degree splines especially in phantom experiments
- Page 8, line 8: what was the distance between the analysis planes
- Did the authors use 3D volume ROIs for the spatial velocity averaging?
- How was the temporal median velocity determined?
- What are the details for the two-times spatial interpolation? And why?
- This sentence is unclear for several reasons "except for the model with FD1 and FD3 due to the near-zero velocities in the aneurysm sac": first, what is FD1 and FD3 and two, what kind of interpolation and why (same as above)
- Missing information on the material of phantoms, how the setup looked like, how it was made sure the pressure was similar to in vivo, and how the inflow profile was made sure to be the same as in vivo.
- The reader finds out about without and with devices (IFD1-10) for the first time here (page 11, line 3)
- Unclear what this means: "Flow and pressure curves measured at the model outlets before MRI ". How were flow and pressure measured then? In the actual patients?
- A precise description of CFD is lacking. Ideally, the authors would provide the model as an open-source download.

Results

- Page 11, line 24: "Different flow rates resulted in a similar time-dependent flow curve featuring one maximum at about 0.2 ms." The reason for this might be that the same flow curves were used just with an offset to increase the flow rate. The authors should critically inspect the flow curves of the pumps.
- Line 25: If it increases, there must be a p-value. Same for decrease
- All results lack p-values, seems like no statistics were performed

- I wish I knew what FD 2, FD 4, etc were
- Page 18, line 11: is this the result of 4D flow or CFD?
- Missing CFD visualizations and results
- Missing correlation coefficients
- Figure 9 is just a scatter plot, how did the authors investigate correlation and came to the conclusion that it was correlating?
- Page 19, line 9 ff: I do not understand how the authors determined a 3D velocity vector from BB MRI. Though I am happy that there are finally p-values.

Discussion

- The discussion is poorly written and does not follow the typical structure. First, the methods and results should be summarized and compared systematically to the literature. The subheadings are unusual. The authors immediately jump into discussing details.
- There is bright blood and black blood methodology in MRA. The authors should avoid talking about "bright BB signal".
- The sequence is an FSE sequence and thus does not have a "spin-echo nature." Please reformulate the entire sentence, it has poor structure and is complicated. (page 22, line 9).
- I do not find the results very astonishing and would not say that this study brought new insights about using BB MRI for angiography. The strength of this study is systematic quantitative analysis. Unfortunately, there are missing details on many of the steps, many things remain unclear, and nothing can be reproduced. However, the conclusions that the authors draw from this in vitro study are not worth this publication. The discussion needs to be reworked and needs to have a different focus. I wish the authors would have shown how to determine velocities from the BB MRI data or anything new.
- Page 22, line 14 ff: 4D flow MRI measures time-resolved 3D velocities. BB TSE cannot measure velocities. I can deal better with artifacts; we all know that. But why is it better than 4D flow? The information is like always with MRI complementary.
- I am missing an outlook on how the authors want to use the BB TSE sequence in the future to determine velocities (without acquiring 2D PC or 4D flow MRI). I do understand the increased signal could be a surrogate marker, but the authors did not show how they want to provide reliable and reproducible information on flow reduction quantitatively.
- The authors highly speculate that BB TSE can be used for treatment outcome. But they do not formulate it carefully enough and make claims on what they do not investigate.

Figures

Please add a figure of the experimental setup.

Supplemental Material

- missing information on pump and flow curve
- A sketch of the experimental setup would be desirable for reproducibility reasons
- I wish I had some knowledge from the supplementary material before reading the manuscript. Especially regarding the different FDM phantoms. I think a lot of information, especially when using abbreviations such as FD 1 or FD 1-2, needs to be mentioned in the manuscript itself.
- Also here, no statistical analysis results are presented, and no p-values were provided.
- Add a figure of the different 20 aneurysm phantoms

Reviewer #2

(Remarks to the Author)

The authors present an interesting work looking at the use of black blood MRI with and without MSDE preparation to look at the residual flow in an aneurysm after treatment with an endovascular device. Overall, the manuscript is well written and is of interest. However, there are manifold details that I believe should be addressed before publication:

Major:

A large amount of effort was taken to describe how the vasculature was derived from a specific patient, which is important details, but the authors then make drastic edits to the morphology of the aneurysm. Would it not have been better to take a population averaged vasculature, as once you begin to edit the aneurysm morphology its not a patient specific model anymore.

The use of a resin with high stiffness was unlikely to be an appropriate choice for modeling vasculature, why wasn't an elastic material such as silicon and PVA used?

It appears that constant flow was used, not pulsatile. I believe that pulsatile flow would have a large impact on the degree of blood signal nulling. The authors either need to justify this decision, perform experiments showing it has no effect, or I would suggest repeating experiments with physiological flow conditions.

Although the authors are correct that the BB images have less metal artifacts, it is not free of them. For any quantitative results they need to be clear that areas of artifact were not considered, and this should be carried out between all models. It would not be consistent to look at different RoIs between devices and make any claim about flow.

For the statistical section, it is unlikely to be consistent with the assumptions of the test that the control (no FMD) can be used multiple times for comparison. I would suggest a different statistical approach, or perform multiple control experiments (before/after implant) in the same model to allow validity to the methods used.

In the discussion: its stated that there was a linear relationship between the BB signal and the flow in the straight model, then the conclusion that this could provide quantitative information about aneurysm occlusion. This is problematic for two reasons: First, no statistics or quantification is performed to prove the linear relationship, so this would have to be done. Second and more importantly, this relationship does not appear to be true in any of the aneurysm models (only straight), and thus the conclusion is likely refuted by the data.

Minor:

At the beginning of the introduction, aneurysms do not reperfuse, the correct term is recanalizations.
An explanation of why water was used in one model and glycerin-water was used in a different model is needed. Viscosity could have large effects around edges.

Reviewer #3

(Remarks to the Author)

This manuscript focuses on addressing imaging artifacts encountered during MRI scans of intracranial aneurysms. The authors assessed the performance of 4D flow MRI and black blood (BB) MRI techniques in the presence of metal implants, such as stents. Furthermore, they proposed the potential use of BB MRI as a biomarker for predicting treatment success based on the reduction in blood flow following treatment. A reduction in blood flow of 35% or more with a flow-modulation device was considered indicative of a successful treatment outcome.

The manuscript is well-written, and the figures are of high quality. It explores an intriguing interdisciplinary topic in quantitative MRI and methods for evaluating intracranial aneurysm treatments. However, there are several issues and questions that the authors need to address before the manuscript can be considered for publication.

1. a) "However, WOT is influenced by factors such as contrast injection technique, which is often performed manually and may vary between pre- and post-FMD deployment." Please provide a reference to support this statement.
b) DSA is a well-established technique that provides 2D blood flow patterns. How does the use of 3D flow patterns contribute to clinical outcomes or decision-making?
2. The authors utilized a patient-derived model and a pulsatile pump to simulate blood flow. Was the elasticity of the vessel considered in the in vitro model and simulation model?
3. In the sub-section "Intracranial aneurysm flow models" of the "Materials and Methods", the maximum dimension values are mentioned. Does this refer to the length or the diameter?
4. Line 178: Why was the temporal median used for quantification?
5. Line 227: Why is a normalization of the BB signal necessary? Are these signals subject to variation under different conditions?
6. The observation of higher BB signals with FDM is intriguing. It would be valuable to include a failure model where FDM does not result in a significant reduction in blood flow.

Version 1:

Reviewer comments:

Reviewer #1

(Remarks to the Author)

Dear Authors,

thank you for the thorough and extensive edits. The manuscript improved significantly and gained clarity. Also the new figures in the supplement as well as the additional information in the main manuscript that was moved from the supplement really helps the understanding.

I do not have further comments and recommend acceptance.

Reviewer #2

(Remarks to the Author)

The authors have considered and answered my questions. In the case where they have not made direct adjustment, they have given an acceptable rationale.

I have no further comments

Reviewer #3

(Remarks to the Author)

Point-by-Point Response to the Reviewers' Comments

Dear Reviewers,

Thank you very much for your thorough and thoughtful review of our manuscript. We have addressed your comments and suggestions to the best of our ability and are confident that the manuscript is now clearer and more reproducible by other research centers. We sincerely appreciate your invaluable contribution to improving the quality of this work.

The full list of references cited in this document is provided at the end.

Kind regards,

Authors of the manuscript COMMSMED-24-1245A

Referee expertise:

Referee #1: flow MRI

Referee #2: brain MRI, aneurysms

Referee #3: clinical brain MRI

Reviewers' comments:

Reviewer #1 (Remarks to the Author):

General comments

Comment 1.1: The authors submitted a manuscript using what they call black-blood (BB) MRI in combination with 4D flow MRI to assess intracranial aneurysms in in-vitro models of two patients. They aim to use the BB MRI sequence as a prognostic biomarker for the success of the treatment of intracranial aneurysms using flow-modulating devices. The title of the manuscript clearly shows one of the significant issues of this manuscript.

I am sure that the authors do not mean that the imaging sequences are biomarkers, but derived parameters or signals at specific locations are potential prognostic markers. However, the language used throughout the manuscript lacks preciseness, which is required to allow reproducibility and thus opens space to speculations the readers must make.

Answer 1.1: Thank you for pointing this out. We have reformulated the title of the paper, as well as clarified language in other parts of the manuscript.

Old text: Black-blood MRI for assessment of flow modulation in intracranial aneurysms— a potential prognostic biomarker for treatment response? A prospective in vitro study.

New text: Evaluating flow modulating treatment response in intracranial aneurysms using black blood MRI: A prospective in vitro study

Comment 1.2: It has been unclear for a long time what kind of BB sequence was used.

Answer 1.2: thank you for this comment. We have clarified the type of MRI sequence with Black blood contrast used from the beginning, in the abstract.

Old text: All models with (w.) and without (wo.) FMDs underwent 4D flow and BB on a 3T MRI system.

New text: All models with and without FMDs underwent 4D flow MRI with non-symmetric gradient encoding and flow compensation, and a 3D turbo spin echo sequence with variable flip angle (Volume ISotropic Turbo spin echo Acquisition, VISTA), with BB contrast on a 3T MRI system (Ingenua CX, Philips).

Comment 1.3. Once the reader finds out, he/she should not be surprised by the signal behavior. Researchers such as Edelman or, recently, Henningson have already published much of it (<https://doi.org/10.1002/jmri.27399>).

Answer 1.3: We agree that the suppression of blood signal using turbo spin echo sequences is known and depends on flow velocity, as explained by (Henningson et al., n.d.). It is also true that TSE MRI has been shown to fail in suppressing blood signals in untreated aneurysms with slow flow (Cornelissen et al., 2019a; Pravdivtseva et al., 2021).

Here, we explore this known and expected effect in a new way: to highlight slow flow in treated aneurysms. One may argue that this (and much more) information is available by phase contrast, gradient echo-based 4D flow, but we find that 3D TSE is more robust when it comes to artefacts (as expected from a TSE sequence).

To our knowledge, the increased MRI signal on TSE BB MRI in treated IA, although expected, has not been investigated before. Here, we analysed the effect in vitro, showing that an increased BB signal was observed in basilar and intracranial aneurysm models treated with FMDs. While some studies have evaluated BB signal in aneurysms during long-term follow-up after thrombosis has already developed (Guan et al., 2017; Larsen et al., 2020a), the immediate response was not investigated.

Most of the literature summarized above has been mentioned in the introduction previously. To improve the readability and understanding of our manuscript, we have rephrased the introduction to highlight better what was already done and what our study brings.

Old text: Spin-echo-based black-blood (BB) MRI is much faster and less sensitive to metal artifacts than gradient-echo-based 4D flow MRI while still depicting flow (Pravdivtseva et al., n.d.). BB MRI can depict slowly flowing blood as bright, making it a promising tool for assessing IVR immediately after FMD placement and potentially predicting treatment outcomes (Cornelissen et al., 2019a; Kalsoum et al., 2018; Pravdivtseva et al., n.d.). Moreover, contrast-enhanced (CE) BB MRI is a widely clinically used sequence, e.g., for evaluating IA wall assessment (Edjlali et al., 2018; Larsen et al., 2020b, 2018; Shimonaga Koji et al., 2018). Previous studies have also employed CE BB MRI to visualize long-term IA healing after coiling (Elsheikh et al., 2020; Larsen et al., 2020a), stenting (Guan et al., 2017; Petridis et al., 2018), and aspirin therapy (Petridis et al., 2018).

New text: Spin-echo-based black-blood (SE BB) MRI is less sensitive to metal artifacts than gradient-echo-based 4D flow MRI. Its ability to suppress blood signal depends on the interplay of sequence parameters and flow velocity: fast-flowing blood appears dark, while slow flow may appear bright (Henningsson et al., n.d.). This known flow-dependent contrast mechanism can be leveraged to visualize regions of slow flow, making it a promising tool for assessing IVR immediately after FMD placement and potentially predicting treatment outcomes. Prior studies have shown areas of increased SE BB MRI signal corresponding to the slow flow regions in untreated aneurysms in vitro (Cornelissen et al., 2019a; Pravdivtseva et al., n.d.). Moreover, BB MRI was employed to visualize long-term IA healing after coiling (Elsheikh et al., 2020; Larsen et al., 2020a), stenting (Guan et al., 2017; Petridis et al., 2018), and aspirin therapy (Petridis et al., 2018).

Comment 1.4: In addition, reading the abstract, the reader long believed that the authors had 20 patients from whom they extracted individual aneurysms. Later, in the methods section of the main manuscript, one finds out that besides the basic two straight tubes, only two patient data sets were used.

Answer 1.4: Thank you for highlighting this issue. We used the number 20 because 20 individual IA models were 3D-printed, and 20 separate flow experiments were conducted. To prevent any confusion, we have revised the sentence to clarify the distinction between the number of unique patient-derived models and the number of printed copies used in the experiments.

Old text: 20 IA models of two patients and two straight vessels were 3D-printed and supplied with pulsatile flow (0 to 320 ml/min).

New text: 22 independent flow experiments were conducted using 20 IA models and two straight vessel models. The IA models were based on two unique patient-specific anatomies—one from the basilar artery (BA) and one from the internal carotid artery (ICA). To enable systematic comparison, the BA model was modified to accommodate various FMD sizes. Both BA and ICA models were 3D-printed multiple times to compare different FMD designs under consistent anatomical conditions.

Comment 1.5: The manuscript also lacks statistics. The authors claim a signal increase or decrease but provide no p-value. A signal remains similar if it is not statistically proven, even if it visually looks like it is increasing/decreasing. This is how it is done in scientific work; I am aware that, unfortunately, this is often not the case in non-scientific work.

Answer 1.5: Thank you for this valuable comment. We agree that statistical analysis is crucial to support visual interpretations. This is the reason why we performed a two-sided Wilcoxon test on the median BB signal values among all models, and the resulting p-values were presented in the abstract and main text already in the first version of the manuscript.

In addition, our initial analysis included voxel-wise comparisons of BB MRI signals for individual models within 3D ROIs of the aneurysm and parent vessel, performed using the Kruskal-Wallis test. Post hoc Dunn pairwise tests were conducted when comparing more than two groups. As expected with large voxel counts, nearly all comparisons yielded statistically significant differences—even in cases where visual changes were minimal. To avoid potential misinterpretation or overemphasis of statistical significance due to large sample sizes, we initially opted to present signal distributions using violin plots, which we felt better reflected the differences of interest. In response

to your comment and to improve transparency, we have now included the p-values from the Kruskal-Wallis and post hoc Dunn tests in the supplementary material, Table S2-3, S5, S7.

We have added the description of statistical analysis to the methods section of the abstract, as well as improved the statistical description in the main text.

New text Abstract: A voxel-wise Kruskal-Wallis test was used to assess the differences in the signal in the aneurysm and parental vessel with and without FMD. A two-sided Wilcoxon test was used to assess the differences in median SE BB signal between all IA models. The Spearman correlation coefficient (ρ) was calculated to assess the relation between the BB signal and velocity.

New text Main text: Normalized BB signal distributions of empty IA models were compared with those of the IA models with implanted FMDs using the Kruskal-Wallis test. In addition, if more than two groups were tested simultaneously, the post hoc Dunn pairwise comparisons with Benjamini-Hochberg p-value adjustment were used. The treatment effect (Δ) of FMDs on BB signal and velocity was calculated as follows, $\Delta BB = \frac{BB_{wo\ FMD} - BB_{with\ FMD}}{BB_{wo\ FMD}}$ and $\Delta velocity = \frac{velocity_{wo\ FMD} - velocity_{with\ FMD}}{velocity_{wo\ FMD}}$, respectively.

The resulting ΔBB and $\Delta velocity$ values were tested against a zero-effect hypothesis using a two-sided Wilcoxon signed-rank test, with statistical significance defined as $p < 0.05$. The relationship between velocity and BB signal was assessed using the Spearman correlation coefficient (ρ). The statistical analyses were performed in MATLAB R2022a and Python 8.2.0 (Jupyter Notebook 6.4.8, library `scipy.stats`).

Comment 1.6: Also, CFD results are not shown. I wonder why the authors went through the hazard of simulating if the results are not shown. They only appear as visualization once in Figure 7.

Answer 1.6: Thank you for this comment. Quantitative CFD results were reported in the original document on page 19, lines 354–359, and were also summarized in Table S7 of the Supplementary Material. Additionally, a scatter plot illustrating the relationship between CFD-derived flow reduction and BB MRI signal was presented in Figure 9.

Our primary use of CFD in this study was to quantify flow reduction caused by IFDs and to validate that this reduction corresponds with an increased BB MRI signal. We chose this approach because 4D flow MRI was not reliable for these devices due to severe metal artifacts, making CFD a necessary supporting method.

Our focus in this manuscript is on MRI-based assessment. However, we recognize that additional hemodynamic parameters (e.g., wall shear stress, oscillatory shear index) may be of interest to the readers. These aspects are discussed in detail in a complementary publication (Korte et al., 2023), which was cited in the method section, page 11, line 9.

To improve clarity, we have also cited this publication in the results section.

New text: The effect of IFDs on other hemodynamic parameters (e.g., wall shear stress, oscillatory shear index) is discussed in detail in our previous publication (Korte et al., 2023)

To improve transparency, we have also moved Figure S5 from the supplementary into the main text to show CFD-derived velocity maps for all IFD1–10. The violin plots have been moved to Figure 9 for better organization.

Comment 1.7: The authors did not show how to make this method reproducible or reliable or how it can be changed to become a quantitative measure. They showed the sensitivity of BB TSE to frequency encoding direction, voxel size, and so on, but unfortunately, they did not suggest how to make this quantitative.

Answer 1.7: Thank you for your insightful comment. You are absolutely right—this study is an initial step toward evaluating the potential of BB MRI for assessing flow reduction after treatment with flow-modulating devices. We do not claim that BB MRI in its current form or later is a quantitative method. Rather, our aim was to explore its sensitivity to flow reduction and examine whether consistent signal changes could be detected under different conditions. Although we, as physicists, prefer a quantitative approach, qualitative assessment can already improve radiological readings of images.

We acknowledge that BB MRI signal depends on several acquisition parameters, such as voxel size, suppression modules. These influences have been demonstrated both in our study and in prior literature (Cornelissen et al., 2019a, 2019b; Pravdivtseva et al., 2021).

In our work, we tested the reproducibility of signal increases in several ways:

1. Across different aneurysm types (ICA vs. BA),
2. Across device types (intraluminal flow-diverters vs. intrasaccular devices),
3. Across implantation variations (multiple independent devices implanted in different geometries, e.g., four devices were implanted in BA model 3),
4. Under different fluid viscosities (glycerol/water vs. water),
5. With varied MRI acquisition settings (MSDE on/off, different readout gradient orientations).

In all these cases, we observed a consistent post-treatment signal increase in the aneurysm sac, suggesting that BB MRI signal increase may serve as a surrogate marker of flow reduction. However, we agree that further work is needed to standardize acquisition protocols and quantify the relationship between signal intensity and flow characteristics.

We have now revised the discussion to clarify this position and to emphasize the current limitations and potential of this method.

New text Discussion/BB MRI parameters and reproducibility: Unlike velocity measurements from 4D flow MRI, BB MRI signal intensity is influenced by a range of acquisition parameters that impact the efficiency of flow suppression. Prior studies have shown that changes in voxel size and the inclusion of additional suppression modules such as MSDE can alter BB signal in untreated aneurysms (Cornelissen et al., 2019a; Pravdivtseva et al., 2021).

In our study, we further demonstrated that readout gradient orientation and MSDE application affect flow suppression efficiency, particularly in straight vessel models and parent arteries. For example, aligning readout gradients with flow direction improves suppression due to enhanced intra-voxel dephasing (Henningsson et al., n.d.). This effect, was less pronounced in aneurysm sacs, likely due to the complex, multi-directional vortex flow patterns.

Despite these dependencies, a consistent increase in BB MRI signal within the aneurysm sac after treatment was observed across all tested variations: 1) BB parameters (MSDE on/off, readout direction); 2) Aneurysm geometry (ICA vs. BA); 3)

Device type (flow-diverters and intrasaccular devices); 4) Repeated independent implantations of the same device (e.g., four devices were implanted in BA model 3); 5) Different fluid viscosities (glycerol/water in ICA experiment vs. water in BA experiment). Although the extent of signal increase varied, the effect was robustly present. This suggests that BB MRI may provide a reproducible, semi-quantitative indication of post-treatment flow stasis—if acquisition parameters are kept constant between pre- and post-implantation imaging. Future work should focus on standardizing acquisition protocols and exploring quantitative thresholds that may allow BB MRI to transition toward a more formal diagnostic role.

Comment 1.8: The manuscript needs some work on the language. The formulations are not very precise, and there are few details about the experimental setup and simulations.

Answer 1.8: Thank you, it was addressed to the best of our knowledge.

Comment 1.9: While the work is interesting and important, I do not understand why the authors want to replace 4D flow MRI with BB MRI. At least, this is how the manuscript reads. If the authors did not mean that, they should reformulate. I think the results show that they should be complementary, given the lack of reproducibility, and the authors should actually discuss how they can be complementary. Another option would be to combine the information gained with CFD simulations, but also that was not done.

Answer 1.9: Thank you for highlighting this important point. We would like to clarify that we do not intend to replace 4D flow MRI with BB MRI. Rather, our aim is to explore BB MRI as a complementary tool for assessing flow reduction, especially in cases where 4D flow MRI is impaired by metal artifacts.

For example, we observed that when intrasaccular devices are implanted in small aneurysms, 4D flow MRI fails to capture velocity data in the aneurysm sac due to severe metal-induced artifacts.

We also acknowledge the potential of combining information from CFD simulations. However, in vivo application of CFD remains limited by the difficulty of acquiring precise boundary conditions and accurate device placement data. In contrast, TSE BB MRI is routinely acquired in clinical workflows for aneurysm assessment, and our goal was to make use of this already-available data to gain additional insights.

To make this clearer, we have revised both the Introduction and Discussion sections of the manuscript to reflect the intended complementary role of BB MRI.

Old text Introduction: We hypothesize that the BB signal increases after FMD placement, reflecting IVR, and could be a 3D surrogate for flow reduction.

New text Introduction: We hypothesize that the BB signal increases after FMD placement, reflecting IVR, and could be a 3D surrogate for flow reduction. This may complement velocity measurements obtained from 4D flow MRI, particularly in cases where metal artifacts prevent reliable velocity assessment within the aneurysm sac.

New text Discussion/BB MRI parameters and reproducibility: We suggest it could serve as a complementary surrogate marker, particularly in cases where metal-induced artifacts limit the utility of velocity-based techniques. Future work should focus on standardizing acquisition protocols and exploring quantitative thresholds that may allow BB MRI to transition toward a more formal diagnostic role. Since SE BB MRI is

already routinely performed for vessel wall assessment in patients with IAs, leveraging this existing data to gain flow-related insights could provide additional benefit to patient care.

Detailed comments

Below are some more detailed critique points.

ABSTRACT:

Comment 1.10: It was not proven that 4D flow can predict the outcome of an intracranial aneurysm (IA) treatment. There are potential parameters derived from 4D flow that can.

Comment 1.11: I find the sentence “However, it is prone to metal artifacts.” strange. All MRI methods, as well as CT and X-ray imaging, are prone to metal artifacts. The only way to avoid those is not to use MRI. I think this needs reformulation.

Comment 1.12: The term “Black-blood (BB) MRI” is very generic. BB can come from using SE or IR pulses. More details are needed on what kind of BB sequence this is.

Answer 1.10, 1.11, and 1.12: yes, you are right.

Old text: 4D flow MRI detects intra-aneurysmal flow reduction after treatment with flow-modulation devices (FMDs) and can predict intracranial aneurysm (IA) treatment outcome. However, it is prone to metal artifacts. Black-blood (BB) MRI is less affected by these artifacts. This study investigates whether BB signal changes in IAs reflecting the flow reduction post-treatment and thus can serve as a prognostic biomarker for successful treatment.

New text: 4D flow MRI detects intra-aneurysmal flow reduction after treatment with flow-modulation devices (FMDs). Flow reduction measured with 4D flow MRI, is associated with the treatment outcome of intracranial aneurysms (IAs). However, gradient echo-based 4D flow MRI is sensitive to metal artifacts. Spin-echo-based black-blood (SE BB) MRI is less sensitive to these artifacts. This study investigates whether SE BB signal changes in IAs reflect the flow reduction post-treatment and thus can serve as a prognostic biomarker for successful treatment.

Comment 1.13: It would be beneficial to provide SI units for flow rate, so ml/s

Answer 1.13: the flow rate was changed from ml/min to ml/s throughout the main manuscript and supporting material.

Comment 1.14: What kind of aneurysms? Which sizes? Was it including the feeding artery and outflow artery, or was it just the aneurysms? Or was it even the entire CoW?

Answer 1.14: Thank you for your question. To improve transparency and reproducibility, we have moved the detailed description of model production from the supplementary material into the main text. Additionally, we have added a new figure that illustrates the entire vascular models, including the aneurysm sac, feeding (parent) arteries, and outflow branches, along with the relevant size measurements.

The models were based on two unique patient-specific intracranial aneurysm anatomies—one located in the basilar artery (BA) and one in the internal carotid artery (ICA). Both included the aneurysm as well as the relevant segments of the parent vessels and downstream branches, but did not encompass the entire circle of Willis.

We have also updated the abstract to state the anatomical origins of the models clearly, and the aneurysm dimensions have been reported in the main Methods section.

New text Abstract: The IA models were based on two unique patient-specific anatomies—one from the basilar artery (BA) and one from the internal carotid artery (ICA).

Comment 1.15: How can BB be scanned in a model? Did you use blood as a fluid?

Answer 1.15: Thank you for your comment. In our experiments, we did not use real blood. The straight tubes and ICA models were perfused with a 40/60 glycerin/water mixture to mimic the viscosity of blood, while the BA models were perfused with water.

The terms black blood (BB) and bright blood originate from in vivo imaging, where they describe MRI contrast based on the presence or absence of the signal from flowing blood. However, these terms are also commonly used in in vitro studies where blood-mimicking fluids are used, even if no actual blood is present.

We chose to retain the term black blood in our manuscript to remain consistent with established terminology in the literature. Introducing a term like “black water MRI” (BW-MRI) would be unfamiliar and potentially confusing to readers. We believe this usage is justified and helpful for readers to understand the imaging contrast and its implications in the context of aneurysm flow modulation.

Comment 1.16: What is meant by velocities are calculated? This is just a conversion from phase to velocity.

Answer 1.16: Thank you for this helpful comment. We agree that the term “calculated” might be misleading if interpreted as a manual derivation from raw phase data. In our work, velocity fields were reconstructed automatically from 4D flow MRI phase data using the scanner’s reconstruction pipeline. When we referred to “velocities were calculated,” we meant that: 1) The velocity fields were extracted from the 4D flow MRI data as part of the standard reconstruction process, and 2) We subsequently computed temporal median velocity values within five predefined 2D planes placed in the aneurysm sac and parent vessels. To avoid confusion, we have updated the abstract text accordingly.

Old text: Velocities and BB signals were calculated in the aneurysm sac and parent vessel, with 4D flow MRI verified by numerical simulations.

New text: Median velocities were quantified within five 2D planes placed in the aneurysm sac and parent vessel. BB signal analysis was performed voxel-wise within the volumetric aneurysm and parent vessel regions.

Comment 1.17: Unclear what is meant by that “with 4D flow MRI verified by numerical simulations”. Did you use CFD simulations of the exact geometry?

Answer 1.17: Thank you for pointing this out. Yes, CFD simulations were performed using the exact same geometries as the MRI experiments, specifically for BA models 1–4. These models included intrasaccular devices, which caused severe metal-induced artifacts that impaired 4D flow MRI velocity measurements. To ensure consistency, the inlet flow and outlet pressure waveforms measured prior to the MRI experiments were used as boundary conditions for the simulations. Details on the CFD

setup were provided on page 11, lines 1-11. To avoid misunderstanding, we have rephrased the relevant sentence in the abstract for clarity.

Old text: Velocities and BB signals were calculated in the aneurysm sac and parent vessel, with 4D flow MRI verified by numerical simulations.

New text: In addition, numerical simulations were performed for BA models using the same geometries and boundary conditions as in the MRI experiments and compared to 4D flow MRI results.

Comment 1.18: In results: “BB signal decreased with increasing flow rate in straight vessels”:

 what is meant by “BB signal is decreased”? Does this mean that the signal in regions with blood is lower? Or is it lower overall? Or do they mean that the blood signal suppression is lower, so it is not working well?

 One does not know if this was a BB sequence because of saturation, spoiling or IR. Was it GRE or SE? If this was a GRE with flow spoiling, this effect is not surprising, but we do not have information on velocities or sequence types and timings.

Answer 1.18: Thank you for your thoughtful and detailed comment. To clarify, we used a Volume ISotropic Turbo spin echo Acquisition (VISTA) spin echo sequence, which provides black-blood imaging contrast. This information has now been added to the abstract for clarity. By stating that the “BB signal decreased with increasing flow rate,” we meant that the measured MRI signal intensities within the flowing volume (i.e., within the lumen of the straight tube) decreased as the supplied flow rate increased. This observation is consistent with known characteristics of SE BB MRI sequences, which exhibit limited flow suppression at slow velocities and more effective suppression at higher flow rates. We agree this effect is not unexpected. However, we believe it is still an important observation, as it demonstrates the reproducibility of this flow dependence in a simplified, straight-tube geometry, and is critical before progressing to more complex analyses involving aneurysm models and flow modulation devices.

To clarify the text in the Results section, we have revised the sentence as follows:

Old text: BB signal decreased with increasing flow rate in straight vessels.

New text: As expected, the SE BB MRI signal in a straight tube was found to decrease with increasing velocity ($\rho = -0.91$, $p\text{-value} < 0.01$).

Comment 1.19: - “FMDs reduced intra-aneurysmal flow ($w.=2$ vs. $w_o.=12.2$ cm/s, $p\text{-value}<0.01$) while maintaining parent vessel flow ($w=29.4$ vs. $w_o.=28.4$ cm/s, $p\text{-value} = 0.18$).” is this now results from the 4D flow MRI or the simulations?

Answer 1.19: Thank you for your question. The reported results refer specifically to flow velocities measured using 4D flow MRI. We have revised the relevant sentence accordingly.

Old text: FMDs reduced intra-aneurysmal flow ($w.=2$ vs. $w_o.=12.2$ cm/s, $p\text{-value}<0.01$) while maintaining parent vessel flow ($w=29.4$ vs. $w_o.=28.4$ cm/s, $p\text{-value} = 0.18$).

New text: FMDs reduced the normalized intra-aneurysmal flow velocities as measured by 4D flow MRI by 0.78 (p-value < 0.01) while the velocities in the parent vessel were not affected (p-value = 0.11).

Comment 1.20: - If this is a SE or GRE with flow spoiling, this result is well known and not surprising “Increased BB signal was colocalized with reduced flow, and the effect was reproducible across models. ”

Answer 1.20: Thank you for your comment. As noted in our response to Comment 1.3, we agree that it is well known that SE BB MRI sequences show poor flow suppression in regions with slow or stagnant flow. However, the novelty in our study lies not in the general behavior of SE BB sequences, but rather in demonstrating that this known effect can be systematically observed following the placement of FMDs in aneurysm models.

Our results show that increased BB signal was consistently colocalized with regions of reduced intra-aneurysmal flow after FMD deployment, and this pattern was reproducible across models. While the underlying MRI physics are established, the application of SE BB signal changes as a non-invasive indicator for evaluating the functional outcome of flow-diverting stents is a novel contribution of our work.

Comment 1.21: - No information was given on the 4D flow sequence. Was it a two-sided 4-point encoding? Or was it one-sided with flow compensation?

Answer 1.21: The 4D flow MRI sequence employed a multi-point velocity encoding scheme (MPS, Philips Healthcare), which uses non-symmetric 4-point velocity encoding to acquire velocity in all three spatial directions with flow compensation. We have clarified this detail in both the abstract and main text.

Old text Abstract: All models with (w.) and without (wo.) FMDs underwent 4D flow and BB on a 3T MRI system.

New text Abstract: All models with and without FMDs underwent 4D flow MRI using a 4-point, non-symmetric velocity encoding scheme with flow compensation, along with Volume Isotropic Turbo Spin Echo Acquisition (VISTA) for black-blood (BB) contrast on a 3T MRI system (Ingenia CX, Philips).

New text Main Text/Methods/MRI in vitro: The 4D flow MRI sequence employed non-symmetric 4-point velocity encoding to acquire velocity in all three spatial directions with flow compensation.

Comment 1.22: - Conclusion: the sentence again lacks preciseness “Increased BB MRI signal found in treated IA models ”. where was the signal increased? In the aneurysm sac or inflow or outflow?

Answer 1.22: corrected

Old text: Increased BB MRI signal found in treated IA models suggests its use as a potential prognostic biomarker for treatment efficacy in the presence of metal implants

New text: The increased BB MRI signal within the aneurysm sac in the treated IA models may be a potential surrogate marker for assessing flow reduction after aneurysm treatment, particularly in the presence of metal implants. Further in vivo validation is warranted to confirm its prognostic value and determine its clinical utility in patient monitoring post-FMD implantation.

MAIN MANUSCRIPT

INTRODUCTION

Comment 1.23: - The first paragraph of the introduction is most important and needs to be understandable for both technical and medical readers

o The reader needs more information. For example: Why is it the goal to promote thrombosis?

o What is the meaning of this sentence for your reasoning? “However, treatment complications like delayed aneurysm rupture and reperfusion still occur” What is delayed rupture? During treatment? What is reperfusion, and why?

o Why velocity reduction? The authors need to first describe why FDMs are chosen and what they are supposed to achieve.

o Again, preciseness is lacking: I am sure that the simulations did not predict the outcome. Instead, certain parameters resulting from the simulations are used. Though was it ever proven? It was only shown that they are potential biomarkers.

Answer 1.23: Thank you for the suggestions. The first paragraph of the introduction was revised accordingly.

Old text: Flow-modulation devices (FMDs) are effective in treating intracranial aneurysms (IAs) by reducing intra-aneurysmal blood flow^{1,2} and promoting thrombosis³. However, treatment complications like delayed aneurysm rupture and reperfusion still occur^{4,5}. Insufficient velocity reduction after FMD placement may lead to treatment failure⁶, whereas a high reduction (> 35%) predicts successful occlusion⁷. Therefore, immediate velocity reduction (IVR) may be a predictive biomarker for successful treatment.

New text: An intracranial aneurysm (IA) is a life-threatening cerebrovascular condition (Hackenberg et al., 2018) that can be treated endovascularly using flow-modulation devices (FMDs) (Chalouhi et al., 2013). These devices reduce intra-aneurysmal blood flow and promote aneurysm healing through neck endothelialization (Chalouhi et al., 2013) and thrombosis (Ravindran et al., 2020), ultimately leading to the occlusion of the aneurysm. Although most IAs are successfully occluded within 6–12 months after FMD treatment, some aneurysms may continue to fill with blood, enlarge, or even rupture after the treatment (Hou et al., 2020; Zhou et al., 2017). This highlights the need for close follow-up to identify aneurysm treatment failure early. It has been shown that insufficient flow velocity reduction after FMD placement is associated with treatment failure (Berg et al., 2018), whereas a high reduction (e.g., >35%) was related to successful occlusion (Ouared et al., 2016). Therefore, immediate velocity reduction (IVR) may serve as a diagnostic marker for evaluating treatment success.

Comment 1.24: - Page 3, line 17 ff: what are image-based simulations? Do the authors mean CFD using medical images to create anatomies and boundary conditions?

Answer 1.24: the sentence was clarified.

Old text: IVR can be estimated preoperatively through highly resolved image-based blood flow simulations, which have been shown to predict and replicate aneurysm treatment outcomes in vivo (Cebral et al., 2014; Hadad et al., 2021; Ouared et al., 2016; Sarrami-Foroushani et al., 2021).

New text: IVR can be estimated preoperatively and at no risk for the patients using computational fluid dynamics (CFD) simulations based on patient-specific vascular

geometries. CFD-derived IVR has been shown to potentially predict and replicate aneurysm treatment outcomes in vivo (Cebal et al., 2014; Hadad et al., 2021; Ouared et al., 2016; Sarrami-Foroushani et al., 2021).

Comment 1.25: - Page 4, line 16 ff: the depiction of blood flow of a SE sequence is not comparable to the parameters that can be derived from 4D flow MRI. Please reformulate

Answer 1.25: reformulated.

Old text: Spin-echo-based black-blood (BB) MRI is much faster and less sensitive to metal artifacts than gradient-echo-based 4D flow MRI while still depicting flow (Pravdivtseva et al., n.d.).

New text: Spin-echo-based black-blood (SE BB) MRI is less sensitive to metal artifacts than gradient-echo-based 4D flow MRI. Fast-flowing blood appears dark, while slow flow may appear bright (Henningsson et al., n.d.). This known flow-dependent contrast mechanism can be leveraged to visualize regions of slow flow, making it a promising tool for assessing IVR immediately after FMD placement and potentially predicting treatment outcomes.

Comment 1.26: - Now that I know that the authors used a SE sequence, how was the tissue differentiated from calcifications or static tissue? Why was it faster? Was it a FSE?

Answer 1.26: Thank you for this important question. Since our study was conducted entirely in vitro, the phantoms were composed of agarose gel (representing static tissue), 3D-printed material (representing vessel walls), and either water or a glycerin-water mixture (representing blood). Therefore, in our setup, there was no need to differentiate between static tissue and calcifications. This simplification allowed us to focus purely on flow-related effects and their depiction in MRI.

Regarding the sequence: yes, we used a fast spin-echo (FSE) sequence—specifically, the Volume ISotropic Turbo spin echo Acquisition (VISTA). When we described the sequence as “faster,” we referred to the shorter acquisition time required to cover the same field of view at comparable spatial resolution, in comparison to 4D flow MRI.

The 4D flow MRI sequence acquires time-resolved velocity data in all three spatial directions and velocity components, which inherently requires longer acquisition time. In contrast, SE BB imaging (like VISTA) provides high-resolution spatial information without time or velocity encoding, and thus can be acquired much more quickly.

However, we fully acknowledge that SE BB cannot replace the quantitative capabilities of 4D flow. It may only serve as a surrogate for detecting regions of flow stagnation, particularly in situations where metal artifacts impair 4D flow data quality. To avoid potential confusion, we have removed the comparison of acquisition times from the Introduction. We believe it is sufficient to highlight that SE BB is less affected by metal artifacts, which justifies its investigation as a potential complementary tool.

METHODS

Comment 1.27: - Page 5, lines 24 ff: Missing reproducibility - I only count 5 models, the original ICA aneurysm and the original BA aneurysm plus three BA alterations. Why were the 20 models?

Answer 1.27: Thank you for pointing this out. You are correct that we used two unique patient-specific aneurysm anatomies: one from the ICA and one from BA. The BA

model was systematically modified to create four variations, resulting in a total of five distinct anatomical configurations (one ICA and four BA variants).

The number "20" refers to the total number of individually 3D-printed physical models used in the study—not unique anatomical shapes. Each of the five base anatomies was reproduced multiple times to allow comparison of different treatment strategies (e.g., different FMD types or control vs. treated conditions), and each physical model was used in a separate flow experiment. This approach ensured experimental reproducibility under consistent anatomical conditions.

To clarify this distinction, we have revised the relevant parts of the Methods section and integrated the supporting information about model fabrication into the main text. The section "Methods / Intracranial Aneurysm Flow Models" now describes only the construction of the unique anatomical shapes, while the section "Methods / FMDs and In Vitro Deployment" explains which multiple replicas of each anatomical shape were fabricated to allow testing of different FMD devices.

Comment 1.28: - Reproducibility: What was the recipe for the water glycerin mixture, and what was the viscosity?

Answer 1.28: Thank you for your comment. We have used a glycerol-water mixture (40:60 by volume) with viscosity of 3.72 cP (Segur and Oberstar, 1951). We have moved the detailed description of the flow setup from the supplementary materials to the main manuscript to improve clarity and enhance reproducibility.

Comment 1.29: - Reproducibility: missing description of flow curves for the different flow rates. Were the flow curves individually adapted for the different flow rates, or is it just (like most do) the same flow curve with an offset?

Answer 1.29: Thank you for your comment. When generating the flow curves, we controlled only the mean flow rate values; the shape of the pulsatile waveform was not individually adapted for each flow rate. To improve transparency and reproducibility, we have included representative flow curves measured at the inlet using the flow sensor in the supplementary material, please see Figure S3.

Comment 1.30: - 4D flow with FA of 8 is very low. Ernst angle is usually between 13 and 15.

Answer 1.30: Thank you for this valuable comment. You are correct—the flip angle (FA) of 8° is commonly used in non-contrast 4D flow MRI (Bissell et al., 2023) to minimize signal saturation. However, since our in vitro setup included a contrast agent in the working fluid, a higher flip angle would have been more appropriate. We appreciate you pointing this out and will take it into consideration for future experiments.

Comment 1.31: - 4D flow and 2D PC lack information on VENCs.

Answer 1.31: We have now added the missing velocity encoding (VENC) values.

New text: To determine the appropriate velocity encoding (VENC) values, a series of 2D PC MRI examinations were conducted with VENCs ranging from 10 to 200 cm/s. Final VENC settings were chosen to be approximately 10% higher than the maximum measured velocity for each condition. Specifically: 1) Straight vessel with ID of 4 mm: VENCs of 10, 20, 60, 120, and 175 cm/s for flow rates of 0, 0.7, 1.4, 3, and 4.7 ml/s, respectively; 2) Straight vessel with ID of 6 mm: VENCs of 10, 15, 40, 65, and 85 cm/s for flow rates of 0, 1, 2.3, 4.8, and 5.4 ml/s, respectively; 3) ICA models: without FMDs:

VENC = 80 cm/s, with FD1: 20, 25, 100, 120 cm/s, with FD2: 80 cm/s, FD3–FD5: 50 and 120 cm/s. Multiple VENC settings were used for some models to optimize velocity sensitivity in both the aneurysm sac and parent vessel regions; 4) All BA models: VENC = 75 cm/s.

Comment 1.32: - 2D PC is not a “2D flow MRI”. The term “flow” can only be used if it is a 3-directional flow encoding

Answer 1.32: The term was corrected throughout the manuscript.

Comment 1.33: - Was the TOF a single slab?

Answer 1.33: TOF MRI was acquired using a multi-slab approach with four slabs. This detail has now been included in the manuscript.

Old text: TOF MRI was performed using a T1-weighted (T1w) gradient-echo sequence (GRE) to plan the subsequent sequences.

New text: TOF MRI was performed using a T1-weighted (T1w) gradient-echo sequence (GRE) with four slabs to plan the subsequent sequences.

Comment 1.34: - Read-out direction would be of interest for 4D flow concerning flow direction

Answer 1.34: Thank you for this comment. We performed two separate sets of 4D flow MRI experiments using ICA models. In the first experiment (Results are in the section “Effect of sequence parameters on the BB signal in the aneurysm model”), only the untreated ICA model and the model with FD1 were included; these were imaged with the readout direction parallel to the parent vessel. In the second experiment (Results are in the section “Effect of FMDs on BB signal in aneurysm models”), which included the untreated ICA model and models with FD1–FD5, the readout direction was perpendicular to the parent vessel. For all BA models, the readout direction was also set perpendicular to the parent vessel. This information has now been added to the main text.

New text: The readout encoding direction for 4D flow MRI was set parallel to the parent vessel for the ICA model without devices and with FD1 in the experiments described in Section 3.1 and 3.2. In the further experiments (Section 3.3), which included the ICA model without devices and with FD1–FD5 and all BA models, the readout was set perpendicular to the parent vessel.

Comment 1.35: - It is phase offset correction and not offset phase correction.

Answer: 1.35: corrected

Comment 1.36: - Phase offsets in phase-contrast imaging are not linear and typically are corrected using higher degree splines, especially in phantom experiments

Answer 1.36: Thank you for this valuable comment. Phase offset correction in our study was performed using the commercial software GTFlow (V3.1.12, Gyrotools, Switzerland), which only had an option to apply a linear correction model. We acknowledge that phase offsets in phase-contrast MRI are typically non-linear and that higher-order polynomial correction methods (e.g., Lorenz et al., 2014) are more accurate. To address this limitation and improve the precision of our velocity measurements, we are developing a custom Python-based post-processing pipeline that will enable more flexible and accurate phase correction using higher-order

models. To acknowledge this, we have added this information into the limitation section.

New text Limitations: Velocity phase offsets were corrected using a linear fit; however, higher-order polynomial corrections would likely improve accuracy and should be considered in future work.

Comment 1.37: - Page 8, line 8: what was the distance between the analysis planes

Answer: 1.37: Information was added

Old text: In 4D flow MRI datasets, five equidistant, cross-sectional 2D planes were placed in the parent vessel and aneurysm sac in areas unaffected by metal artifacts (Figure 1a).

New text: In 4D flow MRI datasets, five equidistant cross-sectional 2D planes were placed in the parent vessel and aneurysm sac (Figure 1, a middle and right). The distance between planes within a parent vessel was set to 1.5 mm for the ICA model and 2 mm for BA models. In the aneurysm sac this value varied depending on the aneurysm size and area unaffected by metal artifacts, specifically, for ICA-model it was 1.5 mm, for BA-model 1 and 2 – 0.5 mm, for BA-model 3 and 4 – 1 mm.

Comment 1.38: - Did the authors use 3D volume ROIs for the spatial velocity averaging?

Answer 1.38: Thank you for your question. No, we did not use 3D volume ROIs for velocity averaging. Instead, velocity measurements were performed using 2D ROIs. Specifically, the vascular lumen was manually outlined in five anatomically defined planes, and velocity was spatially averaged across each 2D ROI. These values were then used to calculate the temporal median velocity.

We opted against using the same 3D ROIs as in the SE BB analysis for two main reasons: (1) the aneurysm sac in 4D flow MRI was more susceptible to metal-induced artifacts than in SE BB imaging, and (2) the extent and location of artifacts varied substantially across different devices and deployments. Using a SE BB-derived 3D ROI would have risked including artifact-affected areas, reducing measurement accuracy. Therefore, we chose to manually select 2D planes that were free of artifacts to ensure more reliable velocity quantification.

New text: The vascular lumen was manually outlined in five predefined 2D planes to create 2D ROIs. For each temporal phase, Velocity values were spatially averaged within each ROI and then averaged across all planes (i.e., one plane for 2D PC MRI and five planes for 4D flow MRI). Finally, the temporal median velocity was calculated for each model.

Comment 1.39: - How was the temporal median velocity determined?

Answer 1.39: The temporal median velocity was determined as follows: for each temporal phase, the vascular lumen was manually outlined in five predefined 2D planes to create 2D ROIs. Velocity values were spatially averaged within each ROI and then averaged across all five planes (or one plane in case of 2D PC MRI). This yielded a single spatially averaged velocity per time frame. Finally, the temporal median was calculated from these per-frame velocity values across the cardiac cycle for each model.

We have added this clarification to the Methods section (see new text in comment 1.38).

Comment 1.40: - What are the details for the two-times spatial interpolation? And why?

Answer 1.40: Thank you for this important comment. A two-times linear spatial interpolation was initially applied to the velocity maps solely for visualization purposes—to provide smoother and more visually appealing images. However, we acknowledge that interpolation can misrepresent the original data resolution. To ensure full transparency and data fidelity, we have now removed the spatial interpolation from all velocity map visualizations in the manuscript. All figures now show the original, unaltered reconstructed data.

Comment 1.41: - This sentence is unclear for several reasons “except for the model with FD1 and FD3 due to the near-zero velocities in the aneurysm sac”: first, what is FD1 and FD3 and two, what kind of interpolation and why (same as above)

Answer 1.41: Thank you for your comment. Flow diverter stents FD1 and FD3 were introduced in the main manuscript under the section “FMDs and in vitro deployment” (Page 6, line 4). To enhance clarity, we have moved relevant information on FMDs from the supplementary material into the main text and added a new figure that visually links each FMD to its corresponding aneurysm model.

Regarding interpolation, velocity maps were previously displayed with two-times linear spatial interpolation for smoother visualization. However, we recognize the potential for confusion and distortion. In response, we have removed all interpolated images from the manuscript.

Comment 1.42: - Missing information on the material of phantoms, how the setup looked like, how it was made sure the pressure was similar to in vivo, and how the inflow profile was made sure to be the same as in vivo.

Answer 1.42: All vascular models were fabricated using Clear Photoreactive Resin Version 4 (Formlabs, USA) with stereolithography (SLA) 3D printing technology (Form 3B printer). The resin is composed of a proprietary blend of methacrylated oligomers and monomers with a photoinitiator. Based on the publicly available Safety Data Sheet (SDS), its components include:

- 1) Methacrylated oligomer: 7,7,9(OR 7,9,9)-trimethyl-4,13-dioxo-3,14-dioxo-5,12-diazahexadecane-1,16-diyl bismethacrylate,
- 2) Methacrylated monomer: Methacrylic acid, monoester with propane-1,2-diol,
- 3) Photoinitiator: Diphenyl(2,4,6-trimethylbenzoyl)phosphine oxide.

This material information has now been added to the main text.

Old text: Two straight tubes with inner diameters (IDs) of 4 and 6 mm were designed (Figure 1a; Fusion 360 2.0, Autodesk) and 3D-printed (Clear Photoreactive Resin, Form 3B, Formlabs, USA).

New text: All models were 3D printed using stereolithography (Clear Photoreactive Resin composed of a methacrylated oligomer, methacrylated monomer and photoinitiator; Form 3B, Formlabs, USA).

Flow setup details: Each phantom was integrated into a closed-loop flow circuit composed of a pump, fluid reservoir, flow and pressure sensors, and connecting tubing. The flow system differed slightly between model types: 1) Straight vessels and ICA models were driven using a pulsatile piston pump (PD-1100, BDC Laboratories,

USA), 2) BA models were supplied using a peristaltic pump (Ismatec MCP Standard, Cole-Parmer, USA).

Flow control: Only mean flow rates were controlled. Flow profiles were recorded using clamp-on ultrasonic flow sensors (ME8PXL-M12, Transonic Systems Inc., USA) positioned on the silicone tubing. Flow rates were selected to match physiological conditions as reported in the literature (e.g., BA: ~150 mL/min, ICA: ~250 mL/min). After adjusting pump speed, the flow curves were recorded.

Pressure control: Pressures were not actively regulated, but pressure build-up was relieved using an air valve at the reservoir. Pressure values were recorded using pressure transducers (Press-N-000, PendoTech, USA) that were placed at the tubing junctions using luer-lock connectors near the model outlets.

Old text: All 20 models were submerged in 3% agarose gel (Special Ingredients Ltd, UK) and integrated into a pulsatile flow setup (Supplementary 1.3). Mean flow rates were set to mimic physiological flows: 0-320 ml/min for straight vessels, 0 and 250 ml/min for ICA models, and 150 ml/min for BA models (Zarrinkoob et al., 2015). A blood-mimicking fluid, water-glycerin mixture for ICA models and pure water for BA models, was pumped with pulsatile (PD-1100, BDC Laboratories) and peristaltic (Ismatec MCP Standard, Cole Parmer, United States) pumps, respectively. The experimental fluids contained 0.3 mmol/l of gadobutrol (Gadovist, Bayer Vital, Germany).

New text: Before MRI experiments could be carried out, all models were submerged in an agarose gel with a 3% concentration of agar (Special Ingredients Ltd, UK) and connected to a closed-loop flow system consisting of a fluid reservoir, pump, pressure and flow sensors, and tubing. The pump and fluid reservoirs were located in the operator room. The printed models were placed at the isocenter of the MRI scanner and connected to the pumps through 3-meter-long reinforced tubing with an ID of 6 mm routed through the wall access port. Near the model inlet and outlets, the tubing was transitioned to silicone tubes to facilitate sensor placement and flexibility. The setup scheme is illustrated in Figure S2a (Supplementary Material).

The straight and ICA models were perfused with a glycerol-water mixture (40:60 by volume, solution viscosity 3.72 cP (Segur and Oberstar, 1951)) using a pulsatile displacement piston pump (PD-1100, BDC Laboratories), while BA models were perfused with water using a peristaltic pump (Ismatec MCP Standard, Cole-Parmer, USA). The experimental fluids contained 0.3 mmol/l of gadobutrol (Gadovist, Bayer Vital, Germany).

The mean flow rate were set by adjusting the pump speed and were 1) straight vessel (ID 4 mm) – 0, 0.7, 1.4, 3, and 4.7 ml/s, 2) straight vessel (ID 6 mm) – 0, 1, 2.3, 4.8, and 5.4 ml/s, 3) ICA models – 0 and 4.5 ml/s, 4) BA models – 2.4 ml/s. The cardiac cycle period was 0.8 s for ICA/straight models and 1.26 s for BA models. The mean supplied flow rates were selected to match physiological flows observed for the intracranial vessels in vivo (Zarrinkoob et al., 2015). Pressures were not actively regulated, but any excess pressure was relieved manually through the reservoir valve (Figure S2, Supplementary Material). The supplied flow rate was measured at the inlets of all models using a clamp-on ultrasonic sensor (ME8PXL-M12, Transonic System Inc., USA). The representative flow profiles are shown in Figure S3 (Supplementary Material). Pressure values were recorded using pressure transducers (Press-N-000, PendoTech, USA) installed at tubing junctions near the IA model inlets

and outlets via Luer-lock connectors. The representative pressure profiles are shown in Figure S4 (Supplementary Material). The flow and pressure measurements were performed in the MRI operator room (Figure S2 b, Supplementary Material).

Comment 1.43: - The reader finds out about without and with devices (IFD1-10) for the first time here (page 11, line 3)

Answer 1.43: Thank you for your comment. All flow modulation devices (FMDs) used in this study—including IFD1–10—were introduced originally in the dedicated section “FMDs and in vitro deployment” on page 6, lines 3–11. To ensure that readers are familiar with the FMDs used, we have moved additional details from the supplementary material into the main text. Additionally, we have included a new figure that schematically illustrates the different FMD types and the IA models in which they have been deployed, please see new Figure 1.

Comment 1.44: - Unclear what this means: “Flow and pressure curves measured at the model outlets before MRI ”. How were flow and pressure measured then? In the actual patients?

Answer 1.44: Thank you for your comment. We agree that the original wording was unclear and have now clarified this in the revised manuscript. As described in detail in our response to Comment 1.42, the flow and pressure measurements were performed using dedicated sensors *before* the MRI experiments, as part of the in vitro setup. Specifically, flow was measured using clamp-on ultrasonic sensors (ME8PXL-M12, Transonic System Inc., USA), and pressure was recorded with pressure transducers (Press-N-000, PendoTech, USA) placed at the tubing near the model inlets and outlets. These measurements were conducted in vitro and not in actual patients. The flow rates were set to approximate physiological values reported for intracranial vessels in vivo (Zarrinkoob et al., 2015).

Comment 1.45: - A precise description of CFD is lacking. Ideally, the authors would provide the model as an open-source download.

Answer 1.45: Thanks for indicating this lack of information about CFD. A further detailed description was added in the methods section. The simulations were performed using the commercial solver STAR CCM+; thus, an open-source download would not be possible.

Old text: To verify 4D flow MRI results, flow simulations based on computational fluid dynamic (CFD) were performed on BA models without and with devices (IFD1-10). Original CAD models of the IAs were used as the basis for the CFD geometries. Flow and pressure curves measured at the model outlets before MRI were used as boundary conditions (ME8PXL-M12, Transonic System Inc, United States; press-N-000; PendoTech, NJ). Regarding the IFDs, first, the geometry of IFDs was acquired and segmented from μ -CT data. Then, CAD models of the IFDs were registered and deformed to match the shape and location of segmented from μ -CT devices. The median temporal velocities were calculated within the IA sac and parent vessel (like ROIs in MRI results as presented in Figure 1b, with minor manual deviations, Matlab R2022a).

New text: To verify 4D flow MRI results, numerical flow simulations based on computational fluid dynamic (CFD) were performed on BA models without and with devices (IFD1-10). The commercial finite volume solver STARCCM+ was used (StarCCM+2021.3 v16.6, Siemens, Erlangen, Germany) and laminar flow conditions

applied. Fluid properties were set mimicking the experiments. Original CAD models of the BA phantom models were set as the basis for the IA geometries in CFD. The vessel walls were assumed to be rigid, corresponding to the in vitro experiment. Regarding the deformation of the IFDs, first, the deployed geometry of IFDs in the phantom models was acquired and segmented from μ -CT data. Then, CAD models of the IFDs were registered and deformed to match the shape and location of the segmentations from μ -CT devices 34. Flow and pressure curves measured at the model inlet and outlets before MRI were used as boundary conditions (ME8PXL-M12, Transonic System Inc, United States; press-N-000; PendoTech, NJ). Spatial resolution was set to a mesh base size of 0.1 mm with a refinement to 0.02 mm at the IFD struts. Temporal resolution was 0.1 ms and three cardiac cycles were simulated, while only the last cycle was considered within the evaluation. Post processing was performed using the software ANSYS EnSight (v2021.R2; ANSYS, USA) and Matlab (Matlab R2022a). The median temporal velocities were calculated within the IA sac and parent vessel (like ROIs in MRI results as presented in Figure 1b, with minor manual deviations). Here, the IA sac was separated from the parent vessel using a planar neck plane. The location of the planes through the phantom models for the evaluation was determined on the basis of the experiments.

RESULTS

Comment 1.46: - Page 11, line 24: “Different flow rates resulted in a similar time-dependent flow curve featuring one maximum at about 0.2 ms.” The reason for this might be that the same flow curves were used, just with an offset to increase the flow rate. The authors should critically inspect the flow curves of the pumps.

Answer 1.46: Thank you for your observation. We agree that the similarity in time-dependent flow curves is likely due to using the same pulsatile pump setup for all experiments, with adjustments only to the mean flow rate. As such, the overall waveform shape remained largely unchanged, with only amplitude scaling based on the mean flow setting. To clarify this, we have added representative flow curves measured at the model inlet to the Supplementary Material, new Figure S3 and revised the manuscript text accordingly.

Old text: Different flow rates resulted in a similar time-dependent flow curve featuring one maximum at about 0.2 ms (Figure 3a).

New text: All flow conditions resulted in a similar time-dependent flow waveform with a peak at approximately 0.2 s, reflecting the consistent cardiac cycle waveform generated by the pulsatile pump (Figure 4 a). The waveform shape remained largely unchanged, while its amplitude was adjusted by modifying the mean flow rate. Representative inlet flow curves measured with a flow sensor for different flow rates are provided in the Supplementary Material, Figure S3.

Comment 1.47: - Line 25: If it increases, there must be a p-value. Same for decrease

Answer 1.47: The sentence was revised.

Old text: The BB signal decreased with increasing flow rates for both vessel IDs across all sequences, BB_{\perp} , $BB_{\perp} MSDE$, BB_{\parallel} , and $BB_{\parallel} MSDE$ (Figure 3b).

New text: The BB signal decreased with increasing flow rates for both vessel IDs across all sequences (Figure 4 b), resulting in negative correlation coefficient between BB signal and velocity, namely BB_{\perp} ($\rho = -0.87$, $p\text{-value} = 1.08E-03$), $BB_{\perp} MSDE$ (ρ

= -0.95, p-value = 2.93E-05), BB_{\parallel} (rho = -0.92, p-value = 1.32E-04), and $BB_{\parallel} MSDE$ (rho = -0.94, p-value = 6.72E-05).

Comment 1.48: - All results lack p-values, seems like no statistics were performed

Answer 1.48: Thank you for this observation. We have now included p-values in the relevant results sections throughout the manuscript. Additionally, the details of the statistical methods used are described in the “Statistical Analysis” section.

Comment 1.49: - I wish I knew what FD 2, FD 4, etc were

Answer 1.49: Thank you for your comment. All flow modulation devices (FMDs) used in this study—including IFD1–10—were introduced in the dedicated section “*FMDs and in vitro deployment*” on page 6, lines 3–11. To ensure that readers are familiar with the FMDs used, we have moved additional details from the supplementary material into the main text. Additionally, we have included a new figure that schematically illustrates the different FMD types and their deployment in the models.

Comment 1.50: - Page 18, line 11: is this the result of 4D flow or CFD?

Answer 1.50: Thank you for your comment. In the section “Results / Effect of FMDs on BB signal in aneurysm models / Intracranial flow-diverting devices,” we begin with a brief summary of the analysis (page 18, lines 11–14), followed by the presentation of results from 4D flow MRI (page 18, lines 15–19; page 19, lines 1–6). This is followed by results from CFD simulations (page 19, lines 6–11), and finally by results from SE BB MRI (page 19, lines 12–22). To make the source of each velocity result clearer to the reader, we have clearly separated 4D flow MRI and CFD result sections into distinct paragraphs.

Comment 1.51: - Missing CFD visualizations and results

Answer 1.51: Thank you for your comment. Visualizations of CFD results for representative cases were originally already included in Figure 8 of the main text and Figure S5 in the supplementary material. To make the presentation of CFD data clearer and more comprehensive, we have now added velocity map visualizations for each device configuration directly into the main text, please see new Figure 9. Quantitative CFD results are reported in the main text (page 19, lines 7–12), illustrated with a scatter plot (Figure 10), and further detailed in the supplementary tables (Table S6 and S7).

Comment 1.52: - Missing correlation coefficients

Comment 1.53: - Figure 9 is just a scatter plot, how did the authors investigate correlation and came to the conclusion that it was correlating?

Answer 1.52 and Answer 1.53: Thank you for your observation. You are correct — we originally described the relationship between SE BB signal and velocity as a negative association, not a correlation. However, we appreciate your suggestion and agree that this point can be strengthened by quantitative analysis. We have now performed a correlation analysis and added the corresponding correlation coefficients to the manuscript. This helps to objectively characterize the relationship between velocity and SE BB signal, as illustrated in Figure 10.

Old text: Once again, the BB signal was inversely associated with intra-aneurysmal velocity, with lower velocity corresponding to a higher BB signal (Figure 9).

New text: Once again, the BB signal was inversely associated with intra-aneurysmal velocity measured with 4D flow MRI ($\rho = -0.73$, p -value = 0.01), with lower velocity corresponding to a higher BB signal (Figure 10). A negative trend was observed between the BB signal and velocity calculated with CFD, but it was not statistically significant ($\rho = -0.5$, p -value = 0.1). No apparent relationship between BB signal and velocity in parental vessel measured by 4D flow MRI ($\rho = 0.06$, p -value = 0.83) and calculated by CFD ($\rho = 0.3$, p -value = 0.29) was found.

Comment 1.54: - Page 19, line 9 ff: I do not understand how the authors determined a 3D velocity vector from BB MRI. Though I am happy that there are finally p -values.

Answer 1.54: Thank you for your comment. We believe you are referring to the results presented on page 21, lines 9–17. To clarify, we did not compute a 3D velocity vector from SE BB MRI data. Rather, as stated in the manuscript, we calculated the median SE BB signal intensity within the aneurysm sac and parental vessel and compared it to the median velocity magnitude obtained from 4D flow MRI in the corresponding locations. No directional or vector-based velocity information was derived from the SE BB images. If this was not the source of your concern, we would be happy to provide further clarification.

DISCUSSION

Comment 1.55: - The discussion is poorly written and does not follow the typical structure. First, the methods and results should be summarized and compared systematically to the literature. The subheadings are unusual. The authors immediately jump into discussing details.

Answer 1.55: Thank you for this valuable feedback. We have revised the Discussion section to follow a more conventional structure. We have added a summary of the study goals and key results at the beginning of the section, as well as an outline of the discussion structure in our manuscript, to set the reader's expectations.

New text Discussion: This study aimed to explore whether the signal behavior of SE BB MRI in regions of slow flow—typically considered as an undesired property—can instead serve as a diagnostic feature. Specifically, we assessed whether this signal can provide complementary information to 4D flow MRI in the presence of severe metal-induced artifacts after aneurysm treatment.

To address this, we conducted *in vitro* experiments using simplified straight vessel phantoms and patient-derived aneurysm models from ICA and BA. These models were treated with FMDs of various sizes, including five flow-diverting stents and ten intrasaccular devices. We evaluated (1) how the SE BB MRI signal intensity changes with varying flow rates, and (2) the influence of acquisition parameters on this relationship. Our results demonstrate a consistent association between the SE BB signal and intra-aneurysmal flow conditions, suggesting the potential for using SE BB imaging as an adjunct to flow quantification together with 4D flow MRI. In the discussion, we summarize the key results related to this study and compare them to the existing literature, as well as discuss the validity of the *in vitro* approach and the overall limitations of our work.

Comment 1.56: - There is bright blood and black blood methodology in MRA. The authors should avoid talking about “bright BB signal”.

Answer 1.56: Thank you for your comment. The phrasing was changed throughout the manuscript.

Comment 1.57: - The sequence is an FSE sequence and thus does not have a “spin-echo nature.” Please reformulate the entire sentence, it has poor structure and is complicated. (page 22, line 9).

Answer 1.57: The sentence was reformulated.

Old text: In this study, BB MRI produced fewer due to its spin-echo nature(Henningsson et al., n.d.) metal artifacts on BB MRI compared to 4D flow and TOF MRI, especially when imaging IFDs (here, Contour Neurovascular System).

New text: In this study, SE BB MRI exhibited fewer metal artifacts than 4D flow and TOF MRI, especially when imaging IFDs (here, using the Contour Neurovascular System).

Comment 1.58: - I do not find the results very astonishing and would not say that this study brought new insights about using BB MRI for angiography. The strength of this study is systematic quantitative analysis. Unfortunately, there are missing details on many of the steps, many things remain unclear, and nothing can be reproduced. However, the conclusions that the authors draw from this in vitro study are not worth this publication. The discussion needs to be reworked and needs to have a different focus. I wish the authors would have shown how to determine velocities from the BB MRI data or anything new.

Answer: 1.58: Thank you for your thoughtful comment. We agree that our study does not introduce fundamentally new insights into angiography itself. Instead, our key contribution is to highlight that the increased SE BB MRI signal observed post-aneurysm treatment can potentially serve as an indicator of flow modulation—specifically, slow flow within the aneurysm sac. To the best of our knowledge, this perspective has not been previously explored, as the non-suppressed fluid signal was generally regarded as an unwanted artifact. Our work encourages viewing this signal behavior as a useful feature rather than a limitation.

Regarding reproducibility, many of the requested details were originally provided in the supplementary materials, but based on your feedback, we have now moved these details into the main text and added further clarifications to enhance transparency.

Regarding the discussion and your suggestion about velocity determination from BB MRI data, we want to clarify that our study was not intended to replace 4D flow MRI nor to derive velocity vectors from BB MRI. Instead, we demonstrate a consistent association between increased SE BB MRI signal and reduced velocity after the deployment of flow modulation devices within the aneurysm. This suggests the potential for SE BB MRI to serve as a complementary tool for indicating reduced flow, especially in situations where other imaging methods suffer from severe metal artifacts.

We appreciate your insightful comments, which have helped us improve the clarity and focus of our manuscript.

Comment 1.59: - Page 22, line 14 ff: 4D flow MRI measures time-resolved 3D velocities. BB TSE cannot measure velocities. I can deal better with artifacts; we all know that. But why is it better than 4D flow? The information is like always with MRI complementary.

Answer 1.59: Thank you for your comment and the opportunity to clarify this point. At page 22, line 14, we are comparing BB MRI primarily to other angiographic modalities such as DSA and CTA—not to 4D flow MRI. In our clinical setting, MRI in general is

still not commonly used for follow-up in patients with intracranial aneurysms treated with intrasaccular devices, largely due to metal artifacts and consequent difficulties in assessing the aneurysm dome. However, recent studies have demonstrated that SE BB MRI can serve as a viable alternative to DSA, and for evaluating aneurysm remnants(Quan et al., 2023) and parent vessel patency(Gomyo et al., 2022; Quan et al., 2023). It has also shown advantages over TOF MRI when visualizing vasculature in the presence of metal artifacts.

In the following paragraph (page 22, lines 15–20), we discuss 4D flow MRI and its limitations, particularly in small aneurysms treated with intrasaccular devices. These limitations include signal voids caused by metal artifacts that can exceed the size of the aneurysm itself (as seen in our BA models 1 and 2). If our wording implied that BB MRI should replace 4D flow MRI, that was not our intention, and we apologize for the confusion.

Our goal is to promote the broader use of MRI-based follow-up in this patient population. Ideally, this would include 4D flow MRI whenever feasible. However, in cases where 4D flow is compromised by artifacts, SE BB MRI may provide useful complementary information by indirectly indicating regions of slow flow. We have revised the paragraph to better reflect this intention and avoid any suggestion that BB MRI could replace 4D flow MRI.

Old text: 4D flow MRI's ability to assess aneurysmal flow was limited by metal artifacts and small aneurysm sizes, especially in the BA models 1 and 2, where aneurysm sacs were smaller than the signal void caused by the artifacts. This challenge highlights the potential advantage of BB MRI, which allowed visualization of nearly the entire aneurysm sac, even in small aneurysms (3.5–6.9 mm in height), a critical feature since small aneurysms make up 20-60% of all detected aneurysms(Júnior et al., 2019) and can also be prone to rupture(Lee et al., 2015). To overcome artifact-induced visibility issues, we conducted numerical flow simulations to supplement MR flow measurements and compare the BB signal with intra-aneurysmal flow(Korte et al., 2023). The calculated reduced flow was also associated with an increased BB signal.

New text: As shown in this study, BB MRI allowed visualization of nearly the entire aneurysm sac, even in small aneurysms (3.5–6.9 mm in height). This is clinically relevant, as small aneurysms represent 20–60% of all detected aneurysms(Júnior et al., 2019) and still carry a risk of rupture(Lee et al., 2015). Beyond anatomical visualization, our results indicate that BB MRI may help highlight regions of slow intra-aneurysmal flow following treatment, potentially supporting the assessment of treatment efficacy. Flow measurements were obtained using 4D flow MRI and, where metal artifacts impaired visibility—particularly in BA models 1–4 treated with intrasaccular devices—were supplemented by numerical simulations. This was especially necessary in BA models 1 and 2, where the artifact size on 4D flow MRI exceeded the aneurysm dome. While SE BB MRI does not directly quantify velocity, it may serve as a complementary tool when 4D flow MRI is compromised by severe metal artifacts.

Comment 1.60: - I am missing an outlook on how the authors want to use the BB TSE sequence in the future to determine velocities (without acquiring 2D PC or 4D flow MRI). I do understand the increased signal could be a surrogate marker, but the authors did not show how they want to provide reliable and reproducible information on flow reduction quantitatively.

Answer 1.60: Thank you for this thoughtful comment. We would like to clarify that we do not propose using the SE BB MRI signal as a direct replacement for velocity measurements obtained via 4D flow MRI. We strongly support the use of 4D flow MRI, as it is a powerful and non-invasive method for measuring flow dynamics. However, in practice, 4D flow MRI can be significantly compromised by metal artifacts, especially in patients treated with intrasaccular devices. In such cases, flow information may be entirely lost or unreliable.

As you noted, our intention is to explore the potential of the BB signal as a surrogate marker for reduced intra-aneurysmal flow—particularly when other methods fail due to artifacts. We have now made this point clearer in the revised Discussion section and explicitly state that BB MRI might complement 4D flow MRI.

We agree that more work is needed to use BB MRI in the clinical routine for assessment of flow reduction. While this was beyond the scope of the current study, future work will focus on testing the hypothesis in vivo, standardizing signal thresholds, calibrating BB signal against known flow regimes, and validating these findings across multiple centers and imaging platforms. We have added a brief outlook to the Discussion to highlight this as a key direction for future research.

New text Limitations and outlook: BB MRI signal does not provide velocity information, unlike 4D flow MRI, and can only be used as a surrogate marker. Future work will focus on testing the hypothesis in vivo, standardizing signal thresholds, calibrating BB signal against known flow regimes, and validating these findings across multiple centers and imaging platforms.

Comment 1.61: - The authors highly speculate that BB TSE can be used for treatment outcome. But they do not formulate it carefully enough and make claims on what they do not investigate.

Answer 1.61: Thank you for this important observation. We agree that some of our original statements may have been too strong and could be interpreted as overclaiming. We have carefully revised the manuscript to clarify that our findings suggest the potential of SE BB MRI signal as a surrogate marker for reduced intra-aneurysmal flow, rather than as a direct indicator of treatment outcome.

FIGURES

Comment 1.62: Please add a figure of the experimental setup.

Answer 1.62: The figure was added to the supplementary material, Figure S2.

SUPPLEMENTAL MATERIAL

Comment 1.63: missing information on pump and flow curve

Answer 1.63: The information was added, see new Figure S3

Comment 1.64: A sketch of the experimental setup would be desirable for reproducibility reasons

Answer 1.64: The figure was added, please see the Figure S2.

Comment 1.65: I wish I had some knowledge from the supplementary material before reading the manuscript, especially regarding the different FDM phantoms. I think a lot of information, especially when using abbreviations such as FD 1 or FD 1-2, needs to be mentioned in the manuscript itself.

Answer 1.65: Thank you for your comment. Most of the information from the supporting material has now been moved to the main text.

Comment 1.66: Also here, no statistical analysis results are presented, and no p-values were provided.

Answer 1.66: The p-values were added.

Comment 1.67: Add a figure of the 20 different aneurysm phantoms

Answer 1.67: A figure representing five different aneurysm phantoms and 15 flow modulation devices that have been placed in 15 replicas of these phantoms has been added to the main text, please see Figure 1.

Reviewer #2 (Remarks to the Author):

The authors present an interesting work looking at the use of black blood MRI with and without MSDE preparation to look at the residual flow in an aneurysm after treatment with an endovascular device. Overall, the manuscript is well written and is of interest. However, there are manifold details that I believe should be addressed before publication:

Answer 0: Thank you for the positive evaluation of our manuscript. To the best of our knowledge, we have incorporated your suggestions and addressed the comments.

MAJOR:

Comment 2.1: A large amount of effort was taken to describe how the vasculature was derived from a specific patient, which is important details, but the authors then make drastic edits to the morphology of the aneurysm. Would it not have been better to take a population averaged vasculature, as once you begin to edit the aneurysm morphology its not a patient specific model anymore.

Answer 2.1: Thank you for this thoughtful comment. You are correct in noting that we made certain modifications to the patient-derived geometries, such as reducing the number of outlets or adjusting the aneurysm sac to allow for proper placement of various devices. These adjustments were necessary to accommodate specific device requirements and to ensure technical feasibility during in vitro experiments.

While we agree that such modifications reduce the “patient-specific” nature of the models, we chose these geometries based on actual cases in which the selected devices (flow-diverting stents and intrasaccular devices) were used clinically. Unfortunately, to our knowledge, there are currently no publicly available population-averaged vascular models that include intracranial aneurysms suitable for device-specific analysis.

To address your point about generalizability, we have now clarified this limitation in the manuscript:

Old text: This study presents in vitro findings from a single center conducted on a limited number of rigid IA models and straight vessels (22 independent flow experiments in total).

New text: This study presents in vitro findings from a single center conducted on a limited number of patient-derived IA models and straight vessels, limiting the generalizability of the results

Comment 2.2: The use of a resin with high stiffness was unlikely to be an appropriate choice for modeling vasculature, why wasn't an elastic material such as silicon and PVA used?

Answer 2.2: Thank you for this important comment. We fully agree that biological vessels are elastic and that ideally, vascular phantoms would replicate this mechanical behavior. Materials like silicone and PVA hydrogel can indeed offer greater compliance; however, they typically require multi-step fabrication workflows that pose substantial challenges when modeling the small and intricate branches of the intracranial vasculature.

The common silicone-based approach involves 3D printing a water-soluble core, casting silicone around it, and subsequently dissolving the inner core (Falk et al., 2019). While effective for larger vessels, this method becomes unreliable at the smaller scales required for intracranial aneurysms due to limited printing resolution and difficulties in uniformly forming thin, flexible vessel walls. Molding would result in a solid silicone block, eliminating the benefit of wall flexibility. On the other hand, manual techniques such as hand-painting thin silicone layers over the core introduce variability that limits reproducibility between models.

was essential. Rigid models—fabricated using stereolithography with a high spatial resolution (up to 0.025 mm)—provided the required level of precision and repeatability.

To emphasize this point, we have added a new text to the limitation section:

New text /Limitations: All vascular models featured rigid walls, which may introduce deviations from the hemodynamic conditions observed in vivo with elastic vessel walls. However, this simplification was necessary to ensure reproducibility of the fabrication of multiple copies of the same IA model geometry. While it may limit physiological realism, it is likely to have a smaller impact on assessing relative treatment efficiency.

Comment 2.3: It appears that constant flow was used, not pulsatile. I believe that pulsatile flow would have a large impact on the degree of blood signal nulling. The authors either need to justify this decision, perform experiments showing it has no effect, or I would suggest repeating experiments with physiological flow conditions.

Answer 2.3: Thank you for pointing this out. We agree that flow conditions—especially pulsatility—can significantly influence signal suppression in SE BB MRI. In our study, time-dependent (pulsatile) flow was indeed used, although the flow systems varied slightly across models:

1. Straight vessels and ICA aneurysm models were supplied using a pulsatile piston pump (PD-1100, BDC Laboratories, USA), which provided physiologically realistic flow waveforms.
2. BA aneurysm models were driven by a peristaltic pump (Ismatec MCP Standard, Cole-Parmer, USA), which also produces a time-varying (though less physiologically accurate) flow pattern.

In both cases, the mean flow rates were adjusted to reflect physiological values based on published in vivo data (Zarrinkoob et al., 2015). We acknowledge that these important experimental details were not clearly described in the original manuscript. To address this, we have now substantially revised the “Methods / Experimental Flow Setup” section to clarify the flow conditions used in each model type. In addition, we

have created a new figure for the Supporting Materials that illustrates representative pulsatile flow patterns for each model type, please see new Figure S3

Comment 2.4: Although the authors are correct that the BB images have less metal artifacts, it is not free of them. For any quantitative results they need to be clear that areas of artifact were not considered, and this should be carried out between all models. It would not be consistent to look at different ROIs between devices and make any claim about flow.

Answer 2.4: Thank you for this valuable comment. We fully agree that regions affected by metal artifacts must be excluded from quantitative analysis, and that ROIs should be consistent across models for valid comparisons.

To ensure consistency of the ROIs in BB MRI, we first defined 3D ROIs based on the inner lumen of the vasculature using CAD models prior to 3D printing. These ROIs were then aligned to each experimental dataset to maintain consistent positioning across all models. To avoid including areas impacted by metal artifacts, we identified the dataset with the most severe artifacts and subsequently reduced the standard ROIs to ensure that these regions were excluded in all models. Fortunately, metal artifacts were minimal for flow-diverting stents and, in the case of intrasaccular devices, artifacts were largely limited to the detachment zone and did not affect the aneurysm sac region in BB MRI.

For the 4D flow MRI analysis, consistency was ensured using MRI-visible markers integrated into each model during fabrication. These markers were positioned at the mid-height of the aneurysm sac and served as reference points for placing 2D evaluation planes. In ICA models, these markers were unaffected by artifacts, allowing for consistent placement of one central and two equidistant planes above and below it. However, in BA models 1 and 2, metal artifacts extended beyond the marker location, making this central plane unusable. In those cases, flow quantification was only performed in artifact-free regions above the marker.

To address this limitation, and to ensure reproducibility in flow quantification, we additionally performed numerical simulations for all BA models. These simulations are free from metal artifacts and enabled consistent analysis across devices. We have clarified these steps in the revised Methods section and added an illustration showing the MRI-visible marker locations on the model design prior to printing, please see new Figure S1.

Old text Methods/MRI in vitro/MRI processing/2D PC and 4D flow MRI: In 4D flow MRI datasets, five equidistant, cross-sectional 2D planes were placed in the parent vessel and aneurysm sac (Figure 1a).

New text Methods/MRI in vitro/MRI processing/2D PC and 4D flow MRI: In 4D flow MRI datasets, five equidistant, cross-sectional 2D planes were placed in the parent vessel and aneurysm sac (Figure 2 a). MRI-visible markers integrated into the models (Figure S1, Supplementary Material) were used to ensure consistent placement of evaluation planes. The central plane was aligned with the marker, with two additional planes placed equidistantly above and below. In BA models 1 and 2, the marker was obscured by metal artifacts; thus, only artifact-free regions above the marker were used for analysis.

Old text Methods/MRI in vitro/MRI processing/BB MRI: BB MRI signal was quantified in volumetric ROIs in the vessel and aneurysm (Figure 1b). First, 3D ROIs

were defined on a computer-aided design (CAD) of straight vessel and aneurysm models.

New text Methods/Methods/MRI in vitro/MRI processing/BB MRI: BB MRI signal was quantified in volumetric ROIs in the vessel and aneurysm sac (Figure 2 b). First, 3D ROIs were defined on a computer-aided design (CAD) of straight vessel and aneurysm models. For each IA model a dataset with the strongest appearance of metal artifacts has been determined and aneurysm 3D ROI have been adjusted to ensure that The size and location of these ROIs were selected to ensure consistent placement across all models and to avoid inclusion of areas affected by metal artifacts.

Comment 2.5: For the statistical section, it is unlikely to be consistent with the assumptions of the test that the control (no FMD) can be used multiple times for comparison. I would suggest a different statistical approach, or perform multiple control experiments (before/after implant) in the same model to allow validity to the methods used.

Answer 2.5: Thank you for this important comment. We have carefully reconsidered our statistical analysis in light of your concern. Rather than repeatedly using the same untreated control values, we now calculate a normalized treatment effect (Δ) for each model, defined as the relative change in velocity and BB signal between the untreated and treated conditions. These Δ values are then statistically compared to a zero-effect distribution using a two-sided Wilcoxon signed-rank test. This approach accounts for our limited number of untreated models while still allowing us to assess the treatment effect in a consistent manner.

We have updated the "Statistical Analysis" section of the manuscript to reflect this new methodology. In addition, we have added a note to the "Limitations and Outlook" section to acknowledge the limited sample size of untreated models, which prevents a full characterization of baseline variability.

Old text Materials and Methods / Statistical analysis: The median BB signal in the presence of the FMDs ($n = 16$; 6 IA models with FDs and 10 IA models with IFDs) was compared to the BB signal in the empty model ($n = 6$; 2 empty ICA models and 4 empty BA models) using the two-sided Wilcoxon test with significance level of p -value < 0.01 . As there was only one empty model per different aneurysm sac, the median values were used multiple times to create pairs of BB signals and velocities with and without an FMD for each treated case ($n = 16$).

New text Materials and Methods / Statistical analysis: The treatment effect (Δ) of FMDs on BB signal and velocity was calculated for each IA model as follows, $\Delta BB = \frac{BB_{wo\ FMD} - BB_{with\ FMD}}{BB_{wo\ FMD}}$ and $\Delta velocity = \frac{velocity_{wo\ FMD} - velocity_{with\ FMD}}{velocity_{wo\ FMD}}$, respectively. These normalized treatment effects (ΔBB and $\Delta velocity$) were then tested against a zero-effect hypothesis using a two-sided Wilcoxon signed-rank test. Statistical significance was defined as $p < 0.05$.

New text Limitations and Outlook: Due to the limited number of untreated models ($n = 5$), it was not possible to assess the variability of baseline BB signal across different IA anatomies. This limitation should be considered when interpreting the treatment effect results.

Comment 2.6: In the discussion: its stated that there was a linear relationship between the BB signal and the flow in the straight model, then the conclusion that this could provide quantitative information about aneurysm occlusion. This is problematic for two

reasons: First, no statistics or quantification is performed to prove the linear relationship, so this would have to be done. Second and more importantly, this relationship does not appear to be true in any of the aneurysm models (only straight), and thus the conclusion is likely refuted by the data.

Answer 2.6: Thank you for this valuable comment. We agree that the original statement in the discussion was speculative and insufficiently supported. To address this, we have now performed a correlation analysis between the mean velocity and BB signal in the straight vessel model and also in the aneurysm models, confirming a significant negative correlation, and included this analysis in the results section. Moreover, we have removed the statement suggesting a general linear relationship and revised the discussion to clarify that while the BB signal may serve as a surrogate qualitative marker for relative flow reduction in specific contexts, further in vivo studies are necessary to validate its use for assessment of aneurysm occlusion outcomes.

Old text: The decreased velocity observed with all FMDs corresponded to an increased BB signal within the aneurysm sac, suggesting that the BB signal may serve as a surrogate marker for flow reduction. Furthermore, we found a linear relationship between the BB_{\perp} signal and velocity in a straight vessel model, indicating that BB MRI could provide a quantitative measure of aneurysm occlusion outcomes, like DSA's washout time (Sadasivan et al., 2019) or 4D flow MRI's and CFD's velocity reduction (Brina et al., 2019; Sarrami-Foroushani et al., 2021).

New text: The decreased velocity observed with all FMDs has been associated with increased BB signal within the aneurysm sac, suggesting that the BB signal may serve as a surrogate marker for flow reduction. However, in vivo studies are needed to demonstrate whether BB MRI could provide a measure of aneurysm occlusion outcomes, like DSA's washout time (Sadasivan et al., 2019) or 4D flow MRI's and CFD's velocity reduction (Brina et al., 2019; Sarrami-Foroushani et al., 2021).

MINOR:

Comment 2.7: At the beginning of the introduction, aneurysms do not reperfuse, the correct term is recanalizations.

Answer 2.7: corrected.

Comment 2.8: An explanation of why water was used in one model and glycerin-water was used in a different model is needed. Viscosity could have large effects around edges.

Answer 2.8: Thank you for this important comment. You are absolutely right that viscosity can influence intra-aneurysmal flow behavior and, potentially, signal suppression in BB MRI.

In our study, different fluids were used for practical and methodological reasons:

1. Water was used in the BA models because these models were originally used for a series of DSA experiments evaluating contrast washout times after intrasaccular device placement (Pravdivtseva et al., 2023). DSA procedures involve catheter navigation and repeated fluid changes, and using glycerol-water mixtures in this context would pose significant issues with leakage, contamination, and cleanup due to the higher viscosity and stickiness of glycerol. To ensure consistency between the BB MRI and DSA-derived flow

behavior, we continued using water during BB MRI experiments on the BA models

2. Glycerol-water mixture was used in ICA models and straight vessel models to better match the viscosity of human blood, where catheter access was not required, and fluid handling was more controlled.

While initially we viewed the inconsistency in working fluids as a limitation, it ultimately allowed us to observe that BB signal increase following flow modulation occurred across both fluid types, suggesting that the observed effect is robust and not highly dependent on precise viscosity. Additionally, blood viscosity naturally varies between patients, further supporting the generalizability of this finding. We have now updated the Methods and Discussion sections to clarify these decisions and highlight the consistency with our previous work

Old text Methods: A blood-mimicking fluid, water-glycerin mixture for ICA models and pure water for BA models, was pumped with pulsatile (PD-1100, BDC Laboratories) and peristaltic (Ismatec MCP Standard, Cole Parmer, United States) pumps, respectively.

New text Methods: The straight and ICA models were perfused with a glycerol-water mixture (40:60 by volume, solution viscosity 3.72 cP (Segur and Oberstar, 1951)) using a pulsatile displacement piston pump (PD-1100, BDC Laboratories). The BA models were perfused with pure water using a peristaltic pump (Ismatec MCP Standard, Cole-Parmer, USA), consistent with our previous work evaluating contrast washout in the same models using DSA (Pravdivtseva et al., 2023).

New text Discussion: In addition, the increased BB signal observed in BA models after placement of IFDs corresponded to the increased contrast washout times measured in our prior study (Pravdivtseva et al., 2023), reinforcing the potential of BB MRI to qualitatively reflect the intra-aneurysmal flow reduction.

Reviewer #3 (Remarks to the Author):

This manuscript focuses on addressing imaging artifacts encountered during MRI scans of intracranial aneurysms. The authors assessed the performance of 4D flow MRI and black blood (BB) MRI techniques in the presence of metal implants, such as stents. Furthermore, they proposed the potential use of BB MRI as a biomarker for predicting treatment success based on the reduction in blood flow following treatment. A reduction in blood flow of 35% or more with a flow-modulation device was considered indicative of a successful treatment outcome.

The manuscript is well-written, and the figures are of high quality. It explores an intriguing interdisciplinary topic in quantitative MRI and methods for evaluating intracranial aneurysm treatments. However, there are several issues and questions that the authors need to address before the manuscript can be considered for publication.

Answer 0: Thank you for your positive feedback. To the best of our knowledge, we have incorporated the suggestions and addressed the comments. A detailed list of references used in the reply is provided at the end of this document.

Comment 3.1a: “However, WOT is influenced by factors such as contrast injection technique, which is often performed manually and may vary between pre- and post-FMD deployment.” Please provide a reference to support this statement.

Answer 3.1a: Thank you for the comment. We have not found a paper that directly states that most radiological centers use a manual injection protocol, although in our center it is the case. Therefore, we have rephrased the sentence and provided the necessary supporting references.

Old text: However, WOT is influenced by factors such as contrast injection technique, which is often performed manually and may vary between pre- and post-FMD deployment.

New text: However, the accuracy of WOT can be dependent on the contrast injection technique. In cases of contrast injection, “spike-like” artifacts have been described (Sadasivan et al., 2019), while a large variation in WOT—up to 50%—was observed in an in vitro study with manual injection (Pravdivtseva et al., 2023).

Comment 3.1b: DSA is a well-established technique that provides 2D blood flow patterns. How does the use of 3D flow patterns contribute to clinical outcomes or decision-making?

Answer 3.1b: We completely agree that the contrast stasis observed on DSA provides unique and immediate information on the device's efficiency. If there is no stasis, then another treatment strategy will be selected while the patient is still on the operation table, e.g., coils can be introduced inside the aneurysm in addition to the flow-diverter to promote flow reduction. However, stasis is not a guarantee that the aneurysm will be occluded; aneurysm growth and even delayed aneurysm rupture have been reported. In addition, in some locations, the aneurysm will overlap with other branches or venous flow, which might prevent accurate assessment (Sadasivan et al., 2019). Moreover, flow changes assessed via optical evaluation with DSA of blood flow patterns did not reveal any links between occlusion status at follow-up (Dang Luu et al., 2024).

We believe that the evaluation of 3D flow patterns can provide a more accurate prognosis on device performance and guide clinical decision-making regarding how often a person should be followed up. 3D flow evaluation does not depend on the chosen 2D projection, nor will overlapping with other arteries play a role. Moreover, 3D imaging provides a more detailed representation of complex vascular structures and flow patterns, enabling better assessment of flow dynamics within aneurysms. This comprehensive view can reveal intricate flow patterns that might be overlooked in 2D imaging, e.g., whether the flow has a jet and small impingement zone or is rather diffuse. Likely, both of these situations will result in DSA stasis. However, the concentrated jet will likely result in high stress on the aneurysm wall.

To clarify this line of argument in the text, we have added the following to the introduction:

Old text: Additionally, DSA provides only 2D projections, limiting the ability to assess the 3D flow patterns inside the aneurysm fully.

New text: Additionally, DSA provides only 2D projections, introducing an additional source of errors. In some cases, it is not possible to obtain a projection of the aneurysm without overlapping with other arteries or veins (Sadasivan et al., 2019). Moreover, there is no clear correlation between flow reduction detected with DSA-based optical

flow methods and aneurysm occlusion (Dang Luu et al., 2024). Assessing 3D flow patterns or 3D stasis might solve this problem by providing comprehensive information on the flow patterns inside the aneurysm and increasing diagnostic accuracy.

Comment 3.2: The authors utilized a patient-derived model and a pulsatile pump to simulate blood flow. Was the elasticity of the vessel considered in the in vitro model and simulation model?

Answer 3.2: Thank you for pointing this out. No, the elasticity of the vessel was not considered in the in vitro experiment or in the simulations. We agree that elasticity (Xu et al., 2016) as well as wall thickness (Voß et al., 2018) have a substantial effect on hemodynamics in intracranial vessels.

Rigid-wall models were chosen because they can be reliably fabricated via direct 3D printing, which ensures high reproducibility. In contrast, producing elastic vascular models typically requires a more complex workflow—such as printing a water-soluble inner core, casting silicone around it, and then dissolving the core (Falk et al., 2019). While feasible for larger vessels, this method becomes increasingly unreliable at the small scale and complex geometries of intracranial aneurysms. Limited print resolution and challenges in achieving uniform wall thickness make it difficult to produce consistent and functional elastic models. Additionally, alternative manual approaches (e.g., hand-painting silicone) introduce variability, compromising model reproducibility.

w-modulation devices. To attribute observed changes specifically to device presence rather than variation in wall mechanics or geometry, model consistency was essential. Rigid models—fabricated using stereolithography with a high spatial resolution (up to 0.025 mm)—provided the required level of precision and repeatability.

To emphasize this point, we have added a new text to the limitation section:

New text / Methods / In vitro vascular models: All models were 3D printed using stereolithography and possess rigid walls (Clear Photoreactive Resin composed of a methacrylated oligomer, methacrylated monomer and photoinitiator; From 3B, Formlabs, USA).

The vessel walls were assumed to be rigid, corresponding to the in vitro experiment.

New text / Methods / Limitations and outlook: All vascular models featured rigid walls, which may introduce deviations from the hemodynamic conditions observed in vivo with elastic vessel walls. However, this simplification was necessary to ensure reproducibility of the fabrication of multiple copies of the same IA model geometry. While it may limit physiological realism, it is likely to have a smaller impact on assessing relative treatment efficiency.

Comment 3.3: In the sub-section “Intracranial aneurysm flow models” of the “Materials and Methods”, the maximum dimension values are mentioned. Does this refer to the length or the diameter?

Answer 3.3: Thank you for your comment. We have clarified the dimensions to which this refers, as well as created a figure indicating the measured dimensions.

Old text: The diameters of the resulting artificial aneurysms (height, neck, and dome width) were 3.5×2.7×3.2 mm (BA model 1), 6.9×2.8×3.3 mm (BA model 2), 8.4×6.7×8.4 mm (BA model 3), and 16.4×9.2×10.2 mm (BA model 4), respectively. The corresponding ICA aneurysm model dimensions were 14.1×7.7×10.9 mm.

New text: The diameters of the resulting artificial aneurysms (height length, neck, and dome width diameters) were 3.5×2.7×3.2 mm (BA model 1), 6.9×2.8×3.3 mm (BA model 2), 8.4×6.7×8.4 mm (BA model 3), and 16.4×9.2×10.2 mm (BA model 4), respectively. The corresponding ICA aneurysm model dimensions (height length, neck, and dome width diameters) were 14.1×7.7×10.9 mm.

Comment 3.4: Line 178: Why was the temporal median used for quantification?

Answer 3.4: The cardiac flow profile is not normally distributed; therefore, we decided to use the median value, instead of mean.

Old text: Then, temporal median velocity was calculated.

New text: Then, temporal median velocity was calculated due to the skewness of the velocity data.

Comment 3.5: Line 227: Why is a normalization of the BB signal necessary? Are these signals subject to variation under different conditions?

Answer 3.5: Yes, the BB signal varies depending on the MRI parameters. For example, the introduction of MSDE reduces the SNR of the entire image (Wang et al., 2007). To enable comparisons between BB with MSDE and without, we normalized the BB signal in the aneurysm and in the vessel by the signal of stationary tissues (agarose gel). We have clarified this in the methods section:

Old text: The BB signal in 3D ROIs was normalized by the mean values of the BB signal in agarose gel, and the normalized BB signal was reported in arbitrary units (a.u.).

New text: To enable the comparison between BB signals derived from different BB sequences without dependence on SNR, the signal in 3D ROIs was normalized by the mean values of the BB signal in agarose gel, and the normalized BB signal was reported in arbitrary units (a.u.).

Comment 3.6: The observation of higher BB signals with FDM is intriguing. It would be valuable to include a failure model where FDM does not result in a significant reduction in blood flow.

Answer 3.6: Thank you for this insightful suggestion. We agree that including a failure case is important to demonstrate the specificity of the BB MRI signal response to actual flow modulation.

Indeed, we observed such a case in the ICA model implanted with the FD2 device. FD2 had a relatively large cell size and resulted in minimal reduction of intra-aneurysmal flow compared to the no-device condition. Correspondingly, the BB MRI signal within the aneurysm sac remained low—very similar to the untreated case—highlighting the failure of this device to achieve effective flow modulation.

To make this more evident to the reader, we have revised the results section to specifically draw attention to FD2 as a non-responding case, thus reinforcing the relationship between BB signal changes and successful flow reduction.

Old text / Results: All flow-diverting stents (FD1-5) reduced the velocity within the aneurysm sac, while the velocity in the parent vessel remained similar across the models (Figure 6b). FD1 reduced velocity most effectively ($w.=0.6$ vs. $w_o.=12.2$ cm/s), while FD2 reduced it the least ($w.=11.2$ vs. $w_o.=12.2$ cm/s) compared to the ICA model

without a device. In the presence of FD1 and FD3-5, the BB signal in the aneurysm sac was elevated for both, BB_{\perp} and $BB_{\perp} MSDE$ sequences (Figure 6c-d, Figure 7a-b).

New text / Results: Flow-diverting stents FD1 and FD3-5 substantially reduced the velocity within the aneurysm sac, while the velocity in the parent vessel remained similar across the models (Figure 7 b). FD1 was the most effective (with [w.] = 0.6 vs. without [wo.] = 12.2 cm/s), whereas FD2 showed minimal flow reduction (w. = 11.2 vs. wo. = 12.2 cm/s), likely due to its large cell size and low metal coverage. In line with these observations, the BB MRI signal in the aneurysm sac increased in the presence of FD1 and FD3–FD5 for BB_{\perp} , while there was no change in the presence of FD2.

References:

- Berg, P., Saalfeld, S., Janiga, G., Brina, O., Cancelliere, N.M., Machi, P., Pereira, V.M., 2018. Virtual stenting of intracranial aneurysms: A pilot study for the prediction of treatment success based on hemodynamic simulations. *Int J Artif Organs* 41, 698–705. <https://doi.org/10.1177/0391398818775521>
- Bissell, M.M., Raimondi, F., Ait Ali, L., Allen, B.D., Barker, A.J., Bolger, A., Burris, N., Carhäll, C.-J., Collins, J.D., Ebbers, T., Francois, C.J., Frydrychowicz, A., Garg, P., Geiger, J., Ha, H., Hennemuth, A., Hope, M.D., Hsiao, A., Johnson, K., Kozerke, S., Ma, L.E., Markl, M., Martins, D., Messina, M., Oechtering, T.H., van Ooij, P., Rigsby, C., Rodriguez-Palomares, J., Roest, A.A.W., Roldán-Alzate, A., Schnell, S., Sotelo, J., Stuber, M., Syed, A.B., Töger, J., van der Geest, R., Westenberg, J., Zhong, L., Zhong, Y., Wieben, O., Dyverfeldt, P., 2023. 4D Flow cardiovascular magnetic resonance consensus statement: 2023 update. *J Cardiovasc Magn Reson* 25, 40. <https://doi.org/10.1186/s12968-023-00942-z>
- Brina, O., Bouillot, P., Reymond, P., Luthman, A.S., Santarosa, C., Fahrat, M., Lovblad, K.O., Machi, P., Delattre, B.M.A., Pereira, V.M., Vargas, M.I., 2019. How Flow Reduction Influences the Intracranial Aneurysm Occlusion: A Prospective 4D Phase-Contrast MRI Study. *American Journal of Neuroradiology* 40, 2117–2123. <https://doi.org/10.3174/ajnr.A6312>
- Cebral, J.R., Mut, F., Raschi, M., Hodis, S., Ding, Y.-H., Erickson, B.J., Kadirvel, R., Kallmes, D.F., 2014. Analysis of Hemodynamics and Aneurysm Occlusion after Flow Diverting Treatment in Rabbit Models. *Int J Numer Method Biomed Eng* 30, 988–999. <https://doi.org/10.1002/cnm.2640>
- Chalouhi, N., Tjoumakaris, S., Starke, R.M., Gonzalez, L.F., Randazzo, C., Hasan, D., McMahon, J.F., Singhal, S., Moukarzel, L.A., Dumont, A.S., Rosenwasser, R., Jabbour, P., 2013. Comparison of flow diversion and coiling in large unruptured intracranial saccular aneurysms. *Stroke* 44, 2150–2154. <https://doi.org/10.1161/STROKEAHA.113.001785>
- Cornelissen, B.M.W., Leemans, E.L., Coolen, B.F., Peper, E.S., van den Berg, R., Marquering, H.A., Slump, C.H., Majoie, C.B.L.M., 2019a. Insufficient slow-flow suppression mimicking aneurysm wall enhancement in magnetic resonance vessel wall imaging: a phantom study. *Neurosurg Focus* 47, E19. <https://doi.org/10.3171/2019.4.FOCUS19235>
- Cornelissen, B.M.W., Leemans, E.L., Slump, C.H., Marquering, H.A., Majoie, C.B.L.M., Berg, R. van den, 2019b. Vessel wall enhancement of intracranial aneurysms: fact or artifact? *Neurosurg Focus* 47, E18. <https://doi.org/10.3171/2019.4.FOCUS19236>

- Dang Luu, V., Xuan Bach, T., Huu An, N., Quang Anh, N., Anh Tuan, T., Hoang Kien, L., Tat Thien, N., Thu Trang, N., Cuong, T., 2024. Evaluation of Hemodynamic Alterations after Flow Diverter Placement using the AneurysmFlow™ tool. *Clin Ter* 175, 146–153. <https://doi.org/10.7417/CT.2024.5055>
- Edjlali, M., Guédon, A., Ben Hassen, W., Boulouis, G., Benzakoun, J., Rodriguez-Régent, C., Trystram, D., Nataf, F., Meder, J.-F., Turski, P., Oppenheim, C., Naggara, O., 2018. Circumferential Thick Enhancement at Vessel Wall MRI Has High Specificity for Intracranial Aneurysm Instability. *Radiology* 289, 181–187. <https://doi.org/10.1148/radiol.2018172879>
- Elsheikh, S., Urbach, H., Meckel, S., 2020. Contrast Enhancement of Intracranial Aneurysms on 3T 3D Black-Blood MRI and Its Relationship to Aneurysm Recurrence following Endovascular Treatment. *American Journal of Neuroradiology*. <https://doi.org/10.3174/ajnr.A6440>
- Falk, K.L., Medero, R., Roldán-Alzate, A., 2019. Fabrication of low-cost patient-specific vascular models for particle image velocimetry. *Cardiovasc Eng Technol* 10, 500–507. <https://doi.org/10.1007/s13239-019-00417-2>
- Gomyo, M., Tsuchiya, K., Goto, S., Hosoi, S., Tahara, T., Yokoyama, K., 2022. Usefulness of black-blood magnetic resonance angiography generated from vessel wall imaging after the stent-assisted treatment of intracranial arterial diseases. *Neuroradiol J* 35, 36–41. <https://doi.org/10.1177/19714009211021775>
- Guan, J., Karsy, M., McNally, S., de Havenon, A., Kalani, M.Y.S., Tausky, P., Kim, S.-E., Park, M.S., 2017. High-resolution magnetic resonance imaging of intracranial aneurysms treated by flow diversion. *Interdisciplinary Neurosurgery* 10, 69–74. <https://doi.org/10.1016/j.inat.2017.07.004>
- Hackenberg, K.A.M., Hänggi Daniel, Etminan Nima, 2018. Unruptured Intracranial Aneurysms. *Stroke* 49, 2268–2275. <https://doi.org/10.1161/STROKEAHA.118.021030>
- Hadad, S., Mut, F., Kadirvel, R., Ding, Y.-H., Kallmes, D., Cebral, J.R., 2021. Evaluation of Outcome Prediction of Flow Diversion for Intracranial Aneurysms. *American Journal of Neuroradiology* 42, 1973–1978. <https://doi.org/10.3174/ajnr.A7263>
- Henningsson, M., Malik, S., Botnar, R., Castellanos, D., Hussain, T., Leiner, T., n.d. Black-Blood Contrast in Cardiovascular MRI. *Journal of Magnetic Resonance Imaging* n/a, e27399. <https://doi.org/10.1002/jmri.27399>
- Henningsson, M., Malik, S., Botnar, R., Castellanos, D., Hussain, T., Leiner, T., n.d. Black-Blood Contrast in Cardiovascular MRI. *Journal of Magnetic Resonance Imaging* n/a, e27399. <https://doi.org/10.1002/jmri.27399>
- Hou, K., Li, G., Lv, X., Xu, B., Xu, K., Yu, J., 2020. Delayed rupture of intracranial aneurysms after placement of intra-luminal flow diverter. *Neuroradiol J* 33, 451–464. <https://doi.org/10.1177/1971400920953299>
- Júnior, J.R., Telles, J.P.M., da Silva, S.A., Iglesias, R.F., Brigido, M.M., Pereira Caldas, J.G.M., Teixeira, M.J., Figueiredo, E.G., 2019. Epidemiological analysis of 1404 patients with intracranial aneurysm followed in a single Brazilian institution. *Surgical Neurology International* 10, 249. https://doi.org/10.25259/SNI_443_2019
- Kalsoum, E., Chabernaud Negrier, A., Tuilier, T., Benaïssa, A., Blanc, R., Gallas, S., Lefaucheur, J.-P., Gaston, A., Lopes, R., Brugières, P., Hodel, J., 2018. Blood Flow Mimicking Aneurysmal Wall Enhancement: A Diagnostic Pitfall of Vessel

- Wall MRI Using the Postcontrast 3D Turbo Spin-Echo MR Imaging Sequence. *AJNR Am J Neuroradiol* 39, 1065–1067. <https://doi.org/10.3174/ajnr.A5616>
- Korte, J., Gaidzik, F., Larsen, N., Schütz, E., Damm, T., Wodarg, F., Hövener, J.-B., Jansen, O., Janiga, G., Berg, P., Pravdivtseva, M.S., 2023. In vitro and in silico assessment of flow modulation after deploying the Contour Neurovascular System in intracranial aneurysm models. *JNIS*. <https://doi.org/10.1136/jnis-2023-020403>
- Larsen, N., Flüh, C., Madjidyar, J., Synowitz, M., Jansen, O., Wodarg, F., 2020a. Visualization of Aneurysm Healing. *Clin Neuroradiol* 30, 811–815. <https://doi.org/10.1007/s00062-019-00854-5>
- Larsen, N., Flüh, C., Saalfeld, S., Voß, S., Hille, G., Trick, D., Wodarg, F., Synowitz, M., Jansen, O., Berg, P., 2020b. Multimodal validation of focal enhancement in intracranial aneurysms as a surrogate marker for aneurysm instability. *Neuroradiology*. <https://doi.org/10.1007/s00234-020-02498-6>
- Larsen, N., von der Brälie, C., Trick, D., Riedel, C.H., Lindner, T., Madjidyar, J., Jansen, O., Synowitz, M., Flüh, C., 2018. Vessel Wall Enhancement in Unruptured Intracranial Aneurysms: An Indicator for Higher Risk of Rupture? High-Resolution MR Imaging and Correlated Histologic Findings. *AJNR* 39, 1617–1621. <https://doi.org/10.3174/ajnr.A5731>
- Lee, G.-J., Eom, K.-S., Lee, C., Kim, D.-W., Kang, S.-D., 2015. Rupture of Very Small Intracranial Aneurysms: Incidence and Clinical Characteristics. *Journal of Cerebrovascular and Endovascular Neurosurgery* 17, 217. <https://doi.org/10.7461/jcen.2015.17.3.217>
- Ouared, R., Larrabide, I., Brina, O., Bouillot, P., Erceg, G., Yilmaz, H., Lovblad, K.-O., Mendes Pereira, V., 2016. Computational fluid dynamics analysis of flow reduction induced by flow-diverting stents in intracranial aneurysms: a patient-unspecific hemodynamics change perspective. *JNIS* 8, 1288–1293. <https://doi.org/10.1136/neurintsurg-2015-012154>
- Petridis, A.K., Suresh, M., Cornelius, J.F., Tortora, A., Steiger, H.J., Turowski, B., May, R., 2018. Aneurysm treatment response prediction in follow up black blood magnetic resonance imaging. A case series study. *Clin Pract* 8. <https://doi.org/10.4081/cp.2018.1047>
- Pravdivtseva, M.S., Gaidzik, F., Berg, P., Hoffman, C., Rivera-Rivera, L.A., Medero, R., Bodart, L., Roldan-Alzate, A., Speidel, M.A., Johnson, K.M., Wieben, O., Jansen, O., Hövener, J.-B., Larsen, N., 2021. Pseudo-Enhancement in Intracranial Aneurysms on Black-Blood MRI: Effects of Flow Rate, Spatial Resolution, and Additional Flow Suppression. *JMRI* 54, 888–901. <https://doi.org/10.1002/jmri.27587>
- Pravdivtseva, M.S., Gaidzik, F., Berg, P., Hoffman, C., Rivera-Rivera, L.A., Medero, R., Bodart, L., Roldan-Alzate, A., Speidel, M.A., Johnson, K.M., Wieben, O., Jansen, O., Hövener, J.-B., Larsen, N., n.d. Pseudo-Enhancement in Intracranial Aneurysms on Black-Blood MRI: Effects of Flow Rate, Spatial Resolution, and Additional Flow Suppression. *Journal of Magnetic Resonance Imaging* n/a. <https://doi.org/10.1002/jmri.27587>
- Pravdivtseva, M.S., Pravdivtsev, A.N., Peters, S., Hensler, J., Larsen, N., Hövener, J.-B., Jansen, O., Wodarg, F., 2023. The effect of the size of the new contour neurovascular device for altering intraaneurysmal flow. *Interv Neuroradiol* 15910199221145985. <https://doi.org/10.1177/15910199221145985>
- Quan, T., Ren, Y., Li, J., Fu, X., Jin, Y., Ran, Y., Guan, S., Cheng, J., Xu, H., 2023. Enhanced vessel wall magnetic resonance imaging in the follow-up of

- intracranial aneurysms treated with flow diversion. *Eur Radiol*.
<https://doi.org/10.1007/s00330-023-10094-4>
- Ravindran, K., Casabella, A.M., Cebal, J., Brinjikji, W., Kallmes, D.F., Kadirvel, R., 2020. Mechanism of Action and Biology of Flow Diverters in the Treatment of Intracranial Aneurysms. *Neurosurgery* 86, S13–S19.
<https://doi.org/10.1093/neuros/nyz324>
- Sadasivan, C., Dholakia, R., Peeling, L., Göllitz, P., Doerfler, A., Lieber, B.B., Fiorella, D.J., Woo, H.H., 2019. Angiographic assessment of the efficacy of flow diverter treatment for cerebral aneurysms. *Interv Neuroradiol* 25, 655–663.
<https://doi.org/10.1177/1591019919860829>
- Sarrami-Foroushani, A., Lassila, T., MacRaid, M., Asquith, J., Roes, K.C.B., Byrne, J.V., Frangi, A.F., 2021. In-silico trial of intracranial flow diverters replicates and expands insights from conventional clinical trials. *Nat Commun* 12, 3861.
<https://doi.org/10.1038/s41467-021-23998-w>
- Segur, J.B., Oberstar, H.E., 1951. Viscosity of Glycerol and Its Aqueous Solutions. *J Ind Eng Chem* 43, 2117–2120. <https://doi.org/10.1021/ie50501a040>
- Shimonaga Koji, Matsushige Toshinori, Ishii Daizo, Sakamoto Shigeyuki, Hosogai Masahiro, Kawasumi Tomohiro, Kaneko Mayumi, Ono Chiaki, Kurisu Kaoru, 2018. Clinicopathological Insights From Vessel Wall Imaging of Unruptured Intracranial Aneurysms. *Stroke* 49, 2516–2519.
<https://doi.org/10.1161/STROKEAHA.118.021819>
- Voß, S., Saalfeld, S., Hoffmann, T., Beuing, O., Janiga, G., Berg, P., 2018. Fluid-structure interaction in intracranial vessel walls: The role of patient-specific wall thickness. *Current Directions in Biomedical Engineering* 4, 587–590.
<https://doi.org/10.1515/cdbme-2018-0141>
- Wang, J., Yarnykh, V.L., Hatsukami, T., Chu, B., Balu, N., Yuan, C., 2007. Improved suppression of plaque-mimicking artifacts in black-blood carotid atherosclerosis imaging using a multislice motion-sensitized driven-equilibrium (MSDE) turbo spin-echo (TSE) sequence. *Magnetic Resonance in Medicine* 58, 973–981.
<https://doi.org/10.1002/mrm.21385>
- Xu, L., Sugawara, M., Tanaka, G., Ohta, M., Liu, H., Yamaguchi, R., 2016. Effect of elasticity on wall shear stress inside cerebral aneurysm at anterior cerebral artery. *Technology and Health Care* 24, 349–357. <https://doi.org/10.3233/THC-161135>
- Zarrinkoob, L., Ambarki, K., Wåhlin, A., Birgander, R., Eklund, A., Malm, J., 2015. Blood Flow Distribution in Cerebral Arteries. *J Cereb Blood Flow Metab* 35, 648–654. <https://doi.org/10.1038/jcbfm.2014.241>
- Zhou, G., Su, M., Yin, Y.-L., Li, M.-H., 2017. Complications associated with the use of flow-diverting devices for cerebral aneurysms: a systematic review and meta-analysis. *Neurosurg Focus* E17. <https://doi.org/10.3171/2017.3.FOCUS16450>